



# Geomorphic indices for unveiling fault segmentation and tectono-geomorphic evolution with insights into the impact of inherited topography, Ulsan Fault Zone, Korea

Cho-Hee Lee[1], Yeong Bae Seong[1], John Weber[2], Sangmin Ha[1], Dong-Eun Kim[3], Byung Yong Yu[4]

[1]Department of Geography, Korea University, Seoul, 02841, Republic of Korea
[2]Department of Geology, Grand Valley State University, Allendale, Michigan, 49401, USA
[3]Active Tectonics Research Center, Korea Institute of Geoscience and Mineral Resources, Daejeon, 34132, Republic of Korea
[4]Laboratory of Accelerator Mass Spectrometry, Korea Institute of Science and Technology, Seoul, 02792, Republic of Korea

*Correspondence to*: Yeong Bae Seong (ybseong@korea.ac.kr)

**Abstract.** Quantifying present topography can provide insights into landscape evolution and its controls, as the present topography is a cumulative expression of the types, distributions, and intensities of past and present processes. The Ulsan Fault Zone (UFZ) is an active fault zone on the southeastern Korean Peninsula that has been reactivated as a reverse fault around 5 Ma. This NNW–SSE-trending fault zone exhibits a predominantly reverse sense of movement today and dips towards the east. This study investigates the history of tectonic activity along the UFZ and the landscape evolution of the hanging wall side of the UFZ, focusing on neotectonic perturbations using $^{10}$Be-derived catchment-wide denudation rate and bedrock incision rates, geomorphic indices, and a landscape evolution model. We evaluated the spatial variation in the relative tectonic intensity from the variation in geomorphic indices along the UFZ. Five geological segments were identified along the fault based on the relative tectonic intensity and fault geometry. We then simulated four cases of landscape evolution using modelling to investigate the geomorphic processes and topographic changes in the study area in response to fault slip. The model results reveal that the geomorphic processes and the patterns of geomorphic indices (e.g., χ anomalies) depend on the inherited topography (i.e., the topography that existed prior to reverse faulting on the UFZ). On the basis of this important finding, we interpret the tectono-geomorphic history of the study area as follows: (1) the northern part of the UFZ has been in a transient state and is in topographic and geometric disequilibrium, as this part underwent asymmetric uplift (westward tilting) prior to reverse faulting on the UFZ around 5 Ma; and (2) its southern part was negligibly influenced by the asymmetric uplift before reverse faulting. Our study demonstrates geomorphic indices as reliable criteria for dividing faults into segments and, together with landscape evolution modelling, to investigate the influence of inherited topography on present topography and to help determine tectono-geomorphic histories.

**Short summary.** Geomorphic indices were used to understand topographic changes in response to tectonic activity. We applied indices to evaluate the relative tectonic intensity of Ulsan Fault Zone, one of the most active fault zones in Korea. We divided the UFZ into five segments based on spatial variation in intensity. We modelled the landscape evolution of study area and interpreted tectono-geomorphic history that the northern part of the UFZ experienced asymmetric uplift, while the southern part did not.



## 1 Introduction

Research in the field of tectonic geomorphology involves identifying the signal of neotectonic activity from geomorphic
characteristics. The classic approach to studies of tectonic geomorphology has been to use geomorphic indices and was developed
in the 1900s (e.g., hypsometric integral, stream length–gradient index, and mountain-front sinuosity; Strahler, 1952; Hack, 1973;
Bull, 1977; Cox, 1994; Keller and Pinter, 1996; Bull and McFadden, 2020). The normalised channel steepness index ($k_{sn}$; Flint,
1974) and knickpoint analyses are also frequently applied to explore the transient states of channels caused by their response to
tectonic activity (Whipple and Tucker, 1999; Duvall et al., 2004; Kirby and Whipple, 2012; Scherler et al., 2014; Marliyani et al.,
2016), as the incision of a channel system is the most obvious response to tectonic uplift. The chi ($\chi$) index was introduced to
handle problems associated with analysing $k_{sn}$ (Perron and Royden, 2013; Royden and Perron, 2013) and has enabled improved
determination of the dynamic evolution of a fluvial system in terms of the geometric equilibrium between tectonic forcing and
river incision (Willett et al., 2014; Forte and Whipple, 2018; Kim et al., 2020; Hu et al., 2021; Lee et al., 2021). As computational
power has improved and powerful modelling programs have become more widely available, it has become possible to simulate
landscape evolution, allowing a variety of site-specific parameters (e.g., coefficient of diffusivity, coefficient of erosion, and local
uplift rate) to be constrained and results to be visualised to facilitate the understanding of geomorphic processes and accompanying
topographic change within general or specific tectonic and climatic settings (Tucker et al., 2001; Braun and Willett, 2013; Goren
et al., 2014; Campforts et al., 2017; Hobley et al., 2017; Barnhart et al., 2020; Hutton et al., 2020). These advances have allowed
researchers to explain the state (steady state or transient state, and equilibrium or disequilibrium) of the present topography, to
interpret tectonic and/or climate events and their influence on landscape, and to predict future trends of landscape evolution within
neotectonically active areas (Attal et al., 2011; Reitman et al., 2019; Zebari et al., 2019; Su et al., 2020; He et al., 2021; Hoskins
et al., 2023).

Most of the above-mentioned tectonic geomorphology studies have focused on explaining how recent tectonic activity has
influenced landscape evolution and how geomorphic analyses can be applied to describe those influences. However, these studies
do not generally consider the effects of inherited topography (i.e., topography prior to the neotectonic events of interest) on
subsequent geomorphic processes, present topographic dynamics, and geomorphic indices. We show that the influence of inherited
topography is non-negligible; this follows in principle considering that: (1) the present topography is a cumulative expression of
tectonic and climatic events from the past to the present, (2) the response time for each geomorphic feature (e.g., longitudinal
stream profile, knickpoint migration, and divide migration) to the same tectonic event is different (Whipple et al., 2017), and (3)
the timescale that each geomorphic index represents is different and not yet fully understood (Forte and Whipple, 2018). Therefore,
we postulate that geomorphic indices reflect the cumulative influence of past and present geomorphic processes and their drivers,
and that drawing inferences from these indices without considering the influence of inherited topography can lead to incorrect
interpretations of landscape evolution.



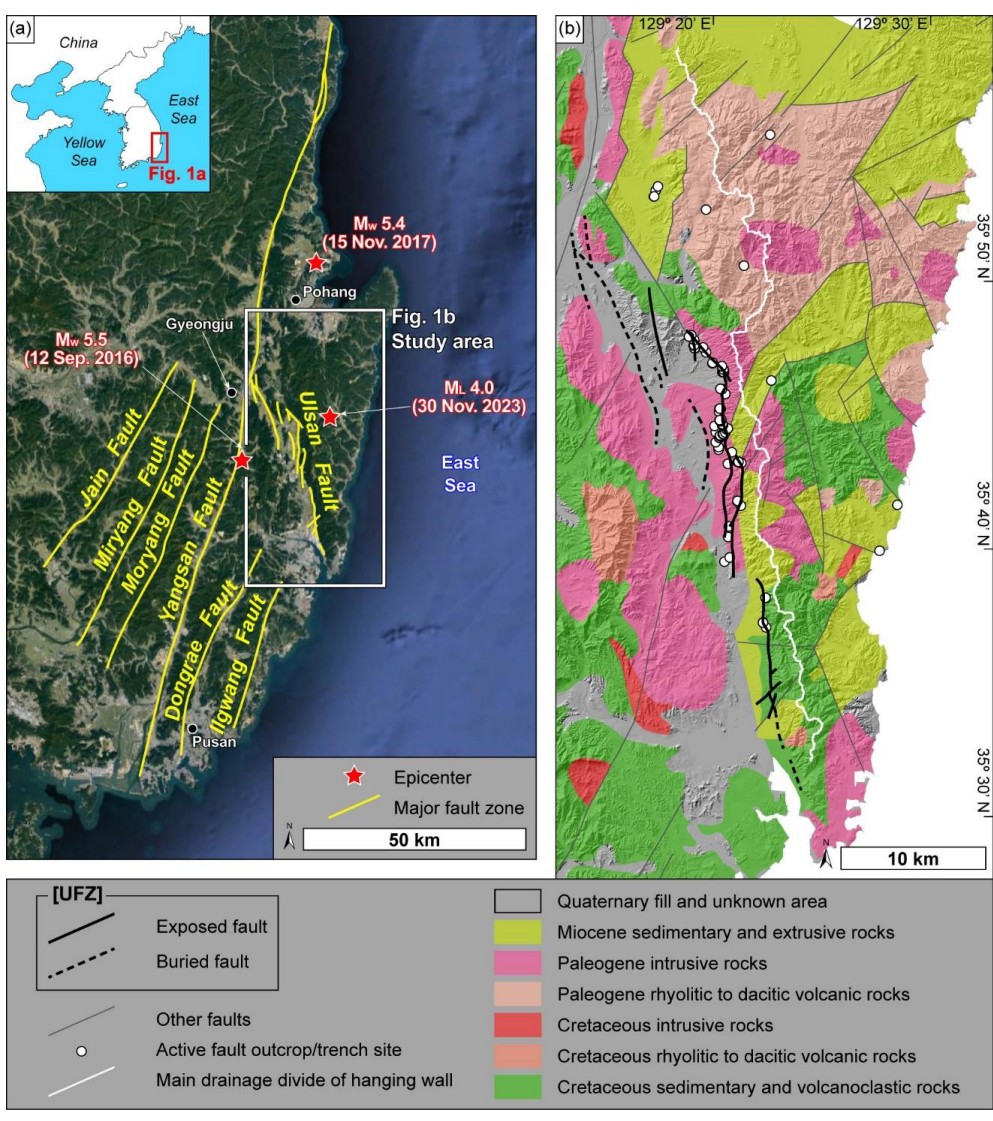

Figure 1: (a) Major fault zones on the southeastern Korean Peninsula (modified from Kim et al., 2016). Our study area is shown by the white box around the Ulsan Fault Zone (UFZ) (base map data: © 2022 Google, TerraMetrics). (b) Lithology in and around the UFZ (modified from Cheon et al., 2020b, 2023). Exposed faults occur along mountain fronts, and buried faults are located in a wide incised valley west of the mountain range. The hanging wall of the UFZ is on the eastern side of the fault zone and form the mountain range. The solid white line represents the main drainage divide of the hanging wall (i.e., the eastern block of the UFZ).

We target an area on the southeastern Korean Peninsula, around the Ulsan Fault Zone (UFZ), as our study area (Fig. 1). This area is somewhat uniquely poised for studying geology, tectonics, geomorphology, and the relationships between them. Many studies along the UFZ have initially reported active faults cutting unconsolidated Quaternary-Holocene sedimentary layers, peat layers, and fluvial terraces (Kyung, 1997; Okada et al., 1998; Cheong et al., 2003; Choi et al., 2012b; Kim et al., 2021). Since these pioneering works, three moderate earthquakes ($M_W$ 5.5 in 2016, $M_W$ 5.4 in 2017, and $M_L$ 4.0 in 2023) occurred around this area (Fig. 1a), and micro-earthquakes continue to swarm around and on the fault (Han et al., 2017). Studies have also established geological constraints on the boundary conditions for landscape evolution modelling and the long-term framework for interpreting



the influence of inherited topography on the present landscape evolution (Park et al., 2006; Cheon et al., 2012; Son et al., 2015; Kim et al., 2016b; Cheon et al., 2023; Kim et al., 2023a).

In this study, we assess the relative tectonic intensity along the UFZ using geomorphic indices for drainage systems that are relevant to the tectonic activity. We then trace variations in the geomorphic indices along the UFZ to describe the spatial distribution of the relative intensity of tectonic activity and use this distribution to divide the fault zone into geological segments by applying the criteria of McCalpin (1996). Evaluation of the relative tectonic intensity using geomorphic indices is particularly valuable in the study area. It is challenging to find surface deformation caused by neotectonic faulting in Korea due to low slip rates, rapid physical

and chemical erosion, and vast urbanisation. Next, we designed several models to simulate the landscape evolution of the study area in response to past and present tectonic activity and compare the geomorphic indices from the modelled topographies with those that we analysed for the study area. Finally, we interpreted the influence of inherited topography on the tectono-geomorphic evolution of the study area using the modelling results and geomorphic indices, which describe the cumulative influence of past and present geomorphic processes and tectonic activity.

## 2 Study area


Our study area includes the UFZ and its hanging wall (i.e., its eastern block). The UFZ is a NNW–SSE- to N–S-striking, east-dipping reverse fault that was first identified by the presence of an extensive incised valley and mountain front on the southeastern Korean Peninsula (Fig. 1; Kim, 1973; Kim et al., 1976; Kang, 1979a, b). Although the UFZ has been subject to considerable geological investigation, as it is one of the most active fault zones in Korea, its precise geometry and location and a full

understanding of its tectonic history remains elusive. Early studies proposed that the main strand of the UFZ is located within the incised valley (Kim, 1973; Kim et al., 1976; Kang, 1979a, b). However, subsequent studies have suggested that the UFZ is located either in and around the incised valley, or that it lies along the mountain front to the east of the incised valley, or possibly in both locations (Okada et al., 1998; Ryoo et al., 2002; Choi, 2003; Choi et al., 2006; Ryoo, 2009; Kee et al., 2019; Naik et al., 2022). A recent study attempted to comprehensively re-interpret previous studies along with adding new field observations and geophysical

data to propose a new definition of the UFZ (Cheon et al., 2023). This definition includes some strands of exposed faults along the mountain front and several strands of buried faults near the centre of the incised valley (Fig. 1b).



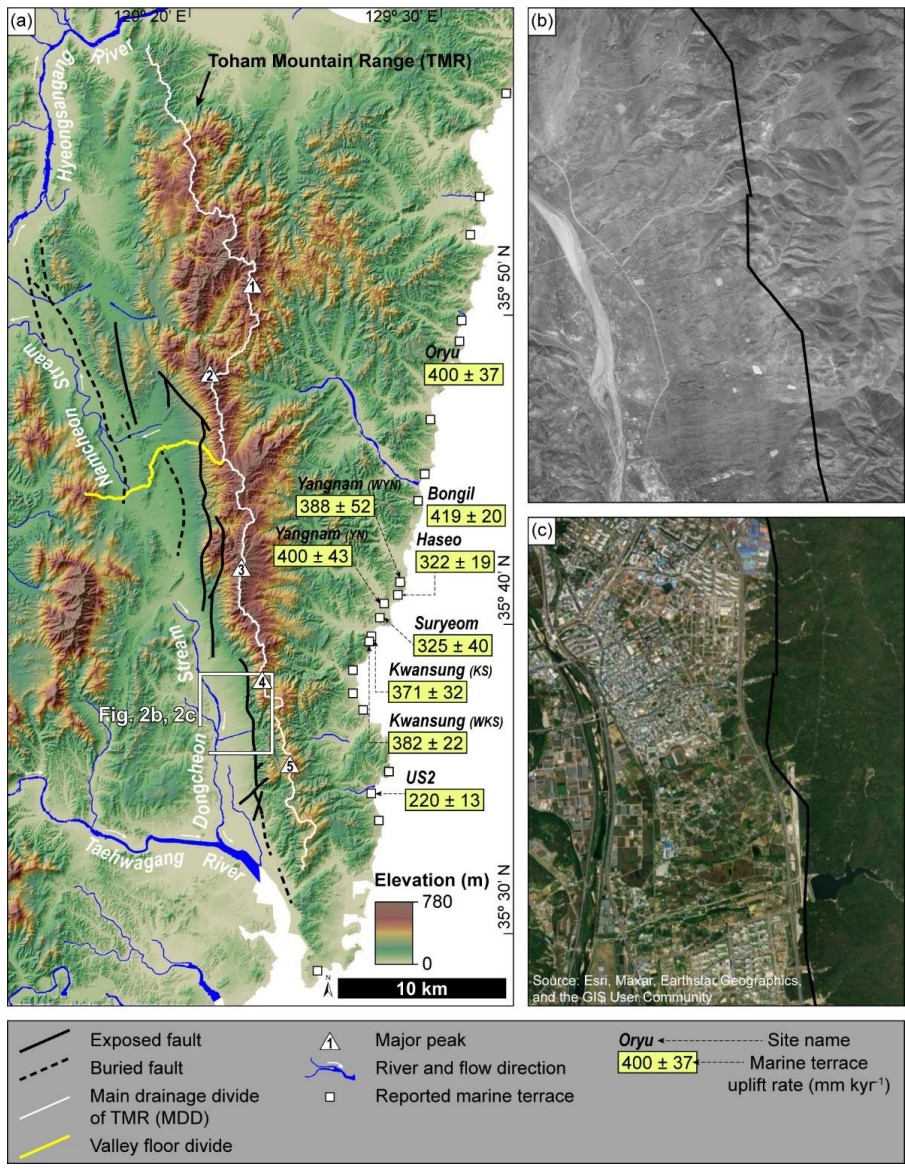

**Figure 2: (a) Previously determined uplift rates (in mm kyr⁻¹) of marine terraces near the UFZ (Choi et al., 2003a, b; Kim et al., 2007; Heo et al., 2014). The drainage system on the western flank of the Toham Mountain Range (TMR) is divided by the valley floor divide**
**(Namcheon and Dongcheon streams). Major peaks in the eastern block of the UFZ are marked by numbers in white triangles (1: Mt. Hamwol, 2: Mt. Toham, 3: Mt. Gwanmoon, 4: Mt. Dongdae, 5: Mt. Muryong). (b) Aerial photograph taken in 1954 of the area depicted by the white box in Fig. 2a. This aerial photograph was taken prior to urbanisation by the industrial complex and residential district in the study area (image source: National Geographic Information Institute of the Republic of Korea). Alluvial fans extend along the western flank of the mountains. The exposed Ulsan Fault (black line) is traced along the boundary between alluvial fans and the TMR.**
**(c) Recent satellite image (ArcMap^TM, ESRI) of the same area as that depicted in Fig. 2b. Urbanisation since the 1960s has made it difficult to observe the natural landforms in this area.**

The Toham Mountain Range (TMR) is located on the eastern hanging wall block of the UFZ and extends parallel–subparallel to the fault zone (Fig. 2a). The TMR includes many peaks, including Mt. Hamwol (584 m), Mt. Toham (745 m), Mt. Gwanmoon

(630 m), Mt. Dongdae (447 m), and Mt. Muryong (451 m) from north to south. Channels on the TMR are divided into eastern-



and western-flank channels by the main drainage divide (MDD; Fig. 2a). Channels on the eastern flank of the TMR flow to the east and drain directly into the East Sea, whereas those on the western flank form a more complex drainage system. The western-flank channels can be divided into northern and southern parts based on the characteristics of the valley floor divide. Channels in the northern part of the valley floor divide flow to the west and join together to form the Namcheon Stream. The Namcheon Stream flows to the north–northwest and joins other tributaries to form the Hyeongsangang River, which drains into the East Sea. Channels in the southern part of the valley floor divide flow to the west and join to form the Dongcheon Stream, which flows to the south. The Dongcheon Stream joins the Taehwagang River, which drains into the Southern Sea of the Korean Peninsula. The landscapes of the western and eastern flanks differ significantly from each other: the western flank is dominated by a clearly defined mountain front with extensive alluvial fans developed along this mountain front (Fig. 2a and 2b), whereas the eastern flank has broader mountainous and hilly landscape that extends from the TMR all the way to the eastern coast (Figs 1a and 2a). The cause of the contrasting landscapes of the western and eastern flanks of the TMR has yet to be unequivocally established, and several explanations have been proposed, including: (1) differential regional rift-margin uplift related to the opening of the East Sea from ca. 20 Ma (Min et al., 2010; Kim et al., 2016a, 2020); (2) differential regional uplift caused by accommodation of the ENE–WSW-oriented neotectonic maximum horizontal stress since 5 Ma (Park et al., 2006; Kim et al., 2016b); and (3) differences in late Quaternary uplift between the western and eastern coasts of the Korean Peninsula, as recorded in marine terraces along the eastern coast (Choi et al., 2003a, b, 2008, 2009; Lee et al., 2015) and shore platform along the western coast (Choi et al., 2012a; Jeong et al., 2021).

In addition, numerous studies have attempted to elucidate the geological and geomorphic history of the southeastern Korean Peninsula. Studies of the UFZ have reported many active faults (Fig. 1b), but age data from those studies need further verification as at present these results lack consensus (Kyung, 1997; Okada et al., 1998, 2001; Cheong et al., 2003; Kim et al., 2021). Further, studies of marine terraces have proposed palaeo-shoreline elevations and ages of beach-sediment layers for each terrace sequence (Choi et al., 2003a, b; Kim et al., 2007; Heo et al., 2014). In this study, we calculated the amount of uplift of each terrace considering local palaeo-sea levels and terrace uplift rates (Table 1 and Fig. 2a).



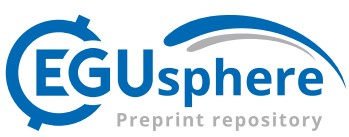

**Table 1: Information on previously studied marine terraces within the study area.**

| Site name[a] | Latitude (°N, dd) | Longitude (°E, dd) | Palaeo-shoreline elevation[b] (m a.s.l.) | Uplift amount[c] (m) | OSL age[d] (ka) | MIS | Uplift rate (mm kyr⁻¹) | Reference |
|---|---|---|---|---|---|---|---|---|
| Oryu | 35.8173 | 129.5112 | 26 | 26 | 65 ± 6 | 4 | 400.00 ± 36.92 | (Choi et al., 2003a) |
| Bongil | 35.7301 | 129.4871 | 26 | 26 | 62 ± 3[†] | 4 | 419.35 ± 20.29 | (Kim et al., 2007) |
| Yangnam (WYN) | 35.6894 | 129.4708 | 26 | 26 | 67 ± 9 | 4 | 388.06 ± 52.13 | (Choi et al., 2003b) |
| Haseo | 35.6801 | 129.4719 | 45 | 39 | 121 ± 7[†] | 5e | 322.31 ± 18.64 | (Kim et al., 2007) |
| Yangnam (YN) | 35.6774 | 129.4607 | 26 | 26 | 65 ± 7 | 4 | 400.00 ± 43.08 | (Choi et al., 2003a) |
| Suryeom | 35.6706 | 129.4571 | 45 | 63 | 194 ± 24 | 7 | 324.74 ± 40.17 | (Heo et al., 2014)[††] |
| Kwansung (KS) | 35.6617 | 129.4516 | 26 | 26 | 70 ± 6 | 4 | 371.43 ± 31.84 | (Choi et al., 2003a) |
| Kwansung (WKS) | 35.6580 | 129.4504 | 26 | 26 | 68 ± 4 | 4 | 382.35 ± 22.49 | (Choi et al., 2003b) |
| US2 | 35.5764 | 129.4517 | 18.5 | 18.2 | 84 ± 5 | 5a | 220.23 ± 13.11 | (Unpublished data)[†††] |

[a] List of sites runs north to south, with some sites sharing names but having different sampling locations (shown in parentheses after each code).

[b] The palaeo-shoreline elevation is based on the present sea level (0 m a.s.l.).

[c] Uplifted amount is calculated considering the palaeo-sea level of each marine isotope stage which corresponds to each marine terrace age. We considered local palaeo-sea level (Lee et al., 2015; Ryang et al., 2022) in our calculations.

[d] Mean ages and 1σ standard deviations are given. These studies used the beach sediment to infer the depositional age.

[†] This age is the average of two samples.

[††] This study applied single grain OSL dating method, while the other studies applied single aliquot OSL dating method.

[†††] Age data is from 'Big Data Open Platform' (https://data.kigam.re.kr/map/) managed by 'Korea Institute of Geoscience and Mineral Resources (KIGAM)'.






## 3 Methods

### 3.1 Morphometric analysis

Previous studies of active faults used a variety of geomorphic indices to infer the relative tectonic intensity, these include stream gradient index, mountain-front sinuosity, valley floor width to valley height ratio, asymmetry factor, hypsometric integral, basin shape index, and drainage density (Harkins et al., 2005; El Hamdouni et al., 2008; Ahmad et al., 2015; Topal et al., 2016; Cao et al., 2022). However, the western front of TMR marking the proposed position of the UFZ has been highly modified by urbanisation
(Figs. 2b and 2c), so applying geomorphic indices related to the range front morphology (i.e., mountain-front sinuosity and valley floor width to valley height ratio) is not appropriate. Some of these indices, such as the stream gradient index and hypsometric integral, can however be substituted for by using similar but more recently proposed geomorphic indices (e.g., the channel steepness index and χ index). Further, the study area likely involves low fault slip rates and high rates of physical and chemical erosion, making it difficult to observe the vertical displacement by neotectonic faulting on the surface. As a result, we adopted and
used alternative morphometries, including swath profile, Gilbert metrics, the χ index, and the normalised channel steepness index ($k_{sn}$), to assess relative tectonic intensity. These morphometries have been widely used to evaluate topography and geomorphic processes over a wide range of tectonic and climatic settings. Gilbert metrics and the χ index have been applied to assess divide stability with respect to tectonic activity, climatic characteristics, and lithological variations (Willett et al., 2014; Forte and Whipple, 2018; Kim et al., 2020; Zondervan et al., 2020). The channel steepness index positively correlates with erosion and uplift rates
(Kirby and Whipple, 2001; DiBiase et al., 2010; Harel et al., 2016). Although the elevation of the swath profile and topographic relief are not the same as cumulative vertical displacement, these two morphometries can reasonably be used as a proxy to infer the latter.

Variation in the normalised channel steepness index along a longitudinal stream profile and the shape of the χ-transformed stream profile can be used to indicate whether a channel or channel reach is in a steady or transient state. We compared the geomorphic
indices of the eastern and western flanks of TMR to identify whether this landscape is in geometric equilibrium, as geometric disequilibrium (e.g., χ anomaly) is a strong indicator of the presence of tectonic perturbations. We used a 5-m-resolution digital elevation model (DEM) to analyse the morphometry of the study area, which was generated using digital contours provided by the National Geographic Information Institute (NGII) of the Republic of Korea (https://www.ngii.go.kr/kor/main.do; accessed 14 Sep 2020).

### 3.1.1 Swath profile

Swath profiles are traditional and simple means of illustrating surface elevation and relief. Swath profiles can be used to investigate and understand the relationship between surface topography and associated or causative variables, such as dynamic topography, which is a topographic change caused by mantle convection (Stephenson et al., 2014), precipitation (Bookhagen and Burbank, 2006), and uplift and exhumation rates (Taylor et al., 2021). We extracted a swath profile along the MDD for the area with a width
of 3 km centred on the MDD which lies parallel to the UFZ, using TopoToolbox (Schwanghart and Scherler, 2014), as along-strike topographic variation is expected to be related to the along-strike variation in the cumulative vertical displacement on the UFZ.

### 3.1.2 Normalised steepness index ($k_{sn}$)

The bedrock channel incision rate, E, can be expressed by Eq. (1), which describes its relationship with channel bed shear stress (Howard and Kerby, 1983; Seidl and Dietrich, 1992; Sklar et al., 1998):



$E = KA^m S^n$          (1)

where K is a dimensional coefficient of erosion with a unit of $[L^{1-2m}T^{-1}]$ and includes the influence of rock resistance, climate, bedload sediment grain size, and channel width–length relationship (Stock and Montgomery, 1999; Whipple and Tucker, 1999; Snyder et al., 2000; Whipple and Tucker, 2002); A is drainage area; S is the slope; and m and n are exponents of drainage area and slope, respectively.

According to Eq. (1), the change in channel elevation (z) with respect to time (t) is:

$\frac{dz}{dt} = U - E = U - KA^m S^n$          (2)

where U is rock uplift rate (Whipple and Tucker, 1999; Snyder et al., 2000; Tucker and Whipple, 2002). If the channel adjusts to a tectonic perturbation and thus attains a steady state, then uplift rate and bedrock channel incision rate will balance each other (dz/dt = 0), assuming that the bedrock properties and climatic characteristics across the entire channel or catchment are uniform.

Then, the channel can maintain a graded profile, following a power-law equation (Hack, 1973; Flint, 1974):

$S = k_s A^{-\theta}$          (3a)

$S = k_{sn} A^{-\theta_{ref}}$          (3b)

where $\theta$ is the concavity index of a channel or channel reach ($\theta$ = m/n). The channel steepness index ($k_s$) may be changed by the concavity index, and this makes it difficult to compare values of the channel steepness index with those of other channels with

different concavity index values and different sizes of drainage basins. To facilitate such a comparison, the normalised channel steepness index ($k_{sn}$) can be calculated by fixing the concavity index with a reference value ($\theta_{ref}$) in the range of 0.36–0.65 (Eq. (3b); Snyder et al., 2000; Wobus et al., 2006; Cyr et al., 2010; Kirby and Whipple, 2012). However, many streams in nature are not graded, particularly if they have undergone base-level changes that resulted from climate change (Crosby and Whipple, 2006), tectonic forcing (Snyder et al., 2000; Kirby and Whipple, 2001), or lithological differences (Cyr et al., 2014). Such streams show

peaks or piecewise-fitted lines in a log S–log A plot and display abrupt variation in $k_{sn}$ along their course, indicating a transient state. We set $\theta_{ref}$ to 0.45 and used LSDTopoTools (Mudd et al., 2014) to compute $k_{sn}$.

### 3.1.3 Gilbert metrics and the chi ($\chi$) index

Gilbert metrics, including mean upstream relief, mean upstream gradient, and channel elevation can be used to assess divide stability (Forte and Whipple, 2018) based on the 'law of divides' of Gilbert (1877). According to this law, there are two opposing

sides of a divide. The steeper side is expected to be eroded and reduced in height more rapidly when compared with the behaviour of the gently sloping side; therefore, the divide should migrate towards the gentle side (Fig. 70 in Gilbert, 1877). The migration will in principle continue until the two sides become symmetric (geometric equilibrium) (Gilbert, 1877). In addition to these metrics, the chi ($\chi$) index at opposing channel heads can also be used to evaluate divide stability (Willett et al., 2014; Forte and Whipple, 2018). The $\chi$ index at a point x on the channel serves as a proxy for the steady-state elevation of the channel and is calculated by

integrating Eq. (3b) from downstream to upstream (Perron and Royden, 2013):

$z(x) = z_b + \left( \frac{k_{sn}}{A_0^{\theta_{ref}}} \right) \chi$          (4a)

$\chi = \int_{x_b}^{x} \left( \frac{A_0}{A(x')} \right)^{\theta_{ref}} dx'$          (4b)

where x' is a dummy variable for x, $z_b$ is a base-level elevation (at x = $x_b$), $A_0$ is an arbitrary scaling area, and A(x) is the drainage area at point x on the channel. The integrand in Eq. (4b) becomes dimensionless, meaning that the $\chi$ index can be expressed with

a unit of length by multiplying by $A_0$ as a coefficient (Perron and Royden, 2013). Equation (4a) illustrates the linear relationship between elevation and the $\chi$ index for a steady-state channel. The scaling area, $A_0$, is set to unity, as the slope of the $\chi$ index–



channel elevation plot ($\chi$–z plot) is equal to $k_{sn}$, based on Eq. (4a). Because the $\chi$ index is sensitive to the base-level elevation ($z_b$; Forte and Whipple, 2018), we analysed the $\chi$ index with two different base-level elevations. We set the base-level elevations as 50 and 200 m a.s.l. for appropriate analysis, since only two drainages (Namcheon and Dongcheon drainages; Fig. 2a) were extracted

when we set a base-level elevation of <50 m. Additionally, there were less than 10 drainages exceeding the threshold drainage area ($10^5$ m$^2$) with an elevation of >200 m in the southern part of study area. We used TopoToolbox (Schwanghart and Scherler, 2014) and DivideTools (Forte and Whipple, 2018) to analyse Gilbert metrics and the $\chi$ index. We then used Student's t-test (two-tailed, $\alpha$ = 0.05) to statistically compare the values of these geomorphic indices between the western and eastern flanks of the TMR.

### 3.1.4 Longitudinal and χ-transformed stream profiles and knickpoint analysis

According to Eq. (3a), a graded stream has a concave longitudinal profile and is represented as a single line on a log S–log A plot. The $\chi$-transformed stream profile of a graded stream ($\chi$–z plot) would be represented by a single line, based on Eq. (4a). However, rivers in transient states are expected to show several piecewise linear segments in a log S–log A plot and $\chi$-transformed stream profile (Perron and Royden, 2013). The boundary between adjacent piecewise lines can be identified physically as knickpoints. A knickpoint can reflect the transient state of a stream that is caused by a base-level change related to climatic change (Crosby and

Whipple, 2006), tectonic forcing (Snyder et al., 2000; Kirby and Whipple, 2001), or an encountered lithological difference (Cyr et al., 2014).

We used TopoToolbox (Schwanghart and Scherler, 2014) and LSDTopoTools (Mudd et al., 2014) to extract longitudinal stream profiles and $\chi$-transformed stream profiles. We set the reference concavity index ($\theta_{ref}$) to 0.45 and the reference scaling area ($A_0$) to unity. Then, we detected the locations of knickpoints using LSDTopoTools (Mudd et al., 2014; Gailleton et al., 2019). This

algorithm automatically extracts knickpoints in a quantitative and reproducible way, thereby avoiding: (1) arbitrary interpretation of slope–area data, (2) the generation of non-reproducible results, and (3) difficulties in detecting knickpoints resulting from noise in slope–area data directly without using the integral method.

### 3.2 In situ cosmogenic [10]Be measurements

Assuming that the channel of interest approaches a steady state, uplift rate can be derived from the bedrock channel incision rate

[Eqs. (1) and (2)]. We used in situ cosmogenic [10]Be measurements to constrain the catchment-averaged denudation rate and bedrock channel incision rate in order to quantify the uplift rate and the stream power variation controlled by tectonic uplift in the study area.

### 3.2.1 Catchment-averaged denudation rate

The concentration of in situ cosmogenic [10]Be from riverine sediment on the present bedrock channels represents the catchment-

averaged denudation rate (CADR). This approach assumes the existence of a geochemical steady state whereby the production and removal (via denudation) rates of cosmogenic [10]Be within the catchment are equal (Brown et al., 1995; Bierman and Steig, 1996; Granger et al., 1996; von Blanckenburg, 2005). Thus, the CADR represents the average rate of denudation over an entire catchment by hillslope and fluvial processes during a given integration time, during which the sediments remained within the catchment (Granger et al., 1996; von Blanckenburg, 2005). The integration time documented in previous studies from a variety of tectonic,

climatic, and topographic environments is in the range of $10^3$–$10^6$ years (Brown et al., 1995; DiBiase et al., 2010; Portenga et al., 2015; Kim et al., 2020).







**Figure 3: Sampling sites and results of catchment-averaged denudation rates (CADRs) and channel incision rates derived from *in situ* cosmogenic ¹⁰Be measurement. (a) CADRs calculated using *in situ* cosmogenic ¹⁰Be measurements and their sampling sites. We collected samples for ¹⁰Be analysis and CADR calculation from eight pairs of basins (16 basins) along the main drainage divide. CADR values on the western flank of the TMR are mostly higher than those on the eastern flank. (b) Bedrock strath sampling sites. (c and d) Photographs of the bedrock strath sampling sites on the western and eastern flanks of the TMR, respectively. We collected samples from the bedrock strath and the present stream bed in the same catchments from which we collected samples W4 and E4 for CADR calculation. The height of the western-flank strath above the present stream bed is 4.1 m, and the height of the eastern-flank strath above the present stream bed is 2.73 m. (e and f). Elevation profiles across the bedrock strath sampling sites and their ¹⁰Be exposure ages on the western and eastern flanks of the TMR, respectively. The age difference between the present stream bed and the strath is 2.94 ± 0.15 kyr on the western flank and 12.68 ± 0.25 kyr on the eastern flank.**

We collected 16 samples of riverine sediment from eight pairs of catchments (a total of 16 catchments) along the MDD of the TMR (Fig. 3a) to trace variations in the CADR along the MDD and to compare the CADRs of the western and eastern flanks of the TMR. The along-MDD variation and across-MDD contrasts were subsequently compared with results from our morphometric analysis to characterise the tectonic intensity and its spatial variability. We obtained samples of fine- to medium-grained sand (250–500 μm) from channel beds. We avoided collecting samples from: (1) catchments containing golf courses and (2) downstream areas where alluvial fans are located, and faults occur (Fig. 2) to avoid possible contamination by anthropogenic debris. The





lithology of the sampled catchments includes mainly sedimentary rocks and igneous rocks of various geological ages (Fig. 1b). The lithology within each pair of basins (basins contacting at the MDD, such as basins W1 and E1 in Fig. 3a) is, however, highly similar, which avoids any influence of lithological difference on CADRs for comparison in the across-MDD direction. However, some lithological variations do occur in the along-MDD direction. The basins W1 and E1 contain rhyolite and dacite bedrock. The basins W2, W3, E2, and E3 contain rhyolite, dacite, and granite bedrock. The other basins (W4−W8 and E4−E8; eight basins)

contain sedimentary, volcanoclastic, and granite bedrock.

We performed chemical treatment of the CADR samples at Korea University, Seoul, South Korea, following the standard protocol for $^{10}Be$ extraction (Kohl and Nishiizumi, 1992; Seong et al., 2016). We leached the samples with an HCl–HNO$_3$ mixture to remove organic and carbonate materials. Then, we used an HF–HNO$_3$ mixture to remove minerals other than quartz and meteoric $^{10}Be$ adsorbed onto the surface of mineral particles. An amount of 15–20 g of pure quartz was yielded after separating magnetic minerals

and picking out other impurities. A $^{9}Be$ carrier with a low background level of $^{10}Be$ was then added to the samples, which were then dissolved with a high-concentration HF–HNO$_3$ mixture. We extracted beryllium using an ion-exchange column, precipitated it into BeOH, dried the BeOH gel, and calcined it into BeO. The samples in BeO form were mixed with niobium powder and targeted into the cathode. Accelerator mass spectrometry measurements were performed at the Korea Institute of Science and Technology (KIST), Seoul, South Korea. Measured $^{10}Be/^{9}Be$ results were normalised to the 07KNSTD reference 5-1 sample

(Nishiizumi et al., 2007) and calculated as $^{10}Be$ concentrations after correction with a process blank (4.37–4.53 × 10$^{-15}$; n = 6).

We utilised the BASINGA (basin average scaling factors, cosmogenic production, and denudation rates) tool (Charreau et al., 2019) to calculate CADRs and integration time from $^{10}Be$ concentrations. This tool calculates the basin-averaged production rate of in situ cosmogenic $^{10}Be$ from every cell of a DEM based on its location and topography. The tool requires raster files of a DEM and topographic shielding and provides the scaling schemes of Lal/Stone (Lal, 1991; Stone, 2000), LSD, and LSDn (Lifton et al., 2014)

and geomagnetic correction based on the virtual dipole moment (Muscheler et al., 2005). We used the same topographic data as those used for morphometric analysis (5-m-resolution DEM based on the digital contours of NGII) and topographic shielding raster calculated using the algorithm of Mudd et al. (2016). We applied the LSDn scaling scheme (Lifton et al., 2014) and geomagnetic correction (Muscheler et al., 2005).

### 3.2.2 Bedrock channel incision rate

The classical model of fluvial strath terrace formation includes widening of the terrace tread by lateral erosion and its abandonment by incision (Burbank and Anderson, 2011). Each abandoned terrace represents the position of the palaeo-channel bed, and bedrock incision is controlled by uplift, as channels incise bedrock while they attain steady state [Eq. (2)]. If the concentration of in situ cosmogenic $^{10}Be$ of a strath surface can be measured, then the exposure age of that bedrock strath can be calculated, which indicates the time elapsed after abandonment of the strath surface.

We collected three samples from western-flank straths and two from eastern-flank straths (Fig. 3) to constrain the exposure age of each tread. The sampled strath terraces are located in the drainage basin from which the W4 and E4 CADR samples were taken. The height of the strath terrace from the channel bed on the western flank was 4.10 m, and that on the eastern flank was 2.73 m (Figs. 3c–3f). On the western flank, the valley is deep and narrow, and the valley wall is steep. On the eastern flank, the valley is wide and gentle, and the exposed valley wall and terrace riser are more weathered than those on the western flank. The terraces in

both valleys are unpaired.

Following laboratory protocol (Kohl and Nishiizumi, 1992; Seong et al., 2016), we performed physical and chemical treatment for *in situ* surface exposure dating samples at Korea University, Seoul, South Korea. We crushed bedrock samples using a jaw crusher and iron mortar and separated fine- to medium-sized sand (250–500 μm) grains by sieving. The further chemical treatments were



the same as those applied to our CADR samples (see section 3.2.1 above). We calculated exposure ages using the CRONUS-Earth

online calculator (Balco et al., 2008; version 3), applying the LSDn scaling scheme (Lifton et al., 2014). Error ranges of exposure

ages were calculated and are given as 1σ values.

### 3.3 Modelling landscape evolution

We next applied the open-source landscape evolution model 'Landlab' (Hobley et al., 2017; Barnhart et al., 2020; Hutton et al., 2020) to comprehensively investigate the geomorphic evolution of the uplifted eastern hanging wall block of the UFZ. The use of

this modelling programme enabled us to verify our results from morphometric analysis and $^{10}$Be measurements and, in conjunction with measured geomorphic indices, to interpret the landscape evolution of the study area. We considered two processes that erode topography and transport sediment: (1) fluvial erosion and (2) hillslope diffusion.

Topographic change caused by fluvial erosion is controlled by the stream power incision law (Howard and Kerby, 1983; Seidl and Dietrich, 1992; Sklar et al., 1998), following Eq. (1). We used values of K = 5.56E-07 m$^{-1.29}$ yr$^{-1}$, m = 1.1448, and n = 2.2896 to

simulate fluvial erosion, which was estimated by averaging values calculated for regions with similar lithology, climate, and tectonic activity to those of our study area (Harel et al., 2016). We applied an incision threshold of 1.0E-05 m yr$^{-1}$, below which no incision is assumed to occur (Tucker and Whipple, 2002; Harel et al., 2016; Hobley et al., 2017).

Topographic change caused by hillslope diffusion is controlled by the diffusion equation (Culling, 1963; Tucker and Bras, 1998):

$$\frac{\partial z}{\partial t} = K_d \nabla^2 z \qquad (5)$$

where $K_d$ is the coefficient of diffusivity; $\nabla^2$ is the Laplace operator, which is the divergence of gradient; and z is elevation. We

used $K_d$ = 0.001 to simulate the hillslope diffusion process, which we adopted because soil is rare on slopes (Fernandes and Dietrich, 1997; Zebari et al., 2019).

The landscape gain in height by tectonic uplift and loss of height by fluvial erosion and hillslope diffusion can be expressed as (Temme et al., 2017):

$$\frac{\partial z}{\partial t} = U - KA^m S^n - K_d \nabla^2 z \qquad (6)$$



Figure 4: Diagram showing configuration of the landscape evolution model (LEM) used to simulate the tectono-geomorphic evolution of the eastern hanging wall block of the UFZ. (a) The two stages of the LEM. The stage 1, corresponding to a duration of 3 Myr, involves simulation of building of the initial topography; i.e., the topography prior to reverse faulting of the UFZ during the Quaternary. The stage 2, corresponding to a duration of 2 Myr, involves simulation of reverse faulting and associated neotectonic surface uplift on the UFZ during the Quaternary. We modelled the location of the fault in the LEM as being 2 km west of the average location of the main drainage divide (MDD) of the initial topography. (b) The four model cases (A1-A2, B1-B2) used to test different conditions of spatial uniformity/non-uniformity of uplift during stage 1 and the width of the modelled area. (c) Detail settings for each case (A1-A2, B1-B2) of the LEM. The settings in the first four rows of this table are universal to all four cases. We applied different uplift rates and spatial gradients in uplift during both stages 1 and 2 for each case. 'U' means average uplift rate (in a unit of mm kyr$^{-1}$), 'W' means the western boundary of the eastern block of the UFZ, and 'E' means the eastern boundary of the eastern block of the UFZ. The numbers in the parentheses represent the uplift amounts at every 20 kyr (i.e., one uplift event cycle). Model uplift rate during stage 1 for Cases A1 and A2 is spatially uniform, whereas that for Cases B1 and B2 is spatially variable, decreasing linearly from east to west according to the criteria listed in Fig. 4b. In the second-to-bottom row of the table, the red triangle denotes the location of the fault, and the blue triangle marks the location of the MDD of initial topography. The uplift rate during stage 2 decreases linearly with increasing distance from the fault. The uplift rates and their spatial gradient during stage 2 depend on the width of the modelled area. Cases #1 share the same uplift rate and its spatial gradient, and Cases #2 have the same values for the uplift rate and its spatial gradient.






We designed the landscape evolution model incorporating two stages (Fig. 4a). The first stage is a pre-Quaternary period during

which initial topography is built; i.e., the topography that already existed before reverse faulting of the UFZ during the Quaternary. This period simulates the regional uplift prior to the Quaternary reverse faulting of the UFZ. The second stage is a period in which to simulate local uplift by reverse faulting, representing neotectonic movement of the UFZ during the Quaternary. In the model, we structured stage 1 to last for 3 Myr and stage 2 to last for 2 Myr, giving a total time of 5 Myr. The total duration corresponds to the duration of the present stress regime, as the regional and local uplift both occurred under the present stress regime (Park et

al., 2006; Kim et al., 2016b).

With this model structure, we tested four cases differentiated by varying two parameters: (1) spatial uniformity of uplift rate in the first stage, and (2) the width of the modelled area (Fig. 4b). First, the cases can be separated into two groups (A and B) based on the spatial uniformity of uplift rate during stage 1. The cases simulating a spatially uniform uplift rate during stage 1 (henceforth 'Cases A#') assume that there was no spatial gradient in uplift rate, namely, that the whole eastern block of the UFZ underwent

uniform uplift during stage 1. The cases simulating a spatially variable uplift rate during stage 1 (henceforth 'Cases B#') assume that there was a spatial gradient in uplift rate whereby the eastern side of this block was uplifted more than the western side (i.e., the modelled area tilted westward). This assumption is based on the overall tendency of high-east and low-west topography of the Korean Peninsula, supported by the long-term, regional westward tilting that was initiated during the Middle Miocene when the East Sea started to widen, and since which time the strongly asymmetric (high-east) Taebaek Mountain Range has been rapidly

uplifted (Min et al., 2010; Kim et al., 2020). In addition, the shore platform on the western coast of the peninsula (0 m a.s.l.; Choi et al., 2012a; Jeong et al., 2021) and marine terraces along the eastern coast (18–45 m a.s.l.; Choi et al., 2003a, b; Kim et al., 2007; Heo et al., 2014; Lee et al., 2015), formed at the same time (i.e., during MIS 5), indicate that this regional differential uplift has lasted until very recently. Second, we divided the cases into two groups (henceforth 'Cases #1 and #2') based on the width of the modelled area (Fig. 4b) to simulate the observation that the eastern block of the UFZ is wide in its northern part and narrows

towards the south (Fig. 1). The width of the wide-modelled area (measured in an E–W direction) is 20 km (henceforth 'Cases #1'), and that of the narrow-modelled area (henceforth 'Cases #2') is 7.5 km, so that Cases #1 and Cases #2 represent the northern and southern parts of the block, respectively.

In all four cases, we employed identical values for the following settings and parameters: (1) the length of the modelled area in the N–S direction; (2) the location of the fault; (3) the parameters used to simulate fluvial and hillslope processes; and (4) the uplift

event cyclicity (Fig. 4c). The N–S length of the modelled area was set to 20 km for all cases. We positioned the model Ulsan Fault 2 km west of the average location for the MDD of the initial topography (Figs. 4a and 4c). This was done because the present-day location of the UFZ is approximately 2 km west of the MDD of the eastern block. The three parameters associated with fluvial process: the coefficient of erosion (K) and exponents of area (m) and slope (n), and the one parameter associated with slope processes; i.e., the coefficient of diffusivity ($K_d$; as described above; Fig. 4c) were set to constants. Finally, we set the uplift event

cyclicity (i.e., the duration between discrete faulting and uplift events) to 20 kyr. Although the earthquake recurrence interval has not yet been definitively determined for the Ulsan Fault, we used a realistic value based on the correlation between earthquake magnitude, recurrence interval, and geomorphic evidence proposed by Slemmons and Depolo (1986), as well as the timing of the most recent and penultimate earthquakes in the study area (Cheon et al., 2020a; Kim et al., 2023b).

We applied different average uplift rates and their spatial gradients during both stages for particular cases (Fig. 4c). The average

uplift rates during stage 1 for Cases A1 and A2 were spatially uniform (i.e., no spatial gradient). The average uplift rate for Cases A# during stage 1 was set to 80 mm kyr$^{-1}$. This value was chosen to set our model uplift rate to be the same as the long-term exhumation rate across the Taebaek Mountain Range (the "backbone" mountain range of the Korean Peninsula) since 22 Ma (Han,



2002; Min et al., 2010; Kim et al., 2016a). Conversely, Cases B1 and B2 incorporate a spatial gradient in uplift rate, with the highest uplift rate in the east, decreasing gradually towards the west. Although the spatial gradient of uplift rate is uncertain, we
chose to model the average uplift rate at the western margin of the eastern block in Case B1 (40 mm kyr$^{-1}$) as half of the maximum uplift rate at the eastern margin (80 mm kyr$^{-1}$), which is equivalent to the uplift rate of Cases A# during stage 1 (Fig. 4c). The same spatial gradient in uplift rate (-2 mm kyr$^{-1}$ km$^{-1}$) was applied in Case B2.

During stage 2 (Quaternary reverse faulting), the average uplift rate is set to be the highest at the location of the fault and to diminish linearly with increasing distance from the fault. To determine the maximum vertical displacement per event, we assumed
that a maximum earthquake magnitude of $M_W$ 7.0 once per 20 kyr is not unreasonable (Slemmons and Depolo, 1986; Kyung, 2010), although different maximum magnitude estimates ($M_W$ 4.6−5.6) have been proposed for the Ulsan Fault (Choi et al., 2014). According to the empirical equation of Moss and Ross (2011), a $M_W$ 7.0 earthquake would generate a maximum vertical displacement of approximately 2.36 m. Therefore, we hypothesised a scenario in which a $M_W$ 7.0 earthquake produces a maximum vertical displacement on the fault of 2.36 m every 20 kyr. Under this scenario, the average long-term surface uplift rate at the fault
location for Cases #1 is 118 mm kyr$^{-1}$ (0.118 mm yr$^{-1}$) as calculated by dividing the maximum vertical displacement (2.36 m) by 20 kyr (Fig. 4c). This rate decreases linearly to 18 mm kyr$^{-1}$ (0.018 mm yr$^{-1}$) at a location 2.5 km east of the MDD of the initial topography. This value (18 mm kyr$^{-1}$) is calculated by multiplying the average uplift rate at the fault location (118 mm kyr$^{-1}$) by the ratio of the eastern-flank channel incision rate (215 mm kyr$^{-1}$) to that in the west (1394 mm kyr$^{-1}$). This calculation reflects the fact that the sampled western-flank strath is located ~2 km west of the UFZ, and the eastern-flank strath is ~2.5 km from the MDD.
For Cases #2, representing the southern part of the block, we applied a lower average uplift rate of 42 mm kyr$^{-1}$ (0.042 mm yr$^{-1}$) at the fault location (Fig. 4c). We used this lower uplift rate because CADRs in the southern part of the study area are lower than those in the northern part (Table 2 and Fig. 3a). This uplift rate (42 mm kyr$^{-1}$) was calculated by multiplying the ratio of the average CADR of W6–W8 (35.95 mm kyr$^{-1}$) to the CADR of W4 (99.91 mm kyr$^{-1}$) by 118 mm kyr$^{-1}$. This choice in parameterization reflects that the western-flank strath terrace is located within the drainage basin from which we collected the W4 CADR sample.
The uplift rate becomes zero 2 km east of the MDD because most of the knickpoints on the eastern-flank channels in the southern part of the study area are located within 2 km of the MDD (Fig. 5a).

Each of the four landscape evolution model cases has a grid spacing of 100 m. We traced the change in topography using a time-step of 100 yr. Comparisons between the resultant topographies from Case A1 to Case B1 and from Case A2 to Case B2 allow us to detect the influence of initial topography on the subsequent geomorphic response to the same pattern of tectonic movement (i.e.,
uplift by faulting during stage 2). Similarly, comparisons of the resultant topographies from Case A1 to Case A2 and from Case B1 to Case B2 enable us to detect the differential geomorphic response controlled by differences in the width of the modelled area or in channel length. In addition, our model results can be used to verify our results obtained from geomorphic indices analysis, CADRs, and channel incision rates calculation from $^{10}$Be measurement, as these were used as inputs for the simulation. We analysed Gilbert metrics and the χ index for the modelled topographies using TopoToolbox (Schwanghart and Scherler, 2014) and
DivideTools (Forte and Whipple, 2018) to quantitatively compare the topographies generated in the four cases and to compare the modelled topographies with the observed topography in the study area.



## 4 Results

### 4.1 Morphometric analysis

#### 4.1.1 $k_{sn}$ and knickpoint analyses on stream profiles



**Figure 5: (a) Spatial distribution of knickpoints on trunk stream channels and map of normalised steepness index ($k_{sn}$) for the entire study area. A reference concavity index ($\theta_{ref}$) of 0.45 was applied to calculate values of $k_{sn}$. Knickpoints (white circle) were detected after excluding artefacts and lithological boundaries. (b and c) Plots of d–z (blue), χ–z (green), and χ–$k_{sn}$ (pink) for a stream on each of the western and eastern flanks, respectively, where d is distance from the outlet and z is elevation. The numbers in pink are the $k_{sn}$ values of each reach of channel separated by the knickpoints. Knickpoints detected at artefacts and lithological boundaries are marked with grey crosses. (d) Longitudinal profiles and knickpoints of all trunk channels in the study area. Knickpoints detected at the artefacts and lithological boundaries are excluded.**







Analyses of $k_{sn}$ and knickpoints on the longitudinal and χ-transformed stream profiles show that the channels on both (western and

eastern) sides are in a transient state (Fig. 5). This result implies that these channels have been disturbed either by lithologies with

different K values or by base-level change. We excluded artefact knickpoints (e.g., known anthropogenic features such as dams

and reservoirs) and lithological boundaries by examining satellite images and geological maps and performing checks in the field.

The remaining knickpoints can be interpreted as being caused by tectonic events, and are in accordance with the findings of a

previous study (Kim et al., 2016a), which suggested on the basis of a 1-D model that the observed major knickpoints in the study

area cannot have been formed by sea level changes since the global Last Glacial Maximum.

**4.1.2 Variations in values of geomorphic indices in the along- and across-MDD directions**

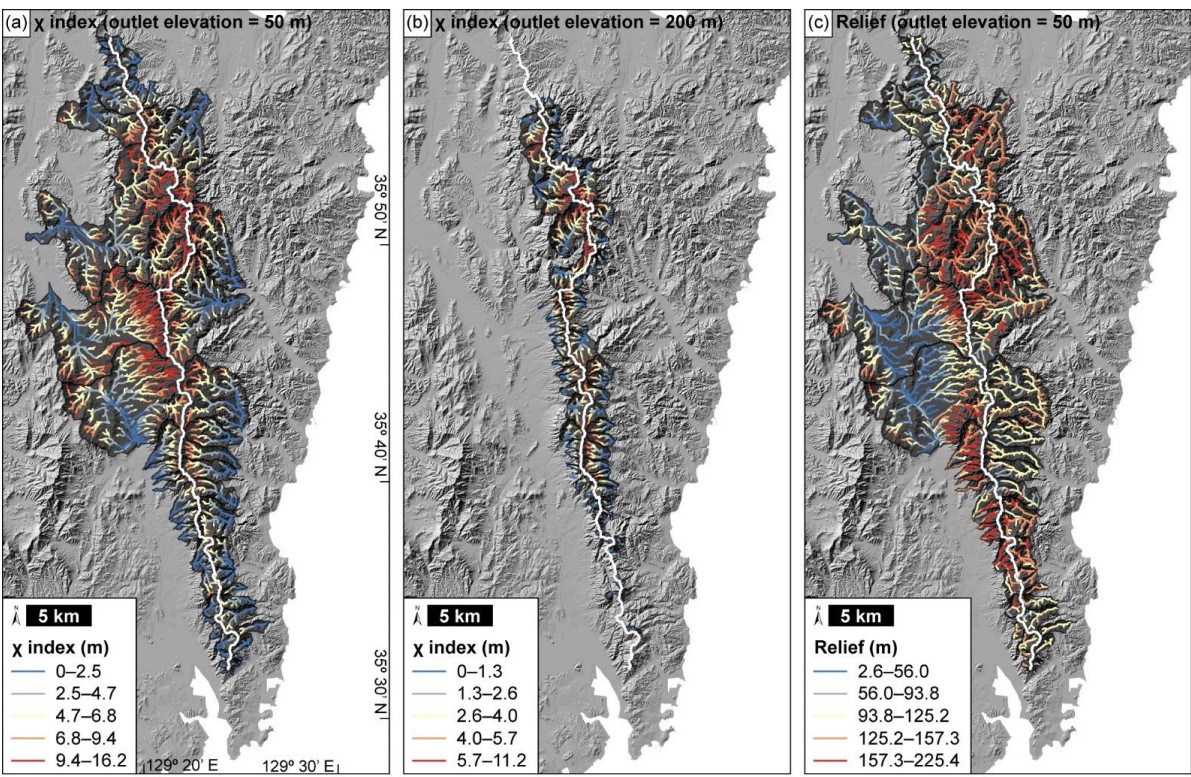

**Figure 6: (a and b) Variations in χ index values analysed with base-level elevations of 50 and 200 m, respectively. A reference concavity index ($\theta_{ref}$) value of 0.45 was applied, and the reference drainage area ($A_0$) was set to unity for calculation. χ index values at channel heads on the western flank are higher than those on the eastern flank in the northern part of the study area, whereas the χ indices on the eastern flank are higher than those on the western flank in the southern part of the study area. (c) Upstream average relief calculated within a radius of 200 m. Relief values at channel heads are mostly higher on the western flank than those at the channel heads on the eastern flank.**

We plotted our morphometric results (Fig. 6) in several ways to determine whether and if so how the morphometric parameters

vary along and across the MDD (Figs. 7 and 8). The along-MDD variation in each morphometry shows that the locations of highs

and lows in their values are similar for all parameters (Fig. 7). On the horizontal axis of Fig. 7, which represents the distance along

the MDD from its northern end, highs for both elevation and values of geomorphic indices appear near 25, 40, 60, and 90 km, and

lows appear near 16, 38, 48, and 86 km. The high near 40 km and the low near 38 km on the horizontal axis are the most pronounced

of the various highs or lows.

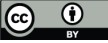

Figure 7: Variation in each morphometry on the western flank in the along-MDD direction and segmentation for the UFZ based on the results of geomorphic analysis. (a) Swath profile along the MDD for area width of 3 km centred on the MDD. The green-shaded area represents the minimum to maximum elevation range in the swath profile. (b) Mean normalised steepness index ($k_{sn}$). (c and d) Average relief and slope (within a radius of 200 m) for the upstream area at channel heads. (e) Mean χ index at channel heads. (f) Mean channel head elevation. The presented geomorphic indices were extracted from the drainage basins of the western flank of the TMR. Base-level elevations ($z_b$) were set to values of 50 m (solid red line) and 200 m (solid orange line). The horizontal axis is the distance along the main drainage divide of the TMR developed on the hanging wall of the Ulsan Fault from the north. Blue-shaded areas are segment boundaries inferred from the fault geometry and the results of geomorphic analysis.







**Figure 8: Along-MDD variations in values of geomorphic indices and CADRs for the western and eastern flanks of the TMR. (a) Mean χ index at channel heads. (b) Mean channel head elevation. (c and d) Average relief and slope (within a radius of 200 m) for the upstream area at channel heads. (e) Mean normalised steepness index ($k_{sn}$). The base-level elevation ($z_b$) was set to a value of 50 m for calculation of the geomorphic indices. (f) Catchment-averaged denudation rate (CADR). Red (blue) solid lines and symbols represent the indices and CADRs along the main drainage divide of the western (eastern) flank of the TMR.**



There are some significant differences in geomorphic indices between those for the western and eastern flanks along the MDD (Figs. 8a–8e). The χ values are contrasted across 60 km: the western flank are 27% higher than the eastern flank between 0–60 km,

whereas the χ index values between the western and eastern flanks do not show a significant difference between 60–90 km (Fig. 8a). The channel head elevations of the western and eastern flanks are generally similar, except for the 40–60 km section on the horizontal axis (Fig. 8b). The mean upstream relief and mean upstream gradient in the along-MDD direction share a similar pattern, with the values for the western flank being higher than those for the eastern flank for 40–90 km section but similar to each other in the 0–40 km section (Figs. 8c and 8d). Values of $k_{sn}$ for the eastern-flank channels are higher than those for the western-flank

channels within the 0–60 km section, whereas those for the western-flank channels are higher than those for the eastern-flank channels within the 60–90 km section (Fig. 8e).

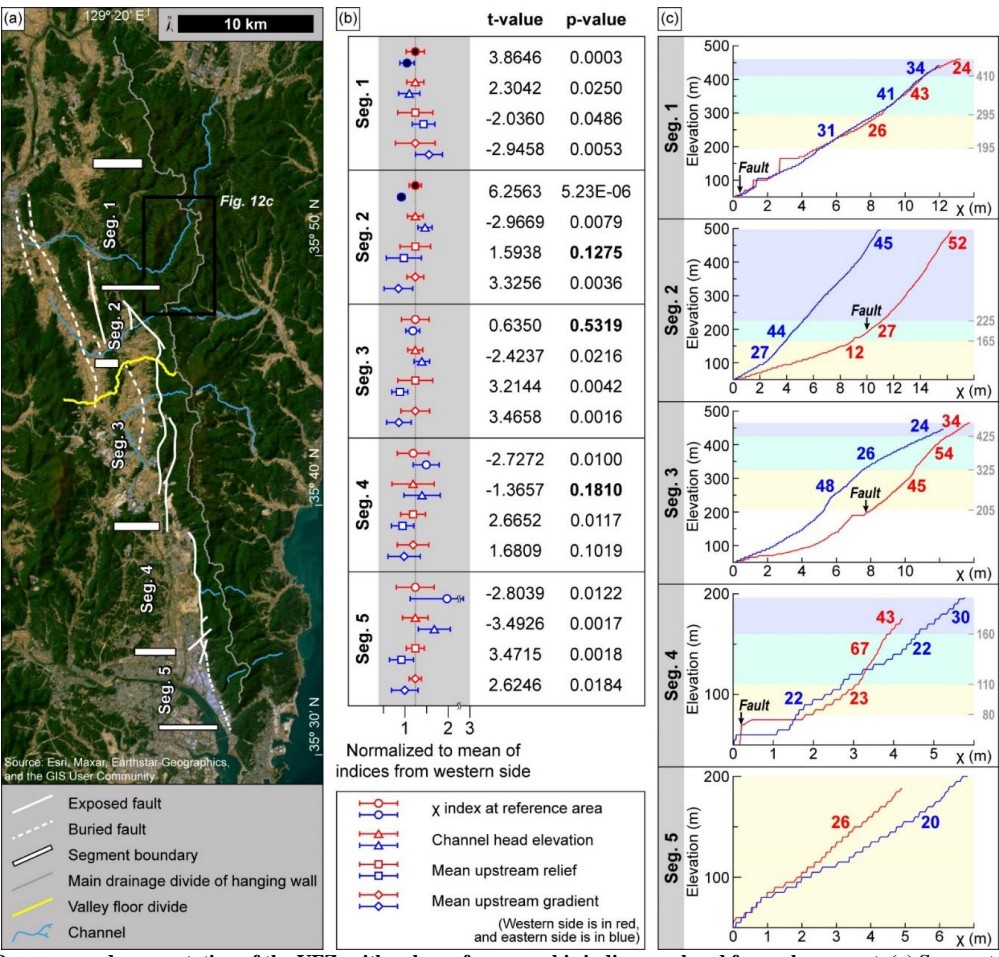

**Figure 9: Our proposed segmentation of the UFZ, with values of geomorphic indices analysed for each segment. (a) Segmentation of the UFZ proposed in this study. Segment boundary locations are the same as those marked in Figs. 7 and 8. The segment boundaries are**

**marked with white bars, and the main drainage divide of the TMR developed on the hanging wall of the UFZ is marked with a solid grey line. The area shown in Fig. 12c is demarcated with black rectangle. The base map is from ArcMap©, ESRI. (b) Mean values and 1σ standard deviation of geomorphic indices within each segment, normalised to the mean value of each index from the western flank. Geomorphic indices on the western flank are in red, and those on the eastern flank are in blue. Symbols filled with black represent decoupled χ indices. We performed Student's t-tests to compare these indices between the western and eastern flanks. P-values exceeding**

**the threshold of 0.05 (statistical significance of 95 %) are given in bold indicate that the mean values of the western- and eastern-flank indices are not statistically significantly different. (c) χ-transformed profiles for one western-flank and one eastern-flank channel from each of the segments shown in Fig. 9a. The numbers represent $k_{sn}$ of each reach of channel. The profiles for channels on the western flank are plotted with red line, and those on the eastern flank are plotted with blue line.**



**4.2 In situ cosmogenic ¹⁰Be**

**4.2.1 Variation in CADR within the study area**

CADRs on the western flank range from $16.79 \pm 0.99$ to $154.00 \pm 15.23$ mm kyr$^{-1}$, and those on the eastern flank range from $7.32 \pm 0.43$ to $104.29 \pm 7.68$ mm kyr$^{-1}$ (Table 2 and Fig. 3a). The integration times of these denudation rates cover the interval range of 5–97 kyr, during which the rates are implicitly assumed to have been steady. We plotted the CADRs in the along-MDD direction

(Fig. 8f) and identified two main patterns, as follows. First, the CADRs on the western and eastern flanks are the highest near the central part of the MDD (W3: $154.00 \pm 15.23$ mm kyr$^{-1}$ and E3: $104.29 \pm 7.68$ mm kyr$^{-1}$) and gradually decrease towards the northern and southern ends (Fig. 3). The CADRs in the vicinity of a distance of 70 km from the north end of the MDD (the horizontal axis of Fig. 8), which were obtained using samples W5 and E6, are higher than those obtained from adjacent samples (W4 and W6; E5 and E7) (Fig. 3). This pattern contrasts with the main spatial trend of CADR but corresponds to the patterns

shown by the other geomorphic indices (Figs. 7 and 8). Second, CADRs on the western flank are generally higher than those on the eastern flank. There is one exception at the southern end of the MDD (W8: $16.79 \pm 0.99$ mm kyr$^{-1}$ and E8: $44.11 \pm 2.80$ mm kyr$^{-1}$), where the CADRs on the eastern flank are higher than those on the western flank (Fig. 3). The significant difference between the western- and eastern-flank CADRs at the southern end (W8 and E8) (Fig. 3) may be contributed by lithological differences (Fig. 1b). The lithologies within the upstream areas of the western-flank (W8) and eastern-flank (E8) sampling sites consist of

73.2% and 87.5% of sedimentary rocks from Cretaceous or Miocene continental basins, respectively, and the rest part of those upstream areas consist of Cretaceous rhyolitic to dacitic rocks, or Paleogene intrusive rocks.

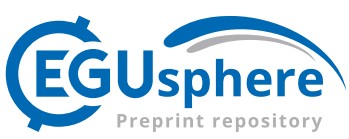

**Table 2: Catchment-averaged denudation rates calculated from cosmogenic in situ ¹⁰Be measurement.**

| Flank | Sample name | Latitude (°N, dd) | Longitude (°E, dd) | Elevation (m) | Topographic shielding[a] | Production rate[b] Spallation (atoms g⁻¹ yr⁻¹) | Production rate[b] Muon (atoms g⁻¹ yr⁻¹) | ¹⁰Be conc.[c,d] (10⁴ atoms g⁻¹) | Denudation rate[d,e] (mm kyr⁻¹) | Integration time[e] (kyr) |
|---|---|---|---|---|---|---|---|---|---|---|
| Western flank | W1 | 35.9409 | 129.3009 | 70 | 0.9828 | 3.63 | 0.049 | 7.69 ± 0.15 | 32.85 ± 2.00 | 22.15 |
| | W2 | 35.8230 | 129.3412 | 190 | 0.9674 | 4.27 | 0.051 | 5.24 ± 0.13 | 55.56 ± 3.54 | 13.24 |
| | W3 | 35.7825 | 129.3402 | 245 | 0.9490 | 4.23 | 0.051 | 1.87 ± 0.15 | 154.00 ± 15.23 | 4.82 |
| | W4 | 35.6964 | 129.3511 | 205 | 0.9966 | 4.45 | 0.052 | 3.02 ± 0.15 | 99.91 ± 7.69 | 7.41 |
| | W5 | 35.6634 | 129.3550 | 153 | 0.9289 | 4.11 | 0.050 | 2.55 ± 0.13 | 110.48 ± 8.45 | 6.70 |
| | W6 | 35.6284 | 129.3719 | 113 | 0.8279 | 3.73 | 0.049 | 4.65 ± 0.15 | 55.32 ± 3.63 | 13.72 |
| | W7 | 35.6026 | 129.3868 | 95 | 0.8509 | 3.71 | 0.049 | 7.19 ± 0.16 | 35.74 ± 2.20 | 20.66 |
| | W8 | 35.5506 | 129.3965 | 65 | 0.8270 | 3.57 | 0.048 | 14.80 ± 0.20 | 16.79 ± 0.99 | 43.73 |
| Eastern flank | E1 | 35.9188 | 129.3527 | 165 | 0.9199 | 4.07 | 0.050 | 7.67 ± 0.20 | 36.38 ± 2.30 | 20.37 |
| | E2 | 35.8213 | 129.3978 | 106 | 0.8932 | 3.98 | 0.050 | 6.80 ± 0.14 | 40.17 ± 2.47 | 18.59 |
| | E3 | 35.7923 | 129.3659 | 230 | 0.9992 | 4.39 | 0.051 | 2.86 ± 0.13 | 104.29 ± 7.68 | 7.00 |
| | E4 | 35.7062 | 129.3964 | 187 | 0.8500 | 4.19 | 0.051 | 8.26 ± 0.17 | 34.18 ± 2.10 | 23.74 |
| | E5 | 35.6612 | 129.3947 | 155 | 0.9745 | 4.13 | 0.050 | 38.78 ± 0.45 | 7.32 ± 0.43 | 97.24 |
| | E6 | 35.6400 | 129.3896 | 144 | 0.8781 | 3.88 | 0.049 | 7.74 ± 0.15 | 34.46 ± 2.10 | 21.54 |
| | E7 | 35.6021 | 129.4092 | 75 | 0.8572 | 3.69 | 0.049 | 14.27 ± 0.21 | 17.93 ± 1.07 | 40.67 |
| | E8 | 35.5670 | 129.4355 | 81 | 0.8103 | 3.52 | 0.048 | 5.56 ± 0.15 | 44.11 ± 2.80 | 16.78 |

[a] Topographic shielding was computed cell-by-cell (Mudd et al., 2016) and averaged for the catchment above the sampling site (Charreau et al., 2019). We used the same DEM with that we used for morphometric analysis.

[b] Catchment-averaged production rate of in situ ¹⁰Be was computed (Charreau et al., 2019), applying the LSDn scaling scheme (Lifton et al., 2014).

[c] Process blank (4.37–4.53×10⁻¹⁵; n = 6) was used for correction of background, and ratios of ¹⁰Be/⁹Be were normalized with 07KNSTD reference sample 5-1 (2.71×10⁻¹¹ ± 4.71×10⁻¹⁵) of (Nishiizumi et al., 2007).

[d] Mean values and 1σ uncertainties are used.

[e] ¹⁰Be half-life of 1.38×10⁶ yr (Chmeleff et al., 2010), density of the sample (ρ) of 2.7 g cm⁻³, and attenuation length (Λ) of 160 g cm⁻² (Braucher et al., 2011) were used for calculation of denudation rate and integration time.



### 4.2.2 Channel incision rates derived from $^{10}$Be exposure ages of straths

We calculated channel incision rates using $^{10}$Be exposure ages of bedrock straths and the present bedrock channel bed (Table 3).
On the western flank, the exposure age of the present channel bed is $2.07 \pm 0.12$ kyr. The strath is 4.10 m higher than the channel

bed, and two samples from the tread yield consistent $^{10}$Be exposure ages of $5.11 \pm 0.13$ and $4.91 \pm 0.13$ kyr (mean: $5.01 \pm 0.09$
kyr) (Fig. 3). Dividing the height of the strath by the age difference between them {4100 mm / [$(5.01 \pm 0.09) - (2.07 \pm 0.12)$] kyr}
yields a channel incision rate of $1394.56 \pm 71.15$ mm kyr$^{-1}$. On the eastern flank, the exposure age of the present channel bed is
$12.30 \pm 0.15$ kyr, and the age of the strath, which is 2.73 m higher than the channel bed, is $24.98 \pm 0.20$ kyr. The calculated incision
rate {2730 mm / [$(24.98 \pm 0.20) - (12.30 \pm 0.12)$] kyr)} is $215.30 \pm 4.24$ mm kyr$^{-1}$. Therefore, at the locations studied, the incision

rate on the western flank is approximately 6.5 times higher than that on the eastern flank.



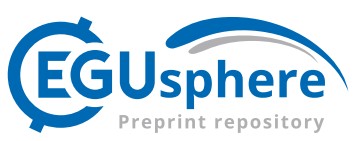

**Table 3: Cosmogenic $^{10}$Be surface exposure ages of bedrock strath terraces.**

| Sample name | Latitude (° N, dd) | Longitude (° E, dd) | Elevation (m) | Thickness (cm) | Topographic shielding | Quartz mass[a] (g) | Carrier mass (g) | $^{10}$Be/$^{9}$Be[b, c] ($10^{-14}$) | $^{10}$Be conc.[c, d] ($10^3$ atoms g$^{-1}$) | Exposure age[c, e] (kyr) |
|---|---|---|---|---|---|---|---|---|---|---|
| WT0-1 | 35.6985 | 129.3514 | 232 | 5.0 | 0.9098 | 21.277 | 0.438 | 1.35 ± 0.04 | 6.20 ± 0.36 | 2.07 ± 0.12 |
| WT1-1 | 35.6985 | 129.3514 | 236 | 3.5 | 0.9968 | 20.620 | 0.413 | 2.15 ± 0.07 | 16.48 ± 0.40 | 5.11 ± 0.13 |
| WT1-2 | 35.6985 | 129.3514 | 236 | 2.5 | 0.9968 | 20.328 | 0.387 | 2.17 ± 0.08 | 15.96 ± 0.41 | 4.91 ± 0.13 |
| ET0-1 | 35.7069 | 129.3921 | 207 | 4.0 | 0.9478 | 20.036 | 0.371 | 4.22 ± 0.12 | 40.71 ± 0.48 | 12.30 ± 0.15 |
| ET1-1 | 35.7069 | 129.3922 | 209 | 5.0 | 0.9518 | 20.171 | 0.440 | 7.16 ± 0.23 | 89.78 ± 0.73 | 24.98 ± 0.20 |

[a] Density of rock ($\rho$) of 2.7 g cm$^{-3}$ was used.

[b] Ratios of $^{10}$Be/$^{9}$Be were normalized with 07KNSTD reference sample 5-1 (2.71×10$^{-11}$ ± 4.71*10$^{-13}$) (Nishiizumi et al., 2007) and $^{10}$Be half-life of 1.38×10$^{6}$ yr (Chmeleff et al., 2010).

[c] Mean values and 1σ uncertainties are used.

[d] Process blank (4.37–4.53×10$^{-15}$; n = 6) was used for correction of background.

[e] Ages are calculated assuming zero erosion via CRONUS-Earth online calculator (version 3.0) (Balco et al., 2008) with scaling factors, applying the LSDn scaling scheme (Lifton et al., 2014).




### 4.3 Landscape evolution modelling

The MDDs of the initial topographies in Cases A#, which were the models simulated using spatially uniform uplift rate during stage 1, occupy their positions in the centre of the modelled areas (Figs. 10a and A1a). In both Cases A1 and A2, after stage 1, the χ indices at the channel heads on the western flank are comparable to those on the eastern flank (Figs. 10a and A1a). However, the MDDs of the initial topographies in Cases B#, which were the models simulated using a spatial gradient in uplift rate during stage 1, are biased towards the eastern flank of the modelled area, and the χ indices at the channel heads are lower on the eastern flank than on the western flank in both Cases B1 and B2 after stage 1 (Figs. 11a and A2a). These differences in the modelled positions

of the MDDs and the pattern of χ indices from initial topographies are results of the difference in the spatial uniformity of uplift rate during stage 1 (uniform versus variable).

### 4.3.1 Cases A1 and A2

Cases A1 and A2 involved spatially uniform regional uplift during stage 1, followed by spatially variable local uplift related to

faulting during stage 2. The modelling results for Cases A1 and A2 show similarities in drainage configuration (Figs. A1 and 10). During stage 2, the drainage patterns of the initial topographies in both cases undergo minimal change throughout the modelled area. The MDDs in both cases remain virtually static, and the channels retain their original routes during stage 2 (Figs. A1a, A1b, 10a, and 10b). The spatial distribution of χ indices in both cases also shows negligible differences compared with the initial topography. In addition, the statistical patterns of geomorphic indices for the resultant topographies at the end of stage 2 are almost

identical in the two cases (Figs. A1c and 10c). In both cases, the western- and eastern-flank χ indices after stage 2 show no statistical difference (p-value > 0.05), and the channel head elevations and mean upstream gradients on the eastern flank are lower than those on the western flank (p-value < 0.05). However, these patterns are inconsistent with the spatial distribution of uplift rate during stage 2. The western flank would show lower χ indices and channel head elevation and higher mean upstream relief and gradient compared with the eastern flank, as the model simulated a higher uplift rate on the western flank than the eastern flank during stage

565    2.

**Figure 10: Modelled landscape evolution for Case A2.** (a) Initial topography and χ index after stage 1, simulating a spatially uniform uplift rate of 80 mm kyr⁻¹ over the modelled area. (b) Topography and χ index after stage 2, simulating fault movement. (c) Histograms, mean values, and 1σ standard deviations for geomorphic indices (χ index, channel elevation, mean upstream relief, and mean upstream gradient) at the channel heads on the western and eastern flanks of the MDD, extracted from the modelled topography at the end of stage 2.



### 4.3.2 Cases B1 and B2

**Case B1**

Figure 11: Modelled landscape evolution for Case B1. (a) Initial topography and χ index after stage 1, simulating a spatially non-uniform uplift rate. The uplift rate is highest at the eastern boundary of the modelled area (80 mm kyr⁻¹) and decreases with a spatial gradient of -2 mm kyr⁻¹ km⁻¹ towards the west so that the uplift rate is halved (40 mm kyr⁻¹) at the western boundary of the modelled area. (b) Topography and χ index after stage 2, simulating the fault movement. (c) Histograms, mean values, and 1σ standard deviations for geomorphic indices (χ index, channel elevation, mean upstream relief, and mean upstream gradient) at channel heads on the western and eastern flanks of the MDD, extracted from the modelled topography at the end of stage 2.





Cases B1 and B2 involved spatially variable regional uplift during stage 1, followed by spatially variable local uplift by faulting during stage 2. Case B1 also exhibits similar results to those for Case B2 (Figs. 11 and A2). The drainage patterns of these two cases however show noticeable changes during stage 2 when compared with Cases A#. During stage 2, the MDD migrates

westwards by 100–800 and 100–1200 m in Cases B1 and B2, respectively (Figs 11a, 11b, A2a, and A2b). The higher sensitivity of MDD to fault slip in Case B2 may be attributable to its shorter channels compared with Case B1. The western-flank channels in both cases shorten during stage 2, losing their upstream areas (Fig. 12b). In addition, the channels undergo subtle changes near the fault. For instance, some channels flowing from north to south or from south to north become shorter or disappear, whereas some new first- or second-order channels that are oriented transversely or obliquely to the fault develop in the vicinity of the fault

(Fig. 12b). Despite these changes in drainage configuration during stage 2, the contrasting χ indices of the western- and eastern-flank channel heads observed in the initial topography persist through stage 2.

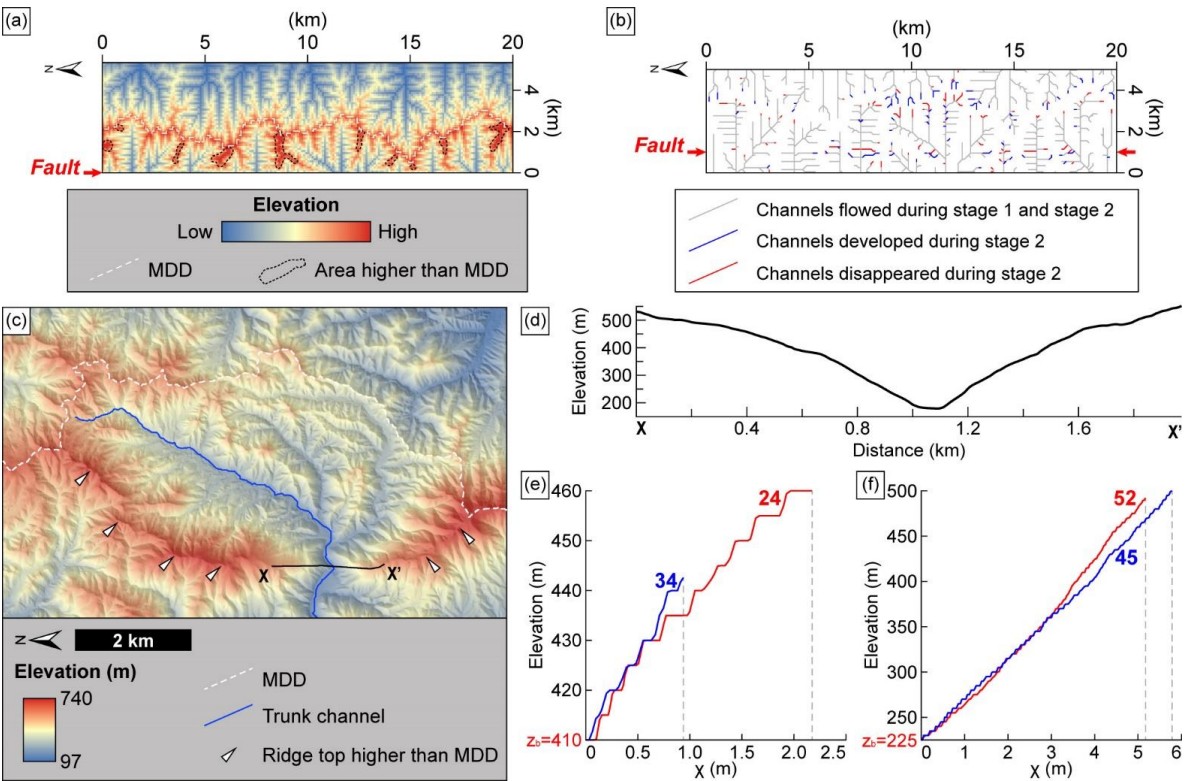

**Figure 12: (a) Resultant topography for Case B1, which simulates the northern part of the study area with asymmetric uplift (westward tilting). Areas with higher elevation than that of the MDD are observed and show similar features to the high ridge top within segment**

**1 (Fig. 12c). (b) Change in the routes of channels during stage 2 (faulting). The western-flank drainage system loses upstream area, whereas the eastern-flank drainage systems gain upstream area. Some channels parallel to the fault disappear, and new channels oriented oblique or transverse to the fault are developed. (c) The ridge top higher than the MDD in segment 1. The location of this figure is marked in Fig. 9a. This ridge has elevated owing to the higher uplift rate on the western flank since reactivation of the UFZ during the late Quaternary. The antecedent stream flowing within the upstream area in the vicinity of the MDD and the ridge has continued to erode**

**this ridge. (d) Cross-profile of X–X' in Fig. 12c. The scale of the horizontal axis and vertical axis of this profile is 1:1. (e and f) χ-transformed profile of the uppermost reach of channels within segments 1 and 2, respectively. These channels are the same as those presented in Figs. 9a and 9c.**

The topography of Case B1 after stage 2 exhibits similar patterns of geomorphic indices to those of Case B2 (Figs. 11c and A2c).

The eastern-flank χ indices, channel head elevations, and mean upstream gradients are significantly lower than their western-flank





counterparts (p-value < 0.05). In contrast, the eastern-flank mean upstream relief values are either comparable to (Case B1) or higher than (Case B2) those on the western flank. As with the results for Cases A#, Cases B# also display patterns of geomorphic indices that differ from expectations. We expect the western flank to have lower χ indices than the eastern flank, in response to higher uplift rate on the western flank during stage 2.

**5 Discussion**

**5.1 Segmentation of the UFZ**

Extensive faults or fault zones typically undergo repeated fault patch rupture along specific portions or segments of their lengths. Such faults are called segmented faults with each segment that ruptures during a single seismic event termed an 'earthquake segment' (McCalpin, 1996). Identification of fault segmentation is crucial for understanding the geometry, mechanism, and
seismological behaviour of the neotectonic faults. However, defining the rupture segment unequivocally can be challenging unless multiple historical ruptures have been preserved and can be observed, such as that observed in segmented normal fault arrays in low-strain arid settings (Moore and Schultz, 1999). Further, McCalpin (1996) distinguished five different types of fault segments: (1) rupture, (2) behavioural, (3) structural, (4) geological, and (5) geometric segments (table 9.7 of McCalpin, 1996). The definitions of these segments are listed in order from the most stringent to least restrictive, and the rupture segment is synonymous
with the previously defined earthquake segment.

In this study, the UFZ was divided into five geological segments defined using geomorphic indicators (McCalpin, 1996) and considering fault geometry. Where segments are arranged across stepovers or bends, there may be a zone of cumulative vertical displacement deficit, which is termed a 'displacement trough', in the intersegment zone (Cartwright et al., 1995; Dawers, 1995; Manighetti et al., 2015). We identified UFZ displacement troughs with a relatively low degree of tectonic intensity on the basis of
geomorphic evidence, such as lows in the swath profile, $k_{sn}$, relief, gradient, χ index, and channel head elevation along the MDD (Fig. 7). We reason that relatively low tectonic intensity correlates with low values in these morphometric parameters, as suggested in earlier pioneering studies (Bull, 1977; Cox, 1994; Keller and Pinter, 1996). Areas with lower swath profile, $k_{sn}$, relief, and gradient along the MDD are interpreted as having a lower degree of tectonic intensity as compared with areas having higher values of these indices.

Accordingly, we divided the UFZ into five segments (segments 1–5; Figs. 7 and 9). The northern boundary of segment 1 is near the northern tip of the NNW–SSE-striking buried fault. There are lows in the swath profile, mean upstream $k_{sn}$, relief, and gradient extracted at the base-level elevation of 200 m (Fig. 7a and orange lines in Figs. 7b–7d). The intersegment zone between segments 1 and 2 is recognised by the low in the swath profile and the relatively low $k_{sn}$ extracted at the base-level elevation of 200 m (Fig. 7a and orange line in Fig. 7b). The strike of an exposed fault along the mountain front also changes abruptly from NNW–SSE to
NW–SE in this intersegment zone. Segment 2 has one of the highest swath profiles and high values of other geomorphic indices, which also coincide with the highest observed CADR for the entire study area found in this segment (Figs. 7 and 8). We divided segment 2 from segment 3 where the next set of lows in the swath profile, $k_{sn}$, relief, and channel head elevation appear (Fig. 7a and orange lines in Figs. 7b, 7c, and 7f). The gradient and χ index also have relatively low values there (orange lines in Figs. 7d and 7e). Further, a NE–SE-striking fault trace terminates between segments 2 and 3, and the strike of the exposed fault changes to
N–S. In addition, we found a valley floor divide in this intersegment 2–3 zone (Fig. 9a). We demarcated the boundary between segments 3 and 4 at a fault step (jog). The swath profile in this intersegment 3–4 zone shows a marked low (Fig. 7a), and relief and gradient are also low (both red and orange lines in Figs. 7c and 7d). The intersegment zone between segments 4 and 5 is characterised by lows in the swath profile, $k_{sn}$, and channel head elevation (Fig. 7a and red lines in Figs. 7b and 7e), and relatively



low values for the other geomorphic indices. The NNW–SSE-striking main fault strand is crosscut by NE–SW-striking minor
faults, and additionally, an exposed fault terminates in this intersegment 4–5 zone.

The longest segment (Segment 3) is about 10.25 km long, and the shortest segment (Segment 2) is about 4.72 km long (Fig. 9a).
Our proposed segment lengths are within the range of surface rupture lengths (SRLs) derived from empirical equations for reverse
faults that describe the relationship between earthquake magnitude and SRL (Bonilla et al., 1984; Wells and Coppersmith, 1994).
Accordingly, assuming a possible maximum earthquake magnitude of $M_W$ 7.0 (Slemmons and Depolo, 1986; Kyung, 2010), the
maximum SRL could be 25.60 km (Bonilla et al., 1984) or 43.58 km (Wells and Coppersmith, 1994). If the SRL is calculated
using the magnitude of the Pohang earthquake ($M_W$ 5.4; Fig. 1a), the predicted SRL would be 0.46 km (Bonilla et al., 1984) or
2.13 km (Wells and Coppersmith, 1994). Given these calculations, our proposed segmentation lengths seem reasonable, and our
proposed segments appear to be physically realistic.

A recent study (Cheon et al., 2023) also divided the incised valley containing the UFZ on the basis of: (1) differences in fault-
hosting rocks, and (2) width of the deformation zone. These authors divided the UFZ into only two northern and southern segments
at the boundary between our segments 3 and 4. We attribute this difference to the different segmentation criteria used and argue
that our geomorphic-based fault segmentation has several advantages, as follows. First, the use of geomorphic indices allows an
indirect assessment to be made of the relative tectonic intensity over wide study areas with spatial continuity. Fault outcrop studies
and trench surveys do not have spatial continuity because they provide only point-specific information. Although determining fault
segmentation based on observed slip rates would be most accurate, it is impractical due to the large number of geological surveys
that would be required to obtain spatially continuous results. Second, geomorphic evidence can be applied even in areas where
surface deformation is only weakly expressed. It is difficult to identify direct surface deformation from faulting in Korea because
of the low slip rates, rapid physical and chemical erosion, and dense vegetation. However, topography can provide meaningful
information on tectonic activity, as the present topography is a cumulative result of all past processes.

### 5.2 Geomorphic evolution of the eastern block of the UFZ in response to tectonic movement

We next used the similarities in the patterns of geomorphic indices for each fault segment and the modelled topographies to
interpret the geomorphic evolution of the eastern block of the UFZ (Figs. 9–11, A1c, and A2c). The χ index represents the longer-
term view for topography owing to its reliance on the integral method from the far downstream to the channel head (Forte and
Whipple, 2018; Zhou et al., 2022). Other geomorphic indices, such as mean upstream gradient and relief, respond sensitively to
tectonic activity near the MDD and reflect shorter-term view for topography. Although we employed realistic settings for all
boundary conditions in the models based on a comprehensive understanding of the tectonic, geological, and geomorphic processes
in the study area, it is acknowledged that there are likely to be discrepancies between the modelled and actual settings of variables
(e.g., coefficient of erosion, uplift rate, and its spatial gradient). Comparing geomorphic indices that are sensitive to minor
variations in boundary conditions could lead to a misinterpretation of the geomorphic evolution. For these reasons, we chose to
focus on a comparison of the pattern of χ indices.

We established the pattern of how geomorphic index values vary between the western and eastern flanks along the MDD (Fig. 8a–
8e), but this variation along the MDD makes quantitative comparisons between geomorphic indices on western- and eastern-flank
difficult. In addition, each of the measured indices has a wide range of values, resulting in large deviations from mean values.
Comparing the geomorphic indices of the western and eastern flanks for the entire study area at once could lead to a Type II error
(false negative), leading to the possible erroneous conclusion that there is no significant difference in values of indices between
the western and eastern flanks on account of the large range of values for each index. Therefore, we compared the western- and
eastern-flank indices segment by segment.





For segments 2–5, all geomorphic indices, except for χ index, generally show a consistent pattern (lower western-flank channel head elevation and higher western-flank mean upstream relief and mean upstream gradient than those of the eastern flank), which

indicate higher erosion rates on the western flank (Fig. 9b). In contrast with all other geomorphic indices, differences between the western-flank and eastern-flank χ index values are inconsistent. The western-flank χ indices in segments 1 and 2 are higher than those of the eastern flank (p-value < 0.05), the same as those of the eastern flank in segment 3 (p-value > 0.05), and lower than those of the eastern flank in segments 4 and 5 (p-value < 0.05). This inconsistent pattern of the χ index throughout the study area is related to its decoupling from the CADR, channel incision rate, and other geomorphic indices in segments 1 and 2. Although the

pattern of the χ index in segment 1 is coupled with the other geomorphic indices, it is decoupled from the higher CADR and incision rate on the western flank (Tables 2 and 3 and Figs. 8 and 9b). In addition, the χ index in segment 2 is decoupled from not only CADR and channel incision rate but also the other geomorphic indices. These decoupled patterns of the χ index in segments 1 and 2 (i.e., lower χ indices on the eastern flank of TMR) contradict what would be generally expected from the higher CADRs, channel incision rates, mean upstream relief, and mean upstream gradient of the western flank compared with the eastern flank.

To facilitate the investigation of the geomorphic evolution of the study area, we grouped the five proposed segments into two distinct parts corresponding to the northern and southern parts of the UFZ. The northern part comprises segments 1 and 2, and the southern part contains segments 3, 4, and 5. Delineation of the boundary between the northern and southern parts was based on the following criteria: (1) the E–W width of the eastern block of the UFZ at the centre of segment 3 is half that of segment 1, (2) the pattern of χ indices shows a noticeable difference at the boundary between the northern and southern parts (i.e., decoupled versus

coupled χ indices), (3) the valley floor divide may be a natural boundary between the two parts, and (4) there is an abrupt change in the orientation of faults across this boundary from NNW–SSE to N–S.

### 5.2.1 Northern part of the UFZ: segments 1 and 2

The geomorphic evolution of the northern part of the UFZ, which is characterised by lower χ indices on its eastern flank, is better explained by Case B1 than Case A1 (Figs. 9b and 11c). χ indices differ markedly between Cases A1 and B1, but the only difference

in boundary conditions between these two models is the spatial uniformity of uplift during stage 1. Case A1 involves spatially uniform uplift, and Case B1 involves spatially variable uplift during stage 1 (Fig. 4). We interpret these lower values of eastern-flank χ indices in segments 1 and 2 compared with western flank to have resulted from the influence of initial topography. The northern part of study area may have been in topographic and geometric disequilibrium (i.e., biased MDD eastwards and asymmetric χ indices) since the area experienced tectonic surface uplift before reverse faulting of UFZ in the Quaternary. This

disequilibrium is caused by the asymmetric uplift during stage 1 and has persisted to the present day, as it is simulated in Case B1 (Fig. 11). Topographic and geometric disequilibrium and its persistence by the asymmetric uplift have also been identified in other modelled cases and natural systems (Willett et al., 2014; Forte and Whipple, 2018; Zhou et al., 2022; Zhou and Tan, 2023). In such cases, disequilibrium caused by the asymmetric uplift persists even after the onset of spatially uniform uplift, until the area reaches equilibrium and steady state through adjustment to the uniform uplift (Willett et al., 2014; Forte and Whipple, 2018). In the present

study, the northern part of the UFZ has likely remained in disequilibrium as a result of the asymmetric uplift pattern prior to Quaternary faulting and is still in transient state to attain equilibrium even after stage 2.

Within the northern part of the study area, segments 1 and 2 also show distinct patterns in geomorphic indices other than the χ index, such as channel head elevation, mean upstream relief, and gradient (Fig. 9). The differences between the two segments can be attributed to two possible factors: (1) the channel length between the fault and the channel head and (2) tectonic activity. Channel

lengths between the fault and the channel heads are longer in segment 1 than in segment 2. In segment 1, buried faults are developed in the incised valley, far to the west from the mountain front (Cheon et al., 2023). The response time of a channel to tectonic events



increases with increasing channel length between the fault and channel head. Therefore, in segment 1, it is plausible that the most recent tectonic signal from Quaternary fault slip has not yet been transferred to the channel head. Secondly, the inferred tectonic intensity, based on geomorphic indices and the CADR (Figs. 7 and 8), is higher in segment 2 than that in segment 1. Geomorphic

indices might be expected to have responded less sensitively to uplift in segment 1 because of its lower tectonic intensity than that of segment 2.

### 5.2.2 Southern part of the UFZ: segments 3, 4, and 5

The pattern of measured $\chi$ indices in the southern part of the UFZ is similar to that of Case A2 (Figs. 9b and 10c). Unlike the northern part of the UFZ, the modelled outcome of Case A2 implies that the southern part of the UFZ had achieved topographic

and geometric equilibrium before stage 2 (Fig. 10a). The lower western-flank $\chi$ indices as compared with those for the eastern flank may indicate the adjustment of channels to a higher uplift and erosion rates on the western flank after Quaternary reverse faulting of the UFZ. This rapid adjustment to the tectonic perturbation resulted from the shorter channel length (between the fault and the channel head) in segments 3–5 compared with the northern part of the UFZ. The western-flank $\chi$ indices in segments 4 and 5 are lower than the corresponding eastern-flank values. However, there is no difference in $\chi$ index values between the western and

eastern flanks in segment 3, which can be attributed to the low base-level elevation ($z_b$) set to calculate the $\chi$ index (50 m a.s.l.). This low base-level elevation resulted in the integration of a much further downstream reach than the fault location in segment 3, which is less relevant to tectonic uplift (Fig. 9a and 9c). The upper reaches of the fault have higher $k_{sn}$ values on the western than the eastern flank, consistent with the pattern of values of the other geomorphic indices, CADR, and channel incision rate.

We next used $\chi$ indices from the modelled topography and those observed in the study area to establish the geomorphic evolution

of the eastern block of the UFZ. The northern part of the block underwent spatially variable uplift prior to Quaternary reverse faulting on the UFZ, resulting in the observed asymmetric topography. Conversely, the southern part of the block underwent spatially uniform uplift, attaining topographic and geometric equilibrium prior to Quaternary reverse faulting. Our interpretation on the geomorphic evolution of the eastern block of the UFZ is based on a generalised concept of the influence of pre-existing topography on subsequent geomorphic processes. The pre-existing topography, henceforth referred to as 'inherited topography',

denotes the topography that existed prior to the event of interest. This inherited topography may include, for example, the channel length governed by the shape of the block and the degree of asymmetry of the topography controlled by the orientations of geological features (e.g., faults) and previous tectonic movement. The four modelling cases in this study simulated different inherited topographies but the same tectonic movement. Our model results demonstrate that the geomorphic response to subsequent tectonic movement is influenced by the inherited topography and that geomorphic indices (such as the $\chi$ index) can be used to

measure this influence.

### 5.3 Migration of the MDD and landscape evolution

Previous assessments of divide mobility have relied on comparing geomorphic indices, such as the $\chi$ index and Gilbert metrics, of opposing sides of a MDD (Gilbert, 1877; Willett et al., 2014; Forte and Whipple, 2018; Kim et al., 2020; Zeng and Tan, 2023). Understanding divide mobility is crucial to the investigation of landscape evolution, as it is a dynamic indicator of how the

landscape evolves and may help determine the driving forces (i.e., tectonic movement and spatial patterns of uplift and erosion) of this evolution. However, we discovered that the $\chi$ anomaly may not accurately reflect divide mobility in the UFZ study area because of the assumptions of $\chi$ index. $\chi$ index can correctly indicate the divide mobility only when the area of interest has spatially uniform uplift, climate, lithology, and erodibility (Perron and Royden, 2013; Willett et al., 2014; Forte and Whipple, 2018). The $\chi$-transformed profiles of channels with low base-level elevation ($z_b$ = 50 m a.s.l.) in segments 2 and 3 show that the uppermost





reaches of western-flank channels exhibit higher $k_{sn}$ and χ index values than the eastern flank (Fig. 9c), giving contradictory interpretations on divide mobility. The higher $k_{sn}$ on the western flank is related to the eastwards divide migration, whereas the higher χ index on the western flank indicates westward divide migration. In this case, instantaneous divide mobility can be accurately evaluated only by comparing the uppermost reaches of channels from opposing sides of the MDD (Zhou et al., 2022). We adopted this approach for the northern part of the UFZ (segments 1 and 2), as the χ indices are decoupled from the CADR and

bedrock incision rate there, as well being decoupled from additional geomorphic indices used in this study (Figs. 9b, 12e, and 12f). Western-flank uppermost reaches in segment 1 have lower $k_{sn}$ and higher χ index values when compared to those from the corresponding eastern flank (Figs. 12c and 12e). This suggests that the MDD is migrating westwards in this segment, and that it is approaching topographic and geometric equilibrium (see section 5.2.1). Interestingly, we observed that the ridge top of an internal sub-basin on the western flank of the MDD in segment 1 is up to 380 m higher than the MDD itself (Fig. 12c). We propose two

possible explanations for this high ridge top: (1) discrete stream capture of western-flank channels owing to the high uplift rate on the western flank; and (2) erosion of the ridge top by westward-flowing antecedent streams. The strongest evidence supporting stream capture is the presence of an elbow of capture, wind gap, gorge-like valley, and lower χ index at the channel head of a captor stream (Bishop, 1995; Willett et al., 2014). However, we could not identify any convincing evidence for stream capture, such as an elbow of capture or a gorge-like valley near the ridge top or wind gap near the MDD in segment 1, either in the field or

on the DEM. The lower $k_{sn}$ and higher χ index values of the western-flank uppermost reach in this segment also do not imply an aerial gain of the western-flank drainage system owing to stream capture (Fig. 12e). Furthermore, modelling results for Case B1 show similar features of high ridge tops without discrete stream capture. The resultant topography of Case B1, which simulates the landscape evolution of the northern part of the UFZ and spatially variable (asymmetric) uplift prior to late Quaternary faulting, shows some areas with higher elevations than that of the MDD (Fig. 12a). Therefore, we interpret that the streams flowing within

the drainage in the vicinity of the MDD and the elevated ridge on the western flank of segment 1 are the results of antecedent streams. The high elevation of the ridge on the western flank of the MDD is ascribed to the higher uplift rate on the western flank since Quaternary reverse faulting of the UFZ. The channels are hypothesised to have been subject to a sufficiently high erosion rate to retain the original stream route of the inherited topography in response to the higher uplift rate on the western flank, but this erosion rate was accordingly insufficiently high to capture parts of the drainage system on the eastern flank.

The western-flank uppermost reach in segment 2 has higher $k_{sn}$ and lower χ index values than those on the eastern flank, as seen in the χ-transformed profile (Fig. 12f), whereas the profile with lower base-level displays higher χ index on the western flank (Fig. 9c). χ indices at the channel heads are sensitive to base-level variations (Forte and Whipple, 2018). In the present case, the lower χ and higher $k_{sn}$ values of the western-flank uppermost reach indicate short-term migration of the MDD towards the east. This inferred divide mobility is consistent with the results of other geomorphic indices providing short-terms view of topographic

evolution, such as elevation, mean upstream relief, and mean upstream gradient at the channel head. Consequently, we interpret that the MDD within segment 2 is migrating eastwards, as is the MDD within segments 3–5, in which patterns of all geomorphic indices are consistent with eastward migration of the MDD.

## 6 Conclusions

The Ulsan Fault Zone (UFZ) has been one of the most active fault zones on the Korean Peninsula since its reactivation ~ 5 Ma.

Our study area, the eastern, mountainous, hanging wall block of the UFZ, has undergone regional uplift under an ENE–WSW-oriented neotectonic maximum horizontal stress after 5 Ma. This study aimed to determine the degree of and variation in tectonic





intensity along the UFZ, characterise the past and present geomorphic processes operating along the UFZ, and infer landscape evolution patterns in response to tectonic perturbation involving reactivation of the UFZ.

We evaluated the relative tectonic activity along the fault zone using geomorphic indices, and catchment-averaged denudation

rates (CADR) and bedrock incision rate derived using in situ cosmogenic $^{10}$Be. We divided the eastern UFZ block into five geological segments based on the relative tectonic intensity that we assessed. This study represents the first investigation involving segmentation based on the relative degree of tectonic intensity of the UFZ inferred from tectonic geomorphic data. Our new segmentation scheme may provide a basic grounding for further investigations the kinematics and seismic hazard of the UFZ.

We also interpreted the tectono-geomorphic evolution of the study area by modelling landscape evolution and comparing the values

and patterns of geomorphic indices of the modelled topography with those observed in the study area. We interpret that the northern UFZ (a combination of our segments 1 and 2) underwent regional asymmetric uplift (westward tilting) prior to Quaternary reverse faulting since ~ 2 Ma. In the northern UFZ, the χ index is decoupled from the CADR, bedrock incision rate, and other geomorphic indices. We interpret that this decoupled χ index in the northern UFZ indicates that this area is in a transitional state of topographic and geometric disequilibrium caused by westward tilting prior to late Quaternary faulting. The southern UFZ (a combination of

our segments 3–5) was negligibly affected by asymmetric uplift before Quaternary reverse faulting, as channel lengths (distance between the Ulsan Fault and the channel head) were sufficiently short to adjust quickly to the uplift. Our analysis and interpretation of the tectono-geomorphic evolution of the UFZ shows that inherited topography can influence the subsequent geomorphic processes and topographic response to neotectonic reverse fault slip. The geomorphic indices we utilized can therefore be regarded to characterize not only the present topography, but also to hold information resulting from the accumulation of a history of

tectonics and erosion.

Lastly, we examined the short-term mobility (migration) of the main drainage divide (MDD) in the study area as a response to variable surface uplift along the UFZ's hanging wall. The MDD within segments 2–5 has migrated eastwards owing to the higher erosion rate of the western-flank drainage network compared with that on the eastern flank due to its closer proximity to the UFZ. The MDD within segment 1 is migrating towards the west, presumably because of the persistent influence of the disequilibrium

caused by inherited topography. The presence of antecedent streams cutting a ridge top that is higher than the elevation of the MDD on the western flank in segment 1 implies a higher erosion rate on the western flank driven by the higher uplift rate relative to the eastern flank. This inference is consistent with our observed CADR values and bedrock incision rates derived from cosmogenic $^{10}$Be measurements.

Our study clearly demonstrates that tectonic geomorphic data can be used to infer differential tectonic intensity (i.e., variable fault

slip and surface uplift) and that modelling can be used to infer possible influences of inherited topography in intraplate regions with extremely low strain rates and fault slip rates and extremely high erosion rates.





**Appendices**

Appendix A. Results of landscape evolution modelling.

Appendix A contains landscape evolution modelling results for the two cases (Cases A1 and B2) that show dissimilar patterns of values of geomorphic indices with northern and southern parts of the UFZ.

**Figure A1: Modelled landscape evolution for Case A1. (a) Initial topography and χ index after stage 1, simulating a spatially uniform uplift of 80 mm kyr⁻¹ over the modelled area. (b) Topography and χ index after stage 2, simulating fault movement. (c) Histograms, mean**
**values, and 1σ standard deviations for geomorphic indices (χ index, channel elevation, mean upstream relief, and mean upstream gradient) at the channel heads on the western and eastern flanks of the MDD, extracted from the modelled topography at the end of stage 2.**

**Figure A2: Modelled landscape evolution for Case B2. (a) Initial topography and χ index after stage 1, simulating a spatially non-uniform uplift. The uplift rate is highest at the eastern boundary of the modelled area (80 mm kyr⁻¹) and decreases with a spatial gradient of -2 mm kyr⁻¹ km⁻¹ towards the west. (b) Topography and χ index after stage 2, simulating the fault movement. (c) Histograms, mean values, and 1σ standard deviations for geomorphic indices (χ index, channel elevation, mean upstream relief, and mean upstream gradient) at channel heads on the western and eastern flanks of the MDD, extracted from the modelled topography at the end of stage 2.**

**Data availability.** All $^{10}$Be data are available in Tables 2 and 3. Questions or request for DEM and shapefiles for morphometric analysis can be sent to the corresponding author.



**Author contributions.** CHL and YBS conceptualized the study and conducted the field investigations with DEK and SMH. YBS was responsible for funding acquisition. CHL designed the $^{10}$Be lab experiments with BYY. CHL and DEK performed all formal

analysis and simulation. CHL, YBS, and JW prepared the manuscript with contributions from all co-authors.

**Competing interests.** The authors declare that they have no conflicts of interest.

**Acknowledgements.** We are grateful to Purevmaa Khandsuren for managing chemical treatment of $^{10}$Be samples.


**Financial support.**

This research was supported by a grant (2022-MOIS62-001(RS-2022-ND640011)) of National Disaster Risk Analysis and Management Technology in Earthquake funded by Ministry of Interior and Safety (MOIS, Korea).



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
