# Peer review of "Geomorphic indices for unveiling fault segmentation and tectonogeomorphic evolution with insights into the impact of inherited topography, Ulsan Fault Zone, Korea"

_EGUsphere, 2024_

## Referee Comment (RC1)

Review of "Geomorphic indices for unveiling fault segmentation and tectono-geomorphic evolution with insights into the impact of inherited topography, Ulsan Fault Zone, Korea"

This is a very interesting paper which is overall well written and clear. Such study coupling geomorphic analyses with neotectonic activities is very enlightening to constrain the geomorphic evolution along a regional major fault belt. In general, the dataset presented is credible and would attract the attention of researchers who work on tectonic and geomorphic evolution of southeastern margin of Korea, as limited data are available from the study area. However, as I will discuss in the following, some middle-minor revisions would be necessary before this manuscript becomes a suitable contribution to the journal.

Major comments:

(1) For geomorphic modelling of cases B1 and B2, the uplift rates of eastern end were set at 18 mm/kyr and 42 mm/kyr, respectively. The uplift rate of 18 mm/kyr for the northern part of the block was calculated based on a relationship between the incision rate and the distance. The authors should give more explanation for its validity, because such an uplift rate is smaller than the CARD value. Similarly, the uplift rate of 42 mm/kyr was obtained based on a relationship between the average CARD and the distance in the southern part.

(2) The UFZ has been divided into five segments based on geomorphology analyses alone. I understand how difficult it will be to obtain some data in an urbanized area. However, it will be more convincing if the authors can provide some other data, for example, the GPS slipping rates, stress accumulation, InSAR deformations…

(3) Base on the modelling results, segment 1 was considered to migrate westward, while segments 2-5 has migrated eastward. However, such a discrepancy was not explained in detail.

Minor comments:

(1) The geomorphic indices should be italic.

(2) For Figures 1a and 1b, I suggest to add the movement properties of the major faults

(strike, normal, or thrust) if possible. Can the active faults and ancient faults be marked by different colors (Red and Black) in Figure 1b? I suggest to add the names beside the major fault, e.g. Ulsan Fault. I also have a question. There are three moderate earthquakes shown in the Figure 1a, but why most of them do not occur along the major fault belts?

(3) I suggest to add the methodology description of the students t-test.

(4) The channel incision rate was calculated based on cosmogenic nuclide. Thus, I suggest to add the outcropping and sampling description. What is the kind of the rock? What is the thickness of the sample?

(5) I suggest to add the Ulsan Fault in Figure 3a.

(6) Channels 5b and 5c should be clearly shown on 5a.

(7) Figure 9 was started to cite in chapter Discussions, behind the Figure 10.

(8) Figure 12c should be clearly shown on Figure 2a.

(9) The chapter Conclusions is too much lengthy. In fact, some of the content are not the conclusions.

---

## Author Comment (AC1)

Reply for the comment on egusphere-2024-198 (Referee #1)

**Title: Geomorphic indices for unveiling fault segmentation and tectono-geomorphic evolution with insights into the impact of inherited topography, Ulsan Fault Zone, Korea**

| Major comments | |
| --- | --- |
| **Comment** | **Reply** |
| For geomorphic modelling of cases B1 and B2, the uplift rates of eastern end were set 18 mm/kyr and 42 mm/kyr, respectively. The uplift rate of 18 mm/kyr for the northern part of the block was calculated based on a relationship between the incision rate and the distance. The authors should give more explanation for its validity, because such an uplift rate is smaller than the CADR value. Similarly, the uplift rate of 42 mm/kyr was obtained based on a relationship between the average CADR and the distance in the southern part. | For both cases B1 and B2, the uplift rates of the eastern end during the second stage are zero (Fig. 4). In Case B1, the uplift rate 2.5 km east of MDD is 18 mm kyr$^{-1}$ with a gradient towards the east of -22.27 mm kyr$^{-1}$ km$^{-1}$, based on the relationship between the incision rate and the distance. We then extrapolated the uplift rate towards the east using this gradient until the uplift rate reaches zero.
In Case B2, the uplift rate at the fault location (western flank) is set to 42 mm kyr$^{-1}$, based on the ratio of CADRs in the northern and southern parts. We set the uplift rate to decrease to zero at the 2 km east of MDD because most knickpoints on the eastern-flank channels in the southern part are located within this distance.
On the western flank, the modelled uplift rates (118 mm kyr$^{-1}$ in the northern part and 42 mm kyr$^{-1}$ in the southern part) are comparable to the CADRs. However, on the eastern flank, the modelled uplift rates are significantly lower than the CADRs. There are several possible reasons for this discrepancy:
(1) The uplift rate gradient could be overestimated.
(2) The CADRs reflects not only the faulting along the UFZ but also the other kinds of tectonic movement.
Both of those reasons are plausible, but we could not quantify the extent to which the uplift rate gradient is overestimated or the degree to which other types of tectonic movement contribute to the uplift rates in the study area. Consequently, we assumed a linear decrease in uplift rate from the fault location and calculated the uplift rates as |

| | described above . |
|---|---|
| The UFZ has been divided into five segments based on geomorphology analyses alone. I understand how difficult it will be to obtain some data in an urbanized area. However, it will be more convincing if the authors can provide some other data, for example, the GPS slipping rates, stress accumulation, InSAR deformations. | Thank you for your valuable suggestion regarding the integration of additional data types such as GPS velocity fields and InSAR measurements to delineate fault segments. Indeed, these geodetic methods are critical for identifying 'rupture segments' that delineate the historical rupture limits for seismic events and are particularly useful in tectonically active regions.

However, our study area in the southeastern part of Korea is characterized by its tectonic quiescence, being situated within an intraplate region. This low level of tectonic activity is a primary reason why neither this study nor other recent research in the area (e.g., Cheon et al., 2023) have employed these geodetic data for segment division.

In this context, our segmentation of the Ulsan Fault Zone (UFZ) was aimed at identifying 'geological segments' based on geomorphic evidence, which is more feasible and justifiable given the regional tectonic setting. We believe that this approach remains valid and appropriate for the geological characteristics and data availability pertaining to the UFZ. |
| Base on the modelling, results, segment 1 was considered to migrate westward, while segments 2–5 has migrated eastward. However, such a discrepancy was not explained in detail. | Thank you for your comment. We address the exceptional westward migration of MDD within segment 1 in the section '5.2.1 Northern part of the UFZ: segments 1 and 2.' The first paragraph of this section explains that both segments 1 and 2, which they have been in topographic and geometric disequilibrium, and the MDD in segment 1 is migrating westwards, approaching equilibrium in section 5.3. In the second paragraph, we explain that the distinct patterns of geomorphic indices are attributed to (1) the channel length between the fault and the channel head and (2) difference in tectonic activity. We intended that these consequently influence the direction of MDD migration. You can find this:

**[Lines 718–726]** "The differences between the two segments can be attributed to two possible factors: (1) the channel length between the fault and the channel head and (2) tectonic activity. |

| | Channel lengths between the fault and the channel heads are longer in segment 1 than in segment 2. In segment 1, buried faults are developed in the incised valley, far to the west from the mountain front (Cheon et al., 2023). The response time of a channel to tectonic events increases with increasing channel length between the fault and channel head. Therefore, in segment 1, it is plausible that the most recent tectonic signal from Quaternary fault slip has not yet been transferred to the channel head. Secondly, the inferred tectonic activity, based on topographic metrics and the CADR (Figs. 7 and 8), is higher in segment 2 than in segment 1. Topographic metrics might be expected to have responded less sensitively to uplift in segment 1 because of its lower tectonic activity than that of segment 2." |
|---|---|

**Minor comments**

| Comment | Reply |
|---|---|
| The geomorphic indices should be italic. | We will change them to italics throughout the manuscript |
| For Figures 1a and 1b, I suggest to add the movement properties of the major faults (strike, normal, or thrust) if possible. Can the active faults and ancient faults be marked by different colors (Red and Black) in Figure 1b? I suggest to add the names beside the major fault, e.g., Ulsan Fault. I also have a question. There are three moderate earthquakes shown in the Figure 1a, but why most of them do not occur along the major fault belts? | The major faults in the Figure 1a were developed during the Mesozoic. However, there is not enough evidence supporting that most of them, except for the Yangsan and Ulsan Fault Zone, have been reactivated under the present stress regime. That is why we did not mark the movement properties of those fault zones. In the same context, identifying active and ancient faults within the Ulsan Fault Zone also remains controversial based on the research cases until now. We may be able to add the movement property of the Yangsan Fault Zone in Figure 1a and the name of major fault in Figure 1b.
The $M_W$ 5.5 earthquake (12 Sep. 2016) occurred near the Yangsan Fault zone, which is one of the biggest fault zones in Korea. The focal mechanism of this earthquake is also consistent with the main slip component (right-lateral strike slip) of this fault zone. The $M_W$ 5.4 earthquake (15 Nov. 2017) is known as an '(anthropogenically) induced earthquake', which is caused by the fluid injection for the geothermal resource development (Grigoli et al., 2018; Kim et al., 2018). This may be the reason why this event did not occur along the major fault zones. The $M_L$ 4.0 earthquake (30 Nov. 2023) |

| | is considered to have occurred due to the reactivation of ENE–WSW-striking strike-slip fault, which is related to the formation of a Tertiary basin in the southeastern Korea. |
|---|---|
| |  |
| | Mw 5.5 (12 Sep. 2016)   Mw 5.4 (15 Nov. 2017)   ML 4.0 (30 Nov. 2023) |
| | **Focal mechanisms of three earthquakes around the study area (Korea Meteorological Administration, 2017; Kim et al., 2018; Korea Meteorological Administration, 2018, 2023).** |
| I suggest to add the methodology description of the students t-test. | We think that it is not appropriate to make a separate section in the 'Methods' part solely for Student's t-test as it is a widely applied statistical method. However, we admit the explanation for Student's t-test is quite simple in the manuscript. We will add several sentences about Student's t-test at the end of sections 3.1.3. |
| | **\* [Lines 227–228]** "*We then used Student's t-test which is a statistical method to determine whether two groups are statistically significantly different from each other. We applied this Student's t-test (two-tailed, $p < 0.05$) to statistically compare the values of the topographic metrics between the western and eastern flanks of the TMR*". |
| The channel incision rate was calculated based on cosmogenic nuclide. Thus, I suggest to add the outcropping and sampling description. What is the kind of the rock? What is the thickness of the sample? | The fluvial terraces where we collected samples are strath terrace (bedrock terrace). , We already included the pictures of those terraces in the Figures 3c and 3d and marked the sample locations on them. |

[Figure]

[Figure]

**Figures 3c and 3d**

The thickness of the sample is already listed in the Table 3.

| Sample name | Latitude (° N, dd) | Longitude (° E, dd) | Elevation (m) | Thickness (cm) |
|---|---|---|---|---|
| WT0-1 | 35.6985 | 129.3514 | 232 | 5.0 |
| WT1-1 | 35.6985 | 129.3514 | 236 | 3.5 |
| WT1-2 | 35.6985 | 129.3514 | 236 | 2.5 |
| ET0-1 | 35.7069 | 129.3921 | 207 | 4.0 |
| ET1-1 | 35.7069 | 129.3922 | 209 | 5.0 |

**A part of Table 3 (the right side of this table is cut because of a lack of space).**

Both strath terraces consist of granite, and we will add this in the section 3.2.2.
**\* [Line 306]** *The sampled strath terraces are located in the drainage basin from which the W4 and E4 CADR samples were taken.* *All terraces consist of granite bedrock.*

| I suggest to add the Ulsan Fault in Figure 3a. | Thank you for your suggestion. Figure 3a is designed to illustrate the locations of the catchments where we collected the samples and to present the CADR results. We considered marking the UFZ on the map, but the boxes displaying the CADR results obscure the fault lines, which led us to omit them. We will revisit the layout to see if |
|---|---|

| | the fault lines can be included without cluttering the visual presentation of the data. |
|---|---|
| Channels 5b and 5c should be clearly shown on 5a. | We agree with your observation regarding the difficulty in discerning the channels in Figures 5b and 5c. However, Figure 5a does not sufficiently clarify this detail. We will mark the channels in Figure 3b. We have also added a sentence to the caption of Fig. 3 to address this change.

**Figure 3b**

**\* [Line 436]** *The locations of these channels are marked in Fig. 3b.* |
| Figure 9 was started to cite in chapter Discussions, behind the Figure 10. | We will adjust the placement and sequence of Figure 9 (between Figure 11 and Figure 12) in the revised manuscript. Consequently, the current Figure 10 will be renumbered as Figure 9, and the current Figure 9 will become Figure 10. |
| Figure 12c should be clearly shown on Figure 2a. | We have already marked the location and area of Figure 12c in Figure 9a, but we acknowledge that it is not clearly visible. We will change it to a bright colour to enhance visibility. |

[Figure]

**Figure 9**

| The chapter Conclusions is too much lengthy. In fact, some of the content are not the conclusions. | We agree that the 'Conclusion' section is overly lengthy and contains content that is not directly related to the conclusions. We will streamline this section by removing the fourth paragraph and reducing the detail in the second and third paragraphs, eliminating a total of 363 words.

* **[Lines 794–826]** *The Ulsan Fault Zone (UFZ) has been one of the most active fault zones* |

<table>
<tr><td></td><td>

*on the Korean Peninsula since its reactivation ~ 5 Ma. Our study area, the eastern, mountainous, hanging wall block of the UFZ, has undergone regional uplift under an ENE–WSW-oriented neotectonic maximum horizontal stress after 5 Ma. This study aimed to evaluate the relative tectonic activity along the UFZ, characterise the past and present geomorphic processes operating along the UFZ, and infer landscape evolution patterns in response to tectonic perturbation involving reactivation of the UFZ.*

*We evaluated the relative tectonic activity along the fault zone using topographic metrics, and catchment-averaged denudation rates (CADRs) and bedrock incision rate derived using in situ cosmogenic $^{10}$Be. We divided the eastern UFZ block into five geological segments based on the relative tectonic activity we assessed. This study represents the first segmentation based on the relative tectonic activity of the UFZ inferred from topographic metrics.*

*We also interpreted the tectono-geomorphic evolution of the study area by modelling landscape evolution and comparing the values and patterns of topographic metrics of the modelled topography with those observed in the study area. We interpret that the northern UFZ (segments 1 and 2) underwent regional asymmetric uplift (westward tilting) prior to Quaternary reverse faulting since ~ 2 Ma. The southern UFZ (segments 3–5) was negligibly affected by asymmetric uplift before Quaternary reverse faulting, as channel lengths (distance between the Ulsan Fault and the channel head) were sufficiently short to adjust quickly to the uplift. Our analysis and interpretation of the tectono-geomorphic evolution of the UFZ show that inherited topography can influence the subsequent geomorphic processes and topographic response to neotectonic reverse fault slip. The topographic metrics we utilized can therefore be regarded as characterizing not only the present topography, but also as holding information resulting from the accumulation of a history of tectonic and erosion.*

*Our study clearly demonstrates that topographic metrics can be used to infer differential tectonic activity (i.e., variable fault slip and surface uplift) and that modelling can be used to infer possible influences of inherited topography in intraplate regions with extremely low strain rates and fault slip rates, and extremely high erosion rates.*

</td></tr>
</table>

**References**

Cheon, Y., Shin, Y. H., Park, S., Choi, J. H., Kim, D. E., Ko, K., Ryoo, C. R., Kim, Y. S., and Son, M.: Structural architecture and late Cenozoic tectonic evolution of the Ulsan Fault Zone, SE Korea: New insights from integration of geological and geophysical data, Front. Earth Sci., 11, 1–14, https://doi.org/10.3389/feart.2023.1183329, 2023.

Grigoli, F., Cesca, S., Rinaldi, A. P., Manconi, A., Clinton, J. F., Westaway, R., Cauzzi, C., Dahm, T., and Wiemer, S.: The November 2017 Mw 5.5 Pohang earthquake: A possible case of induced seismicity in South Korea, Science (80-. )., 360, 1003–1006, 2018.

Kim, K. H., Ree, J. H., Kim, Y. H., Kim, S., Kang, S. Y., and Seo, W.: Assessing whether the 2017 Mw 5.4 Pohang earthquake in South Korea was an induced event, Science (80-. )., 360, 1007–1009, https://doi.org/10.1126/science.aat6081, 2018.

Korea Meteorological Administration: 9.12 Earthquake Response Report, 140 pp., 2017.

Korea Meteorological Administration: Pohang Eartquake Analysis Report, 41 pp., 2018.

Korea Meteorological Administration: Gyeongju Earthquake Analysis Report, 11 pp., 2023.

---

## Author Comment (AC2)

Reply for the comment on egusphere-2024-198 (Referee #2)

**Title: Geomorphic indices for unveiling fault segmentation and tectono-geomorphic evolution with insights into the impact of inherited topography, Ulsan Fault Zone, Korea**

| Major comments | |
|---|---|
| **Comment** | **Reply** |
| The landscape evolution model's setup and parameterisation, particularly regarding stream-power erosion, feel somewhat like an arbitrary choice of conditions. The manuscript would benefit from additional sensitivity analyses, such as exploring a range of values for stream-power parameters, to document how different parameterisations might affect the results. | We appreciate the suggestion for comprehensive sensitivity analyses on stream-power parameters such as erosion coefficient (K), and exponents (m and n) to assess their impact on our landscape evolution models (LEMs). We have been conducting sensitivity analyses for these parameters since we started to write our response to the comments; however, the process is time-consuming and-intensive and ongoing, hence not included in this submission, but to be included in the final version. Regarding the site-specific values, such as the distance between the fault and the main drainage divide (MDD) and uplift rates, we chose not to perform sensitivity analyses. The primary objective of this study is to demonstrate the applicability of our hypothesis to the specific environmental conditions of the Ulsan Fault Zone (UFZ), rather than to generalize or determine precise conditions across varied settings. Consequently, we focused on how inherited topography influences present landscapes under given conditions, rather than exploring a broad range of hypothetical scenarios. Therefore, while we recognize the potential influence of these factors on our results, we did not vary these parameters in our analyses. |
| The methodological framework for the quantitative topographic analysis is unclear and needs a comprehensive overhaul for clarity and reproducibility. This includes adding a more detailed description of what was done (e.g., | We appreciate your feedback on the clarity and reproducibility of our methodological framework. In response, we will revise the manuscript to include a more detailed description of our methods, particularly focusing on the extraction of knickpoints and the parameters used in our analysis. These additions aim to enhance the clarity and ensure that our methodology can be reproduced effectively by others. |

| knickpoint extraction), thus ensuring reproducibility, as well as modifying some of what was done. For instance, there is no need to quantify four topographic metrics with the same function (i.e., to document cross-divide steepness asymmetry) under the umbrella of 'Gilbert Metrics'. | Regarding your comment on the use of multiple topographic metrics, we have re-evaluated their necessity and effectiveness in documenting cross-divide steepness asymmetry. After careful consideration, we have decided to exclude mean upstream gradient and channel head elevation from our analysis. We found that mean upstream relief and the χ index provide a more direct and effective measure of the phenomena we are studying. This change simplifies our methodology while maintaining the integrity and focus of our research. These revisions are intended to address your concerns and improve the manuscript's overall clarity and reproducibility. |
|---|---|
| The manuscript's text, especially within the Methods, Results, and Discussion sections, requires extensive revision for clarity and detail. For example, the topographic analysis results should be quantitatively detailed to better support the study's conclusions. | Thank you for your constructive feedback. We acknowledge the need for clearer and more detailed exposition in the Methods, Results, and Discussion sections of our manuscript. In response, we will undertake a comprehensive revision of these sections to enhance clarity and provide the necessary quantitative details to robustly support our conclusions. Please refer to our specific responses to your detailed comments below, where we outline the changes made in each section. These revisions aim to address your concerns and significantly improve the manuscript's readability and academic rigor. |
| The paper should address how lithological variations might influence the cosmogenic nuclide results. This addition is crucial for interpreting the data accurately. | Thank you for your insightful comment regarding the influence of lithological variations on cosmogenic nuclide results. We recognize the importance of considering lithological differences in interpreting cosmogenic nuclide concentrations due to their potential effect on erosion rates and chemical composition.

In our study, we initially focused on catchment average denudation rates (CADRs) using cosmogenic nuclides as a proxy for erosion. We acknowledge that different rock types can exhibit varying resistance to erosion, which could influence cosmogenic nuclide concentrations. While our sampling strategy aimed to minimize lithological heterogeneity by selecting areas of uniform rock type, primarily granite, we did not |

| | explicitly discuss this in the manuscript. To address this oversight, we will enhance the manuscript by including a section detailing the lithological composition of the sampling sites. This section will discuss the predominant rock types within the catchments and consider how their erosion resistance might affect the cosmogenic nuclide concentrations. Furthermore, we will discuss the implications of lithological variability on our results and ensure that our interpretations consider these potential variations. |
|---|---|
| The calculation of the integral metric χ should be revised to accurately reflect non-uniform spatial variations in background rock uplift, a critical factor for interpreting spatial patterns in χ in this context. | We acknowledge the importance of accurately calculating the χ index to reflect non-uniform spatial variations in background rock uplift, which is critical for interpreting spatial patterns of χ in our study context. We will recalculate the χ index following Equation (5) from Willett et al. (2014), which is specifically designed for non-uniform spatial conditions, including variations in uplift rate and erodibility. We will update the results accordingly in the revised manuscript. |

**Specific comments**

**1 Introduction**

| Comment | Reply |
|---|---|
| Lines 34-35: I suggest using 'topography' or 'topographic data' instead of 'geomorphic characteristics' throughout the manuscript. | We agree that 'topography' is better than the other expressions.
* [Lines 34–35] *Research in the field of tectonic geomorphology involves identifying the signal of neotectonic activity from* topography. |
| Line 35: I suggest using 'topographic metrics' instead of 'geomorphic indices'. The former depicts the output of such methods much better, and changing it consistently throughout the text would be beneficial. | Thanks. We will change 'geomorphic indices' to 'topographic metrics' throughout this manuscript.
* [Lines 35–37] *The classic approach to studies of tectonic geomorphology has been to use* topographic metrics *and was developed in the 1900s (e.g., hypsometric integral, stream length–gradient index, and mountain-front sinuosity; Strahler, 1952; Hack, 1973; Bull, 1977; Cox, 1994; Keller and Pinter, 1996; Bull and McFadden, 2020).*
* [Lines 54–56] *However, these studies do not generally consider the effects of inherited topography (i.e., topography prior to the neotectonic events of interest) on subsequent* |

| | |
|---|---|
| | *geomorphic processes, present topographic dynamics, and topographic metrics.*

We will also revise the other parts of this manuscript in the same way. |
| Line 37: You should include the reference of Wobus et al. (2006) when citing ksn. | Thanks. We will add that reference.
**\* [Lines 37–40]** *The normalised channel steepness index ($k_{sn}$; Flint, 1974; Wobus et al., 2006) and knickpoint analyses are also frequently applied to explore the transient states of channels caused by tectonic activity (Whipple and Tucker, 1999; Duvall et al., 2004; Kirby and Whipple, 2012; Scherler et al., 2014; Marliyani et al., 2016), as the incision of a channel system is the most obvious response to tectonic uplift.* |
| Lines 35-43: The sentences describing progress in extracting quantitative information from topographic data focusing on tectonic geomorphology starting at "The classic approach…" and following to "between tectonic forcing and river incision (…)" can use some rephrasing to depict better decades of theoretical, numerical and empirical advances in quantitative topographic analysis. Here are some suggestions of papers that have compellingly done a similar task (for inspiration): Wobus et al., 2006; Kirby and Whipple, 2012; Whittaker, 2012; Lague, 2014; Demoulin et al., 2017; Mudd et al., 2018. | *Thank you. We acknowledge the current text is rather confusing. We will revise the whole paragraph to enhance clarity and coherence.*

*\* [Lines 35–43] The classic approach to studies of tectonic geomorphology has traditionally relied on topographic metrics, with origins dating back to the 1900s (e.g., hypsometric integral, stream length–gradient index, and mountain-front sinuosity; Strahler, 1952; Hack, 1973; Bull, 1977; Cox, 1994; Keller and Pinter, 1996; Bull and McFadden, 2020). The normalised channel steepness index ($K_{sn}$; Flint, 1974; Wobus et al., 2006) and knickpoint analyses are also frequently applied to explore the transient states of channels caused by tectonic activity (Whipple and Tucker, 1999; Duvall et al., 2004; Kirby and Whipple, 2012; Scherler et al., 2014; Marliyani et al., 2016), as channel incision is a direct response to tectonic uplift. The chi ($\chi$) index was introduced to address limitations associated with slope–area analysis for calculating $K_{sn}$, which can be influenced by (1) noise and errors in topographic data, and (2) the resolution of data itself (Perron and Royden, 2013; Royden and Perron, 2013). Notably, the $\chi$ index facilitates straightforward comparison of $K_{sn}$ values across different channel reaches as the slope of the $\chi$–elevation profile directly reflects the $K_{sn}$ value (Perron and Royden, 2013). It is applied to determine whether a landscape under specific conditions is in a steady state or transient state, and to assess long-term drainage divide mobility (Willett et al., 2014; Forte and Whipple, 2018; Kim et al., 2020; Hu et al., 2021; Lee et al., 2021).* |

| | |
|---|---|
| Line 38: What causes transience is not the response of rivers to tectonic activity but rather the spatial or temporal change in tectonic forcing itself. So, it would be nice to rephrase the text. | We will revise the sentence.

**\* [Lines 37–40]** *The normalised channel steepness index ($k_{sn}$; Flint, 1974; Wobus et al., 2006) and knickpoint analyses are also frequently applied to explore the transient states of channels caused by tectonic activity (Whipple and Tucker, 1999; Duvall et al., 2004; Kirby and Whipple, 2012; Scherler et al., 2014; Marliyani et al., 2016), as the incision of a channel system is the most obvious response to tectonic uplift.* |
| Lines 40-41: You can elaborate better on why the integral analysis was introduced to the geomorphic community. For example, you could argue that it does not require deriving slope from elevation data to extract ksn or instead focus on the many benefits it presents (e.g., the slope of rivers' long profile in elevation-chi space is equal to ks or ksn, whereas the normal long profile is just the local channel slope). | We agree. So, we will explain the necessity of χ index and its advantages over other indices much clearer.

**\*[Lines 40–43]** *The chi (χ) index was introduced to make up for limitations associated with slope–area analysis for calculating the $k_{sn}$, which is influenced by (1) the noise and errors in topographic data, and (2) the resolution of data itself (Perron and Royden, 2013; Royden and Perron, 2013). Notably, the χ index facilitates handy comparison of $k_{sn}$ values across different channel reaches as the slope of the χ–elevation profile directly shows the $k_{sn}$ value (Perron and Royden, 2013). The χ index is applied to determine whether a landscape under specific conditions is in a steady state or transient state, and to assess long-term drainage divide mobility (Willett et al., 2014; Forte and Whipple, 2018; Kim et al., 2020; Hu et al., 2021; Lee et al., 2021).* |
| Lines 42-43: "enabled the determination of the dynamic evolution of a fluvial system" is awkward and difficult to follow. I guess you want to say that it can be used to determine whether a landscape is in a steady or transient state, given its boundary conditions, and that it can be further used to assess long-term drainage divide instability. | |
| Line 45: The "site-specific parameters" are constrained by empirical data, not simulations, so the phrasing here is strange. With the modelling, you could determine a range of reasonable values given feasible parameters from empirical data. | We will modify the sentence.

**\* [Lines 43–48]** *As computational power has improved and powerful modelling programs have become widely available, it has become possible to simulate landscape evolution. We can test the site-specific parameters constrained by empirical data (e.g., coefficient of diffusivity, coefficient of fluvial erosion efficiency, and local uplift rate) and determine a range of reasonable values through modelling (Tucker et al., 2001; Braun and Willett, 2013; Goren et al., 2014;* |

| | |
|---|---|
| | *Campforts et al., 2017; Hobley et al., 2017; Barnhart et al., 2020; Hutton et al., 2020). It also facilitates the understanding of geomorphic processes and accompanying topographic changes in given tectonic and climatic settings by providing visualisation.* |
| Line 49: "(steady state or transient state, and equilibrium or disequilibrium)" is awkward as they mean the same thing. It may be beneficial to define what you mean by those terms. | We will modify the sentence.
**\* [Lines 49–52]** *These advances have allowed researchers to explain the state (equilibrium or disequilibrium) of the present topography and to predict future landscape evolution within neotectonically active areas (Attal et al., 2011; Reitman et al., 2019; Zebari et al., 2019; Su et al., 2020; He et al., 2021; Hoskins et al., 2023).* |
| Lines 53-54: Most of these studies focus on extracting quantitative information from topographic data to identify spatial and temporal variations in tectonics, climate conditions, and lithology. So, I suggest rephrasing. | We will rephrase that sentence.
**\* [Lines 53–54]** *Most of the above-mentioned studies have focused on explaining how topographic analyses can be applied to identify the spatial and temporal variations in lithological, tectonic, and climatic conditions.* |
| Line 57: I suggest: "… inherited topography is non-negligible because (1) the present…"

Lines 57-60: Although I can understand the rationale for the three points why inherited topography is influential, the writing is challenging to follow. Please rephrase it to make it concise. The following sentence starting with "Therefore" presents the same idea more concisely and clearly. | We will re-structure this part and rephrase the sentences. We will propose our hypothesis first concisely and clearly, and then, provide the reason why we hypothesize the influence of inherited topography in the present topography and topographic metrics.
**\* [Lines 56–63]** *We hypothesize that the influence of inherited topography is non-negligible, and topographic metrics reflect the cumulative influence of past and present geomorphic processes and their drivers. Our hypothesis is grounded in the notion that (1) the present topography is a cumulative expression of tectonic and climatic events from the past to the present, (2) the response time for each geomorphic feature (e.g., longitudinal stream profile, knickpoint migration, and divide migration) to the same tectonic event is different (Whipple et al., 2017), and (3) the timescale that each topographic metrics is also different and not fully understood (Forte and Whipple, 2018).* |
| Line 56: The structure of this paragraph is strange. You start with "We show" before stating what you will do. My point is the simple structure with "In this | We will modify the sentence.
**\* [Lines 56–60]** *We hypothesize that the influence of inherited topography is non-negligible, and the topographic metrics reflect the cumulative influence of past and present geomorphic* |

| | |
|---|---|
| study, we …" should come earlier than 'we show'. | *processes and their drivers.* |
| Line 65: The figure placement is off here, making reading more difficult. I will address figures/captions in the end. | We inserted the figures and captions in the main text, following the author's guide of this journal.

    *Author's guide says: …Figures and tables as well as their captions must be inserted in the main text near the location of the first mention (not appended to the end of the manuscript) and the figure composition must embed any used fonts. …*

If the placement of Figure 1 impairs a readability, we will move this figure between the '1 Introduction' and '2 Study area'. |
| Line 72: Suggestion: "… studying relationships between geology, tectonics and geomorphology". | We will modify the sentence.
**\* [Lines 71–72]** *This area is somewhat uniquely poised for studying relationships between geology, tectonics, and geomorphology.* |
| Line 73: I am not sure 'along' is the right word here. Maybe across? | It was 'about' the UFZ. We will modify the sentence, making it clear.
**\* [Lines 72–74]** *Many studies about the UFZ have initially reported active faults cutting unconsolidated Quaternary-Holocene sedimentary layers, peat layers, and fluvial terraces (Kyung, 1997; Okada et al., 1998; Cheong et al., 2003; Choi et al., 2012b; Kim et al., 2021).* |
| Line 71: The start of this paragraph is strange: "We target an area… as study area." You should first explain to the reader what you want to do, for example, using something like: "To explore the role of inherited topography …, this study …"

Lines 80-89: Okay, so here is the paragraph I expected. As I mentioned in my previous comment, this should come earlier. I would make it the introduction's third paragraph, merging it with the formerly third paragraph. You could use here what is written at the beginning of your conclusion: "The Ulsan Fault Zone (UFZ) has been one of the most | First, we will relocate the paragraph starting with 'In this study, we assess…' ahead of the previous paragraph. Then, the beginning part of conclusion will replace some parts of the paragraph starting with 'We target an area…' to improve comprehension of information and clarity.
**\* [Lines 71–89]** *In this study, we assess the relative tectonic activity along the UFZ using topographic metrics for drainage systems that are relevant to the tectonic activity. (…) Finally, we interpret the influence of inherited topography on the tectono-geomorphic evolution of the study area using the modelling results and topographic metrics, which describe the cumulative influence of past and present geomorphic processes and tectonic activity. We target an area on the southeastern Korean Peninsula around the Ulsan Fault Zone (UFZ), as our study area (Fig. 1). The UFZ has been one of the most active fault zones on the Korean Peninsula since its reactivation ~ 5 Ma. Many studies about the UFZ have initially reported active faults cutting* |

| Comment | Reply |
|---|---|
| active fault zones on the Korean Peninsula since its reactivation ~ 5 Ma. Our study area, the eastern, mountainous, hanging wall block of the UFZ, has undergone regional uplift under an ENE–WSW oriented neotectonic maximum horizontal stress after 5 Ma." This addition would make the introduction more compelling and help give the reader context. Those sentences alone could replace the former third paragraph, making the information more concise and precise. | *unconsolidated Quaternary-Holocene sedimentary layers, peat layers, and fluvial terraces (Kyung, 1997; Okada et al., 1998; Cheong et al., 2003; Choi et al., 2012b; Kim et al., 2021).* Since these pioneering works, three moderate earthquakes ($M_W$ 5.5 in 2016, $M_W$ 5.4 in 2017, and $M_L$ 4.0 in 2023) occurred around this area (Fig. 1a), and micro-earthquakes continue to swarm around and on the fault (Han et al., 2017). Studies have also established geological constraints on the boundary conditions for landscape evolution modelling and the long-term framework for interpreting the influence of inherited topography on the present landscape evolution (Park et al., 2006; Cheon et al., 2012; Son et al., 2015; Kim et al., 2016b; Cheon et al., 2023; Kim et al., 2023a). |
| Lines 83-85: I suggest you rephrase the sentences: "Evaluation of the relative tectonic intensity using geomorphic indices is particularly valuable in the study area. It is challenging to find surface deformation caused by neotectonic faulting in Korea due to low slip rates, rapid physical 85 and chemical erosion, and vast urbanisation." I follow the point, but how it is written makes it difficult to understand. | We will rephrase those sentences.
* **[Lines 83–85]** *Due to low slip rates, rapid physical and chemical erosion, and vast urbanisation, it is challenging to find the evidence of neotectonic faulting in Korea. Therefore, evaluation of the relative tectonic activity using topographic metrics is particularly valuable in the study area.* |

**2 Study area**

| Comment | Reply |
|---|---|
| Lines 95-98: These sentences can use some rephrasing to improve clarity and readability: "Early studies proposed that the main strand of the UFZ is located within the incised valley (Kim, 1973; Kim et al., 1976; Kang, 1979a, b). However, subsequent | We will rephrase these sentences, making it simple and clear.
* **[Lines 95–98]** *Early studies proposed that the main strand of the UFZ is located within the incised valley (Kim, 1973; Kim et al., 1976; Kang, 1979a, b). Later studies suggested that it might be within and around the valley, along the mountain front, or even in both locations (Okada et al., 1998; Ryoo et al., 2002; Choi, 2003; Choi et al., 2006; Ryoo, 2009; Kee et al., 2019; Naik et al., 2022).* |

| | |
|---|---|
| studies have suggested that the UFZ is located either in and around the incised valley, or that it lies along the mountain front to the east of the incised valley, or possibly in both locations (Okada et al., 1998; Ryoo et al., 2002; Choi, 2003; Choi et al., 2006; Ryoo, 2009; Kee et al., 2019; Naik et al., 2022)." It was difficult to understand, particularly the last sentence, which started with 'however'. | |
| Line 115: I prefer 'rivers draining the TMR' to 'channels on the TMR' because rivers flow away from it in opposing directions. Using 'rivers' would be preferable to 'channels' throughout. | In sections introducing the study area or when referring to specific rivers, we will use 'river' instead of 'channel'. However, we think it is more appropriate to use the term 'channel' in most cases throughout this manuscript. The term 'channel' includes the technical meaning as we intended. *\* [Lines 115–116] Rivers draining the TMR are divided into eastern- and western-flank rivers by the main drainage divide (MDD; Fig. 2a).* *\* [Lines 116–117] Rivers draining the eastern flank of the TMR flow to the east and drain directly into the East Sea, whereas those draining the western flank form a more complex drainage system flowing north or southward from a low-elevation valley floor divide.* |
| Lines 116-118: You could merge the two sentences to make them clearer, like "whereas those on the western flank form a more complex drainage system flowing north or southward from a low-elevation valley floor drainage divide." | We will merge those sentences. *\* [Lines 116–117] Rivers draining the eastern flank of the TMR flow to the east and drain directly into the East Sea, whereas those draining the western flank form a more complex drainage system flowing north or southward from a low-elevation valley floor divide.* |
| Lines 126-132: The contrasting western/eastern landscape morphology is very interesting, and the hypotheses you list are even more interesting. They could make a more compelling framing of the narrow problem you will solve with your study. You | Thank you for your constructive comment. If you are suggesting that the width of model domain should remain constant while only the width of uplifted region changes, this approach might not accurately reflect the actual conditions of the UFZ's hanging wall block. The width disparity between northern and southern parts of hanging wall block of the UFZ is more than double, and similar variations are observed in the channel |

| | |
|---|---|
| could potentially map out the empirical consequences of these hypotheses and use your data to test them. | systems of the blocks. Maintaining a consistent model domain width could introduce unforeseen issues, such as response times. Therefore, we opted to model the study area in a way that closely mimics its complex real-world setting, despite the added complexity this approach may entail. |
| Lines 137-138: Can you elaborate more on how you did the calculations? | The calculation method of marine terrace uplift rate is not necessary in this context. So, we revised and added several sentences of caption of Table 1 to explain the marine terrace uplift rate calculation.

**\* [Line 142]** [b] *Paleo-shoreline elevation is the present-day elevation of the paleo-shoreline for each terrace.*

**\* [Lines 143–144]** [c] *Uplifted amount is calculated by subtracting the elevation of the sea level at the marine terrace formation from the paleo-shoreline elevation. We considered the elevation of local sea level of each Marine Isotope Stage (MIS) corresponding to each marine terrace age (Lee et al., 2015; Ryang et al., 2022) in our calculations.*

**\* [Between Lines 145 and 146]** [e] *Uplift rate is calculated by dividing the uplifted amount by the age of marine terrace.* |

**3 Methods**

| Comment | Reply |
|---|---|
| Line 150: 'Topographic analysis' instead of 'Morphometric analysis'. | Thanks. We will change the expression throughout this manuscript.
**\* [Line 150]** *3.1 Topographic analysis*
**\* [Lines 271–272]** *The along-MDD variation and across-MDD contrasts were subsequently compared with results from our topographic analysis to characterise the tectonic activity and spatial variability.*

We will also revise the other parts of this manuscript in the same way. |
| Line 151: Suggestion: "Previous studies of tectonic geomorphology used a variety of topographic metrics to infer relative magnitudes of tectonic | In the first draft of this manuscript, we used the term 'relative tectonic activity', which is used in other studies (Keller and Pinter, 1996; Yildirim, 2014; Luo et al., 2023). We changed the term 'relative tectonic activity' to 'relative tectonic intensity' or 'relative |

| | |
|---|---|
| forcing…" I am unsure if 'intensity' is the best word. Maybe 'magnitude'? | intensity of tectonic activity' (Cao et al., 2022) as some of authors prefer the latter. However, we disagree using 'magnitude' as this term definitely has other meaning than what we intended. So, we are going to use the term 'relative tectonic activity' because it has been used for a long time (Keller and Pinter, 1996; Yildirim, 2014; Luo et al., 2023) and is commonly used by geomorphologists.

**\* [Lines 159–161]** *We used these metrics to assess relative tectonic activity and divide geological segments, though there is very less case study (Lee et al., 2021).*
**\* [Lines 271–272]** *The along-MDD variation and across-MDD contrasts were subsequently compared with results from our topographic analysis to characterise the tectonic activity and its spatial variability.*

We will also revise the other parts of this manuscript in the same way. |
| Lines 151-156: This list of arguably 'old' topographic metrics can be removed from the text as you do not use them in your paper. | We acknowledge concerns regarding the mention of traditional topographic metrics, as they might divert attention from the core findings. Indeed, the $k_{sn}$, Gilbert metrics, and χ index have proven to be the most effective in this domain. Notably, few, if any, studies have used these metrics to assess relative tectonic activity along faults and categorize them into segments as we have. These metrics are primarily designed to analyse topographic equilibrium or disequilibrium of topography. Thus, we included |
| Lines 159-160: "As a result, we adopted and used alternative morphometries, including" should be rephrased. This is a strange way to introduce some of the most successful topographic metrics, perhaps the 'gold standard' of tectonic geomorphology. In contrast, the "old" topographic metrics you point to at the beginning of the paragraph, such as Hack's SL index, are barely used in quantitative topographic studies anymore. Some of the papers I have suggested discuss the reasons for this. So, I suggest you rework this whole paragraph. | traditional metrics to establish a context for introducing the 'new' metrics we adopted, clarifying our rationale for not relying on the 'old' ones. Nevertheless, in response to your feedback, we have toned down the focus on traditional metrics in the revised manuscript (Lines 150–175).
**\* [Lines 151–167]** *We used a 5-m-resolution digital elevation model (DEM) to extract the following topographic metrics: (1) normalised channel steepness index ($k_{sn}$), (2) stream profiles, (3) metrics for assessing drainage divide mobility, and (4) swath profile. These metrics have been widely used to quantitatively measure topography and geomorphic processes across a diverse range of tectonic and climatic settings. We employed these metrics to assess relative tectonic activity and to delineate geology-based fault segments, although there are very few* |

| | |
|---|---|
| | *case studies (Lee et al., 2021). The DEM was generated using digital contours provided by the National Geographic Information Institute (NGII) of the Republic of Korea (https://www.ngii.go.kr/kor/main.do; accessed 14 Sep 2020) and was projected to WGS84 UTM coordinates. We corrected the DEM using 'carving' option of TopoToolbox (Schwanghart and Scherler, 2014) for analysis, which decides the flow route to the deepest path. The channel initiation is determined by the threshold drainage area of $10^5$ $m^2$.* |
| Lines 158-159: The sentences "Further, the study area likely involves low fault slip rates and high rates of physical and chemical erosion, making it difficult to observe the vertical displacement by neotectonic faulting on the surface" should have come earlier in the methods section and been more extensively elaborated. Moreover, the climate context was not described before. | In the introduction section, we mentioned that: **[Lines 83–85]** (this is a modified sentence) *Due to low slip rates, rapid physical and chemical erosion, and vast urbanisation, it is challenging to find the evidence of neotectonic faulting in Korea. Therefore, evaluation of the relative tectonic activity using topographic metrics is particularly valuable in the study area.* In addition, as this manuscript is not about to emphasize the speed of physical and chemical erosion, we did not provide the details of the climatic context. |
| Line 161: Change 'morphometries' to 'topographic metrics' and be consistent throughout the manuscript. I am unsure about the word 'intensity'. Maybe "relative magnitudes of tectonic forcing"? | We will change 'morphometries' to 'topographic metrics' or 'metrics' throughout the manuscript. We made a reply to the same comment about the term 'intensity' above (Line 151). Our reply for your comment was: *In the first draft of this manuscript, we used the term 'relative tectonic activity', which is used in other studies (Keller and Pinter, 1996; Yildirim, 2014; Luo et al., 2023). We changed the term 'relative tectonic activity' to 'relative tectonic intensity' or 'relative intensity of tectonic activity' (Cao et al., 2022) as some of authors prefer the latter. However, we disagree using 'magnitude' as this term definitely has other meaning than what we intended. So, we are going to use the term 'relative tectonic activity' because it has been used for a long time (Keller and Pinter, 1996; Yildirim, 2014; Luo et al., 2023) and is commonly used by geomorphologists.* **\* [Lines 159–161]** *We used these metrics to assess relative tectonic activity and divide geological segments, though there is very less case study (Lee et al., 2021).* |

| | |
|---|---|
| | *[Lines 271–272] The along-MDD variation and across-MDD contrasts were subsequently compared with results from our topographic analysis to characterise the tectonic activity and its spatial variability.*

*[Lines 624–626] We identified UFZ displacement troughs with a relatively low tectonic activity on the basis of geomorphic evidence, such as lows in the swath profile, $k_{sn}$, relief, gradient, $\chi$ index, and channel head elevation along the MDD (Fig. 7).* |
| Line 166: What does 'elevation' of a swath profile mean? A swath profile presents maximum minimum and mean elevation values. I could not follow. | This sentence will be removed after re-structuring this part. |
| Line 164: Add 'normalised' to channel steepness and be consistent. | This sentence will be removed after re-structuring this part. |
| Lines 165-166: The phrasing in "Although the elevation of the swath profile and topographic relief are not the same as cumulative vertical displacement, these two morphometries can reasonably be used as a proxy to infer the latter" is challenging to follow. Please rephrase. Moreover, how and why can these topographic metrics be used to infer cumulative vertical displacement? This is not obvious. Finally, 'morphometries' must be replaced by 'topographic metrics' throughout the manuscript. | First of all, we noted that the high values on the swath profile (i.e., high elevation) and topographic relief are not directly equivalent to the cumulative vertical displacement. However, we infer that if the cumulative vertical displacement is large enough, the average elevation and topographic relief would also be high, provided that the spatial variation in elevation of inherited topography is minimal. This is the basis for our assertion that they can reasonably serve as proxies to infer the cumulative vertical displacement. This idea is quite similar with how (1) the valley floor width to valley height ration and (2) hypsometric integral, which are the traditional topographic metrics, can represent the relative tectonic activity.
We will remove these sentences to avoid any confusion and also change 'morphometries' to 'topographic metrics' throughout the manuscript. |
| Lines 168-169: How and why are these topographic metrics used to identify topographic transience? What does steady or transient state mean here? It should be clearly defined. | This sentence will be removed after re-structuring this part. |
| Lines 150-175: The entire Methods section needs | We will re-structure the entire Method section, following your suggestion. |

rework. Suggestion: a three-subsection structure with 1) Topographic analysis, 2) Cosmogenic nuclide analysis, and 3) Landscape evolution modelling.

For the topographic analysis section, instead of these two long paragraphs, I suggest one single paragraph starting with something like: "We used a 5-m-resolution digital elevation model (DEM) to extract the following topographic metrics: …." Next sentence: "These metrics have been commonly used to reveal the pattern and style of landscape adjustment due to … " "The DEM was generated using digital contours provided by the National Geographic Information Institute (NGII) of the Republic of Korea (https://www.ngii.go.kr/kor/main.do) and was projected to WGS84 UTM coordinates" or something along those lines.

The following paragraph will describe each topographic metric used in the paper. I would start with the normalised channel steepness, then the metrics for assessing long- and short-term drainage divide stability, and finally, the swath profile. Alternatively, you could still have each metric described in its own section, but the initial paragraph should follow something similar to what I described above.

We also modified the first paragraph of '3.1 Topographic analysis' section.

**\* [Lines 151–167]** *We used a 5-m-resolution digital elevation model (DEM) to extract the following topographic metrics: (1) normalised channel steepness index ($k_{sn}$), (2) stream profiles, (3) metrics for assessing drainage divide mobility, and (4) swath profile. These metrics have been widely used to quantitatively measure topography and geomorphic processes across a diverse range of tectonic and climatic settings. We employed these metrics to assess relative tectonic activity and to delineate geology-based fault segments, although there are very few case studies (Lee et al., 2021). The DEM was generated using digital contours provided by the National Geographic Information Institute (NGII) of the Republic of Korea (https://www.ngii.go.kr/kor/main.do; accessed 14 Sep 2020) and was projected to WGS84 UTM coordinates. We corrected the DEM using 'carving' option of TopoToolbox (Schwanghart and Scherler, 2014) for analysis, which decides the flow route to the deepest path. The channel initiation is determined by the threshold drainage area of $10^5$ $m^2$.*

| Line 176-180: Suggestion: "Swath profiles quantify how minimum, mean, and maximum elevation varies across a region along a profile" or something similar instead of the present phrasing.

The following sentence is very confusing and should be reworked: "Swath profiles can be used to investigate and understand the relationship between surface topography and associated or causative variables, such as dynamic topography, which is a topographic change caused by mantle convection (Stephenson et al., 2014), precipitation (Bookhagen and Burbank, 2006), and uplift and exhumation rates (Taylor et al., 2021)." Maybe adding 'or' before "precipitation" does the trick. Additionally, perhaps adding "or spatial and temporal patterns on precipitation" could be helpful. | Thanks. We will rephrase those sentences.
**\* [Lines 176–179]** *Swath profile quantifies how minimum, mean, and maximum elevation varies across a region along a profile. It can be used to understand the relationship between surface topography (i.e., swath profile) and associated or causative variables, such as dynamic topography, which is a topographic change caused by mantle convection (Stephenson et al., 2014), or spatial patterns on precipitation (Bookhagen and Burbank, 2006), and uplift and exhumation rates (Taylor et al., 2021).* |
|---|---|
| Line 180: Why use a line centred on the MDD to produce swath profiles instead of the MDD itself, like Fonte-Boa et al. (2022)? | Yes, that is exactly what we meant. We used the MDD itself as a centreline to produce a swath profile, just like a figure we attached below. As you know, the swath profile shows minimum, mean, and maximum elevations within the area. More precisely, it shows those elevations on the line transverse to the MDD at every (Figure 1 in Hergarten et al., 2014). We set the width of that area as 3 km as we mentioned in the manuscript, 1.5 km each for the left and right sides of the MDD. If the sentence is not clear and can cause misunderstandings, we will revise it.
**\* [Lines 179–181]** *We extracted a swath profile along the MDD and set the width as 3 km, using TopoToolbox (Schwanghart and Scherler, 2014), as along-strike topographic variation is expected to be related to along-strike variation in the cumulative vertical displacement on the UFZ.* |

| | |
|---|---|
| | **MDD**

1.5 km   1.5 km |
| Line 182: Normalised 'channel' steepness index (ksn) | Thanks. We will modify it.
**\* [Line 182]** *3.1.2 Normalised channel steepness index ($k_{sn}$)* |
| Line 183: Use italics for every variable in the main text. | We will change all of them to italics. |
| Line 186: Instead of "… a dimensional coefficient of erosion," use "a dimensional coefficient of fluvial erosion efficiency." Instead of "… and includes", use "encapsulating different controls on erosion, such as …" | Thanks. We will modify the sentence, following your suggestions.
**\* [Lines 186–189]** *where K is a dimensional coefficient of fluvial erosion efficiency with a unit of [$L^{1-2m}$ $T^{-1}$] encapsulating different controls on erosion, such as rock resistance, climate, bedload sediment grain size, and channel width–length relationship (Stock and Montgomery, 1999; Whipple and Tucker, 1999; Snyder et al., 2000; Whipple and Tucker 2002); A [$L^2$] is drainage area; S [$L\,L^{-1}$] is the slope; and m and n are exponents of drainage area and slope, respectively.* |
| Lines 188-189: If you added units for K, do the same for other variables, such as A. | We will add the units.
**\* [Lines 186–189]** *where K is a dimensional coefficient of fluvial erosion efficiency with a unit of [$L^{1-2m}$ $T^{-1}$] encapsulating different controls on erosion, such as rock resistance, climate, bedload sediment grain size, and channel width–length relationship (Stock and Montgomery, 1999; Whipple and Tucker, 1999; Snyder et al., 2000; Whipple and Tucker 2002); A [$L^2$] is drainage area; S [$L\,L^{-1}$] is the slope; and m and n are exponents of drainage area and slope, respectively.* |
| Line 206: 1) You need to justify the choice of reference concavity here. Okay, you could say that you used the same value as most previous studies | We calculated the concavity index of channels using 'mnoptimvar' function in TopoToolbox (Schwanghart and Scherler, 2014). The method that we applied to identify the concavity index optimizes the concavity index by minimizing the variance of |

and that the chosen value is within the range of feasible empirical values. However, there are several studies with a somewhat similar setup as ours (using topographic data to extract quantitative information on the rates of tectonic processes) showing systematic variability in the concavity of bedrock river profiles due to spatial variations in rates of tectonic processes such as Kirby and Whipple (2001), and Clubb et al., (2020). Furthermore, if you have variations in concavity across the UFZ, then you are likely to misinterpret patterns of channel steepness and knickpoints (e.g., Mudd et al., 2018; Gailleton et al., 2021). Thus, I strongly recommend you investigate how concavity varies across the study area. The topographic software you use in the paper (LSDTopoTools and TopoToolbox) have algorithms that can be used to carry out this task readily.

2) You must provide proper information on how you computed ksn from topographic data. For example, how was the DEM hydrologically corrected to ensure channel bed elevation decreases monotonically as you move along the river profile? What was the threshold for channel initiation? What was the flow routing method? Have you computed ksn as the derivative of χ and elevation, which I suppose to be the case given the reference cited

elevation in χ–z-relationship. To investigate the variation in the concavity index across the study area, we calculated it by segments that we divide. The results are shown in the table below.

|  | S1 | S2 | S3 | S4 | S5 |
|---|---|---|---|---|---|
| Concavity index | 0.3597 | 0.5561 | 0.4400 | 0.4665 | 0.4665 |

As can be seen from the results, the reference concavity index that we used does not differ significantly from the result obtained from the topographic data. It also corresponds to the empirical values used in the previous studies. Therefore, we believe that there would be no major problem in using 0.45 as a reference concavity index.

*\* [Line 206] To validate the use of empirical value we use, we calculated concavity indices across the study area, which range from 0.36 to 0.47. Therefore, we believe that using 0.45 as $\theta_{ref}$ should not pose any major issues..*

The pre-processing of DEM for hydrological analysis and decision of channel initiation (channel head) are related to calculating not only normalised steepness index but also the other topographic metrics. So, we added several sentences in the first paragraph of '3.1 Topographic analysis' section.

*\* [Lines 151–167] We used a 5-m-resolution digital elevation model (DEM) to extract the following topographic metrics: (1) normalised channel steepness index ($k_{sn}$), (2) stream profiles, (3) metrics for assessing drainage divide mobility, and (4) swath profile. (…) We corrected the DEM using 'carving' option of TopoToolbox (Schwanghart and Scherler, 2014) for analysis, which decides the flow route to the deepest path. The channel initiation is determined by the threshold drainage area of $10^5$ $m^2$.*

Yes, we computed $k_{sn}$ as the derivative of χ and elevation as you noted, and we cited Mudd et al. (2014). We will modify the sentence for clarity.

*\* [Line 206] We computed $k_{sn}$ as the derivative of χ and elevation as noted by Eq. (4a) with $\theta_{ref}$*

| | |
|---|---|
| (e.g. Mudd et al., 2014)? | *of 0.45, using LSDTopoTools (Mudd et al., 2014).* |
| Line 209: The phrasing needs rework. It is not the 'law of divides' that makes a topographic elevation separating two adjacent hillslopes downslope to opposite sides. | We will modify several sentences in the first paragraph of '3.1.3 Gilbert metrics and the chi (χ) index'.

**\* [Lines 208–211]** *The divide mobility is determined by the contrasts in erosion rates of adjacent drainage basins. As the erosion rates depend on topography, we can use topographic metrics to assess the divide mobility and drivers of divide migration. We used the mean upstream relief which is the most reliable metrics among the Gilbert metrics (Forte and Whipple, 2018) and the χ index to evaluate topographic asymmetry and divide mobility. This is based on the 'law of divides' of Gilbert (1877), which suggested that the steeper slope is expected to be eroded and reduced in height more rapidly when compared with the gentle slope (Fig. 70 in Gilbert, 1877).* |
| Lines 208-211: The phrasing can use some rework. Start explaining divide stability/instability as a function of contrasts in erosion rates in adjacent river basins separated by a drainage divide. Then state that because erosion rates depend on topography, we can use topographic metrics to infer the degree of instability of divides. A helpful reference for your rephrasing here is He et al. (2024). I suggest you describe how one can use topographic data to predict drainage divide migration direction similarly to He et al. (2024). Moreover, I cannot understand why you use three different metrics (i—mean upstream relief, ii—mean upstream gradient, and iii—channel head elevation) throughout the manuscript that yields a similar measurement (i.e., cross-divide steepness asymmetry). You need a single metric instead. I suggest you use the across-divide difference in hillslope relief (ΔHR) normalised by the across-divide sum in hillslope relief (∑HR), referred to as the divide asymmetry index (DAI) introduced by Scherler and Schwanghart (2020) and readily | This research has two major aims. The first one is to assess the relative tectonic activity with a variety of topographic metrics and divide geological segment. The second one is to evaluate the topographic asymmetry and build landscape evolution model for the hanging wall of UFZ. We tried to cross-check the different topographic metrics and achieve those two aims. The single metric that you recommend (i.e., DAI) is appropriate to achieve only second goal, but the metrics we used are well-suited for achieving both gaols. As you mentioned, the channel head elevation and mean upstream gradient may not be reliable, and this is also proposed by Forte and Whipple (2018) which suggested using Gilbert metrics and χ index to evaluate the topographic asymmetry and divide mobility. Forte and Whipple (2018) also recommended to use mean upstream relief and χ index because those are the most reliable metrics, and they may represent the different timescale for the landscape. So, we are going to use mean upstream relief and χ index only according to your comment and Forte and Whipple (2018).

In addition, we decided the location of channel head with the threshold area of $10^5$ m², |

| | |
|---|---|
| implemented in TopoToolbox. Alternatively, you could compute variations in mean upstream relief using TAK. Channel head elevation and mean upstream gradient are unnecessary here. For example, you do not have accurate information about channel heads and have not extracted them using a more sophisticated algorithm (e.g., Clubb et al., 2014). | following the classical method. We admit that this method is not as sophisticated as the algorithm to extract the channel head that you recommended (Clubb et al., 2014). However, we think that it does not influence the relief result so much, so are going to keep our result. |
| Lines 213: Add 'long-term' to the phrase "evaluate long-term divide stability". | We will modify the sentence.

**\* [Lines 212–214]** *In addition to these metrics, the chi ($\chi$) index at opposing channel heads can also be used to evaluate long-term divide stability (Willett et al., 2014; Forte and Whipple, 2018).* |
| Line 218: Start with "where x is the distance upstream from an arbitrary baselevel zb (at x = xb)". Remove the "x' dummy variable" part. | We will modify the sentence.

**\* [Lines 218–219]** *where x is the distance upstream from an arbitrary base-level, $z_b$ is a base-level elevation (at $x = x_b$), $A_0$ is an arbitrary scaling area, and $A(x)$ is the drainage area at point x on the channel.* |
| Line 220: Delete "by multiplying by A_0 as a coefficient". This sentence can be integrated with the previous one for conciseness. | We will remove that phrase 'by multiplying by $A_0$ as a coefficient'. However, for clarity, we will keep this sentence separate from the previous one.

**\* [Lines 219–220]** *The integrand in Eq. (4b) becomes dimensionless, meaning that the $\chi$ index can be expressed with a unit of length (Perron and Royden, 2013).* |
| Lines 220-221: The sentence "Equation (4a) illustrates the linear relationship between elevation and the $\chi$ index for a steady-state channel" needs some rephrasing to make clear that this is the case when rock uplift, bedrock erodibility, and climate conditions are invariant along-profile, provided that the reference concavity is adequate. As such, spatial variations in these boundary conditions will | We will modify the sentence.

**\* [Lines 220–221]** *Equation (4a) establishes the linear relationship between the elevation and $\chi$ index when the rock uplift, bedrock erodibility, and climate conditions are invariant along the channel, and the $\chi$ index is calculated with the adequate $\theta_{ref}$. If such boundary conditions spatially vary, the elevation and $\chi$ index will have piecewise-linear relationship.* |

| | |
|---|---|
| be expressed by non-linear shapes in elevation-chi profiles. Moreover, I suggest using "establishes" instead of "illustrates". | |
| Lines 222-223: I would move the sentence "Because the χ index is sensitive to the base-level elevation (zb; Forte and Whipple, 2018), we analysed the χ index with two different base-level elevations" to the end of the paragraph, using "Finally, because … we calculated the χ metric assuming two different …" | The sentence in lines 222–223 means that we analysed the χ index with two different base-level elevations (50 m and 200 m). The following sentences in the lines 223–226 is explaining why we decided the base-level elevations as 50 m and 200 m for χ index analysis. So, if the sentence "Because the χ index is sensitive to the base-level elevation ($z_b$; Forte and Whipple, 2018), we analysed the χ index with two different base-level elevations." should be relocated to the end of the paragraph, the following sentences explaining the reason for our decision of base-level elevations (50 m and 200 m) also should be relocated. We will move this part at the end of the paragraph. |
| Lines 223-228: I could not follow the text or the rationale. The most straightforward baselevel elevation for extracting the drainage network in your case should be 0 m a.s.l. It is hard to imagine that using a baselevel elevation of less than 50 m, you extract fewer river segments than using a higher baselevel elevation. For instance, your Fig. 6 shows the opposite, with more river segments extracted using baselevel = 50 m than when baselevel = 200 m. Therefore, I strongly recommend you extract the drainage network using baselevel 0 m. This would allow the extraction of complete drainage networks, facilitating the visualisation of patterns in ksn and knickpoints. For example, it is challenging to grasp spatial patterns in ksn, chi, or flow directions, in Fig. 6. | Yes, we agree with you in that the most straightforward base-level of channels in our study area is 0 m a.s.l. However, with lower base-level elevation, the extracted channel should have more river segments than the channel extracted with a higher base-level elevation.
[Figure]
 Please look at the figure on the left side. Each channel segment is distinguished by a different colour. For example, if I extract channel with the base-level elevation of 50 m, the extracted channel will include eight segments. However, with the same channel, if I extract channel with the base-level elevation of 200 m, the extracted channel will include only two segments. So, more river segments can be extracted |
| Lines 226-227: It is necessary to describe how you | channel will include only two segments. So, more river segments can be extracted |

extracted Gilbert metrics and performed the chi-transformation with sufficient detail to ensure the reproducibility of the results. There is nearly no information about the parameters or algorithms used here.

when we use a lower base-level elevation.

[Figure]

In the present study, we did not extract the drainage network using base-level elevation of 0 m. This is because the individual drainages cover too wide area to describe the variation of the topographic metrics along the UFZ when we extract the drainage network with the base-level elevation of 0 m. Please look at the figure on the left side. This figure is a drainage network extracted with the base-level elevation of 0 m. As you can see, the western flank of the MDD has only two big drainages, which are called 'Hyeongsangang' and 'Taehwagang' rivers in the Figure 2. If we extract the drainage network with the base-level elevation of 0 m, we cannot trace the variation of the topographic metrics, evaluate the relative tectonic activity, and finally divide the geological segment of the fault (zone), which is one of the major aims of this study.

However, we admit that our expression of the reason why we decided those two elevations for χ index calculation. We will rephrase those sentences for clarity and add some sentences to provide information about how we calculated those parameters.

**\* [Lines 222–228]** *We used TopoToolbox (Schwanghart and Scherler, 2014) and DivideTools (Forte and Whipple, 2018) to analyse Gilbert metrics and the χ index. The mean upstream relief*

| | |
|---|---|
| | *and mean upstream gradient among the Gilbert metrics is calculated within the radius of 200 m, considering the resolution of topographic data and the distance between the channel head and MDD. Finally, because the χ index is sensitive to the base-level elevation ($z_b$; Forte and Whipple, 2018), we analysed the χ index with two different base-level elevations (50 and 200 m). Those base-level elevations were applied as the numbers of drainage networks extracted with the base-level elevations lower than 50 m and higher than 200 m are not enough to describe the variation of topographic metrics along the UFZ. We then performed Student's t-test (two-tailed, α = 0.05) to determine whether two groups are statistically significantly different from each other. We applied this Student's t-test (two-tailed, α = 0.05) to statistically compare the values of these topographic metrics between the western and eastern flanks of the TMR.* |
| Line 229: Suggestion: "River profile analysis and knickpoint extraction" | We will modify the sentence.

**\* [Line 229]** *3.1.2 Stream profile analysis and knickpoint extraction* |
| Lines 230-234: Why have you framed the analysis using log S-log A profiles? You are not analysing logS-logA profiles. So, I suggest focusing on the shape of river profiles on elevation-distance or elevation-chi spaces. | The first two sentences are general description of the longitudinal stream profile with equations (3a) and (4a). We will remove the phrases related to log S–log A relationship from the third sentence.

**\*[Lines 231–233]** *However, rivers in transient states are expected to show several piecewise linear segments in a χ-transformed stream profile (Perron and Royden, 2013).* |
| Line 234: Rephrase the sentence, "The boundary between adjacent piecewise lines can be identified physically as knickpoints." Delete 'physically'. Moreover, defining what you mean by knickpoint from the first use is necessary. | We will modify this sentence.

**\* [Line 233]** *The boundary between adjacent piecewise lines can be identified as a knickpoint, which is a part of a channel with an abrupt change in slope and elevation of channel bed.* |
| Line 235: Suggestion: "or exposure of a previously buried rock-type". | We will modify this sentence.

**\* [Lines 233–236]** *A knickpoint can reflect the transient state of a stream that is caused by a base-level change related to climatic change (Crosby and Whipple, 2006), tectonic forcing (Snyder et al., 2000; Kirby and Whipple, 2001), or lithological difference (Cyr et al., 2014).* |
| Lines 236-237: More detail is needed here to describe how you extracted long profiles and | We will add more details for extraction of longitudinal stream profiles and χ-transformed stream profiles. |

| | |
|---|---|
| performed the integral transformation of the x coordinate. Otherwise, one could not reproduce any of your results. Did you use carving or filling procedures to hydrologically correct the DEM? Did you use a threshold for channel initiation? | **\* [Lines 237–239]** *We used TopoToolbox (Schwanghart and Scherler, 2014) to extract the longitudinal stream profiles. To visualize the changes in normalised channel steepness index more easily, we extracted the χ-transformed stream profiles, using LSDTopoTools (Mudd et al., 2014). This tool employs an algorithm to analyse the best fitting piecewise line for each channel segment (Mudd et al., 2014). We set the reference concavity index ($\theta_{ref}$) to 0.45 and the reference scaling area ($A_0$) to unity for integral transformation of the χ coordinate.*

However, the pre-processing of DEM for hydrological analysis and decision of channel initiation (channel head) are related to other topographic analysis, such as normalised channel steepness index and mean upstream relief. So, we added several sentences in the first paragraph of '3.1 Topographic analysis' section.
**\* [Lines 151–167]** *We used a 5-m-resolution digital elevation model (DEM) to extract the following topographic metrics: (1) normalised channel steepness index ($k_{sn}$), (2) stream profiles, (3) metrics for assessing drainage divide mobility, and (4) swath profile. (…) We corrected the DEM using 'carving' option of TopoToolbox (Schwanghart and Scherler, 2014) for analysis, which decides the flow route to the deepest path. The channel initiation is determined by the threshold drainage area of $10^5$ $m^2$.* |
| Lines 239-242: 1) Much more detail is needed here. If you used the method Gailleton et al. (2019) introduced to extract knickpoints, then you need to describe the user-defined parametrisation. Otherwise, your approach is not reproducible, which is precisely the point the method introduced by Gailleton et al. (2019) addressed. 2) These sentences highlight why you should not frame the beginning of this sentence based on patterns of slope-area data. | Thanks. We acknowledge the need for detailed methodological descriptions to ensure reproducibility, which Gailleton et al., (2019) emphasized.
The key user-defined parameter for in their knickpoint extraction algorithm is the regulation parameter for the Total Variation Denoising, so called 'TVD_lambda'. We set 'TVD_lambda' to 400, with all other parameters at their default values. We chose this specific value because (1) it aligns well with our reference concavity index of 0.45, and (2) we wanted to exclude knickpoints with minimal changes in $\Delta k_{sn}$. We recognize this might seem overly detailed for inclusion in the main text.

Regarding the initial discussion on longitudinal stream profiles and the log S–log A relationship, it was intended to simplify the explanation of stream profile forms before |

| | introducing the more complex χ-transformed stream profiles. However, to avoid potential misinterpretation, we are open to removing the introductory sentence if it leads to confusion. |
|---|---|
| Line 244: I suggest you define what you mean by a steady state in the first usage. Otherwise, the text gets confusing. | We have added a detailed explanation of the term "the steady state" at its first mention to enhance clarity and understanding.
**\* [Lines 244–245]** *Assuming that the channel of interest approaches a topographic steady state where the channel bed keeps constant elevation due to the balance between uplift and incision, uplift rate can be derived from the bedrock channel incision rate [Eqs. (1) and (2)].* |
| Line 249: Instead of "represents", use "can be interpreted as …". It is also necessary to add a few citations after this sentence. | We will modify the expression.
**\* [Lines 249–250]** *The concentration of in situ cosmogenic $^{10}$Be from riverine sediment on the present bedrock channels can be interpreted as the catchment-averaged denudation rate (CADR).*

The references are in the following sentence. We did not add the references in this sentence as it shares the same references with the following sentence. |
| Lines 251-252: This was a good example of using 'steady-state'. | Thank you. |
| Line 253: Change one of the two 'during' in the sentence for a synonym. | We will rephrase the sentence.
**\* [Lines 252–254]** *Thus, the CADR represents the average denudation rate across the entire catchment by hillslope and fluvial processes over a given integration time, during which the sediments remained within the catchment (Granger et al., 1996; von Blanckenburg, 2005).* |
| Line 258: The placement of the figures makes it difficult to read the manuscript. | We inserted the figures and captions in the main text, following the author's guide of this journal.
    *Author's guide says: …Figures and tables as well as their captions must be inserted in the main text near the location of the first mention (not appended to the end of the manuscript) and the figure composition must embed any used fonts. …*
If the placement of Figure 3 harms a readability, we will move this figure between the |

| | '3.2 Cosmogenic nuclide analysis' and '3.2.1 Catchment-averaged denudation rate'. |
|---|---|
| Lines 269-273: Did the sample strategy target catchments with comparable upstream drainage areas? This is not clear. | The table below shows the upstream area for each sampling site.

| Sample name | Upstream area (m$^2$) | Sample name | Upstream area (m$^2$) |
|---|---|---|---|
| W1 | 4,171,925 | E1 | 2,451,375 |
| W2 | 12,169,575 | E2 | 3,088,775 |
| W3 | 208,125 | E3 | 1,238,575 |
| W4 | 2,177,300 | E4 | 2,413,200 |
| W5 | 1,055,075 | E5 | 1,323,375 |
| W6 | 234,300 | E6 | 1,998,375 |
| W7 | 1,940,225 | E7 | 914,225 |
| W8 | 1,629,850 | E8 | 1,653,275 |

Some paired basins (e.g., W4–E4, W5–E5, and W8–E8) have comparable upstream drainage areas, while the others do not. |
| Line 270: 1) 'Document' instead of 'trace'. 2) I could not follow. Maybe explain that you want to compare across and along-MDD variations in CADRs instead of having the complicated sentence "…to trace variations in the CADR along the MDD and to compare the CADRs of the western and eastern flanks of the TMR". | We will rephrase the sentence.
*  [Lines 269–271] *We collected 16 samples of riverine sediment from eight pairs of catchments (a total of 16 catchments) along the MDD of the TMR (Fig. 3a) to document variations in the CADR along the MDD. In addition, we also compare the CADRs of the western and eastern flanks of the TMR to reveal the direction of divide migration.* |
| Line 271: Suggestion "topographic" instead of "morphometric". | We will make a change.
*  [Lines 271–272] *The along-MDD variation and across-MDD contrasts were subsequently compared with results from our topographic analysis to characterise the tectonic intensity and its spatial variability.* |
| Lines 273-274: Rephrase the sentence "We avoided collecting samples from: (1) catchments containing golf courses and (2) downstream areas | We will modify the sentence.
*  [Lines 273–274] *To prevent possible contamination by anthropogenic debris, we avoided collecting samples from the catchments containing golf courses and downstream areas where alluvial fans are located, and faults occur (Fig. 2).* |

| | |
|---|---|
| where alluvial fans are located, and faults occur (Fig. 2) to avoid possible contamination by anthropogenic debris" to "To avoid possible contamination by anthropogenic debris, we …" | |
| Lines 275-280: 1) Rework the sentences: "The basins W1 and E1 contain rhyolite and dacite bedrock. The basins W2, W3, E2, and E3 contain rhyolite, dacite, and granite bedrock. The other basins (W4–W8 and E4–E8; eight basins) contain sedimentary, volcanoclastic, and granite bedrock." You can combine them into a single sentence starting with "… For example, …"

 2) It was unclear from reading these sentences how you explored potential lithological variations in the results. This seems important given that you do have lithological variations along the MDD and are assessing how erosion varies along the MDD. There is no subsequent table or figure with any lithological information (e.g., distribution of rock types per sampled catchment or the areal contribution of quartz-bearing lithologies). Please add the sampled catchments in Fig.1b. | 1) We will rephrase the sentences.
 * **[Lines 274–280]** *The basins W1 and E1 contain rhyolite and dacite bedrock. The basins W2, W3, E2, and E3 contain rhyolite, dacite, and granite bedrock (Fig. 1b). The other basins (W4–W8 and E4–E8; eight basins) contain sedimentary, volcanoclastic, and granite bedrock.*

 2) We did not account for the lithological variation across the catchments when we calculate CADRs. In this study, interpretations drawn from the variations in CADRs along the MDD were minimal, acknowledging that the lithological variations could influence the CADR values as you pointed out. Instead, we primarily focused on comparing CADRs of paired basins, which are adjacent across the MDD (e.g., W1–E1) where lithological differences are not significant (see figure below). This approach helped support our analysis of the relationship between divide migration and erosion rates.

 We will add the locations of the sampled catchments in Fig. 1b. |

[Figure]

| | |
|---|---|
| Line 281: Suggestion: "following a standard protocol" instead of "following the standard protocol". | We will revise the sentence.

 * **[Lines 281–282]** *We performed chemical treatment of the CADR samples at Korea University, Seoul, South Korea, following a standard protocol for $^{10}$Be extraction (Kohl and Nishiizumi, 1992; Seong et al., 2016).* |
| Line 291: Later in the text (Line 314), you state you used the "CRONUS-Earth online calculator (Balco et al., 2008; version 3), applying the LSDn scaling scheme (Lifton et al., 2014)." Why use a different approach to estimate erosion rates from cosmogenic 10Be abundances here? I suggest you be consistent throughout the study.

 Furthermore, if you had access to Mudd et al.'s (2016) CAIRN program, which uses the same software you used to calculate topographic metrics (i.e., LSDTopoTools), you could have used it to calculate catchment-averaged atmospheric pressure using CAIRN, which could then be fed into CRONUS-Earth to estimate catchment-averaged denudation rates.

 Finally, you should not use topographic shielding to compute catchment-averaged denudation rates for your study area (see DiBiase, 2018). I suggest you recalculate your rates. | The CRONUS-Earth online calculator provides (1) exposure age calculation, (2) (bedrock) erosion rate calculation, and (3) production rate calibration functions, and does not provide the CADR calculation. This is why we used CRONUS-Earth online calculator to calculate the exposure age of the strath terrace and the present channel bed in the text (Line 314) and did not use it to calculate CADR.

 Yes, we could have used CAIRN to calculate the CADR directly. Or, we also could have used CAIRN to calculate the catchment-averaged atmospheric pressure and could have fed it into CRONUS-Earth to calculate CADR. However, the latter one that you recommended is old-fashioned way when there are no tools to calculate the production rate cell-by-cell with topographic data. The tools CAIRN and BASINGA are the ones that provide the cell-by-cell calculation of production rate so that they can draw more precise result. In addition, BASINGA provides the geomagnetic correction (Muscheler et al., 2005). That is why we used the BASINGA to calculate the CADR.

 We re-calculated CADR without topographic shielding, using BASINGA. We updated the new result below to the Table 2 and the text.

| Sample name | CADR (mm kyr$^{-1}$) | Sample name | CADR (mm kyr$^{-1}$) |
|---|---|---|---|
| W1 | 32.94 ± 2.01 | E1 | 36.65 ± 2.32 |
| W2 | 55.94 ± 3.56 | E2 | 40.52 ± 2.49 |
| W3 | 155.23 ± 15.35 | E3 | 104.85 ± 7.71 |
| W4 | 100.56 ± 7.73 | E4 | 34.91 ± 2.14 |
| W5 | 111.27 ± 8.51 | E5 | 7.35 ± 0.43 |
| W6 | 55.90 ± 3.66 | E6 | 34.72 ± 2.11 |
| W7 | 35.95 ± 2.22 | E7 | 18.02 ± 1.07 |
| W8 | 16.89 ± 1.00 | E8 | 44.34 ± 2.81 |

| | |
|---|---|
| Line 311: Rephrase "following laboratory protocol…" to something like "following the same laboratory protocol described above…" | We will rephrase it.
**\* [Lines 311–312]** *Following the same laboratory protocol described above (Kohl and Nishiizumi, 1992; Seong et al., 2016), we performed physical and chemical treatment for in situ surface exposure dating samples at Korea University, Seoul, South Korea.* |
| Lines 314-315: 1) Again, I suggest you calculate cosmogenically-derived erosion/exposure ages in a consistent manner. 2) Suggestion: Use 'uncertainty' instead of 'error' here. | 1) As we mentioned above, CRONUS-Earth online calculator is an appropriate tool to calculate the exposure age of bedrock surface. So, we will keep this manner.
2) We will consistently use 'uncertainty' throughout this manuscript.
**\* [Line 315]** *Uncertainties of exposure ages were calculated and are given as 1σ values.* |
| Line 317: Suggestion: "Landscape evolution modelling" instead of "Modelling landscape evolution". | We will modify it.
**\* [Line 317]** *3.3 Landscape evolution modelling* |
| Line 318: 1) Delete 'next'. 2) Use "landscape evolution model toolkit …" instead of "landscape evolution model". | We will modify that sentence.
**\* [Lines 318–319]** *We applied the open-source landscape evolution model toolkit 'Landlab' (Hobley et al., 2017; Barnhart et al., 2020; Hutton et al., 2020) to investigate the specific landscape evolution model setups to get insights about the evolution of the uplifted eastern hanging wall block of the UFZ.* |
| Line 319: Rephrase. You will investigate the evolution of specific model setups to get insights about the evolution of the uplifted eastern hanging wall block of the UFZ rather than "comprehensively investigating" its evolution. | We will modify that sentence.
**\* [Lines 318–319]** *We applied the open-source landscape evolution model toolkit 'Landlab' (Hobley et al., 2017; Barnhart et al., 2020; Hutton et al., 2020) to investigate the specific landscape evolution model setups to get insights about the evolution of the uplifted eastern hanging wall block of the UFZ.* |
| Lines 319-320: Suggestion: "… These simulations were compared to results from topographic analysis and 10Be measurements …". Delete "in conjunction with measured geomorphic indices". | We will modify the sentence.
**\* [Lines 319–321]** *These simulations were compared to results from topographic analysis and $^{10}$Be measurements and to interpret the landscape evolution of the study area.* |
| Line 324: This parametrisation for stream-power river incision is awkward and distant from the | We appreciate your detailed observations concerning the parametrization values used in our stream-power river incision model. The values we employed were derived from |

commonly used values in modelling studies or reported values from empirical studies. First, the K value seems to be way too low, especially given the tectonic context of the study area (compare, for example, with values reported by Stock and Montgomery, 1999; Whipple et al., 2000; Kirby and Whipple, 2001; Zondervan et al., 2020; and Peifer et al., 2021). This will have significant implications for your simulations involving perturbation phases related to changes in the tectonic field. More reasonable values would be between 10-5 to 10-6. Furthermore, you have m and n values that are also awkward. The most straightforward parametrisation would have n = 1, while m could vary between 0.4 to 0.6. In this n = 1 case, the river response to the perturbation does not depend on the channel slope. In contrast, when n > 1, the river response depends on channel slope, leading to complexity that we do not understand fully. So, while it would be okay to have scenarios using a set of stream-power parameters such as m = 0.6 and n = 1.5, it should not be your only parametrisation scenario.

Suggestions: 1) use your catchment-averaged denudation rates to parametrise reasonable values for K (e.g., Gallen (2018)); 2) perform a sensitivity analysis with different parametrisations using

an extensive review of global $^{10}$Be denudation rates by Harel et al. (2016), which considered factors such as vegetation, climate, seismicity, tectonic activity, and glaciated status. This study provided a comprehensive dataset from which we extracted values representative of regions with lithological, climatic, and tectonic characteristics similar to our study area. Based on this, we averaged the parameters suitable for our specific geological and environmental context.

Moreover, we acknowledge the range of the range of K values reported in the literature, as highlighted in your references (e.g., Stock and Montgomery, 1999; Kirby and Whipple, 2001; Zondervan et al., 2020), which span from $10^{-7}$ to $10^{-2}$. These variations largely depend on factors such as lithology, uplift rate, and climate, aligning with our choice of parametrization tailored to our study area's specific conditions.

In response to your suggestion, we have initiated a sensitivity analysis to explore how different values of the erosion coefficient (K) and the exponents for drainage area (m) and slope (n) affect the model outcomes. This analysis is currently in progress and will be included as supplementary data to provide a robust basis for understanding the implications of varying these parameters.

We believe these steps will enhance the robustness of our modelling approach and provide clarity on the impacts of different parametrization scenarios on our results.

| empirically feasible values, preferably with values higher and lower than the obtained in 1. | |
|---|---|
| Line 326: Why did you use an incision threshold? This is not clear, and even if you have a compelling explanation, it should not be your only parametrisation. Most modelling studies do not account for incision thresholds. So, why is it important here? If you want to include it, I suggest you perform a sensitivity analysis to determine its influence on simulations. | The incision threshold was employed to limit landscape erosion, based on the premise that sufficient shear stress is necessary for incising topographic surfaces. This concept suggests that the initiation of incision is controlled by the threshold value (Snyder et al., 2003; Theodoratos and Kirchner, 2020). The inclusion of an incision threshold in our landscape evolution model follows the nonlinear relationship between erosion rate and channel steepness, as described in Desormeaux et al. (2022). Moreover studies such as Harel et al. (2016), which analysed the parameters of the stream power law using global $^{10}$Be denudation rates, also considered incision threshold in their analysis (Harel et al., 2016). Thus, we tried to consider incision threshold. We adopted a commonly used value ($10^{-5}$ m yr$^{-1}$; Hobley et al., 2017) in our landscape evolution model. |
| Line 330: Add units for the diffusivity coefficient. | We will add the unit of diffusivity coefficient.

**\* [Line 330]** *where $K_d$ is the coefficient of diffusivity with a unit of [$L^2$ $T^{-1}$]; $\nabla^2$ is the Laplace operator, which is the divergence of gradient; and z is elevation.* |
| Line 331: While the Kd value does seem reasonable, a sensitivity analysis with lower and higher values is also necessary. One scenario could perhaps scale K/Kd similarly to Whipple et al. (2017), with K/Kd = 0.002. | We fully acknowledge the importance of validating the $K_d$ value. However, our goal in this study is not to generalize our hypothesis nor find the specific boundary conditions but rather to reveal the landscape evolution process of our study area. Therefore, we intend to utilize this $K_d$ value without conducting a sensitivity analysis as long as the value falls within a reasonable range. |
| Line 335: Change the signal for the hillslope erosion to plus in the equation. | Equation (6) in this manuscript illustrates that the topography gains height by tectonic uplift ($U$) and loses height by fluvial erosion ($KA^mS^n$) and hillslope diffusion ($K_d\nabla^2z$). So, we think the signal for the hillslope erosion is minus. The same equation is used in Zebari et al. (2019). We will keep this equation. |
| Lines 355-358: The rationale for the modelling | Thank you for the constructive comment. We acknowledge the need for clarity |

| | |
|---|---|
| setup, with the two phases, should be better introduced earlier in the text. This would improve the readability of the text considerably. | regarding our modelling setup. We will introduce rationale for the two-phase modeling approach earlier in the manuscript to enhance readability.

**\* [Line 354]** *We designed the landscape evolution model to incorporate two stages: the first to establish the inherited topography and the second to simulate the fault movement (Fig. 4a). By applying different boundary conditions during the first stage (Fig. 4b), we could simulate various inherited topographies. This approach allowed us to test our hypothesis that the inherited topography significantly influences the present landscape and the patterns of topographic metrics.* |
| Lines 356-357: Although reasonable, as it was configured based on the constraints for the study area, having the first stage running for only 3 Myr feels somewhat awkward, given that landscape equilibration concerning phase 1 will be important later in the paper. This feels particularly important, given the usage of such a low K value (K = 5.56E-07). From this K alone, I'd expect that reaching a steady state configuration could take two orders of magnitude longer than 3 Myr. So, my first suggestion would be to run additional scenarios with different durations for phases 1 and 2. Because river response timescales depend on K (and S in case n is more than 1), the sensitivity tests will be critical to evaluate simulation outputs. | Thank you for your observations regarding the duration of the model phases and the associated K values used in our simulation. We acknowledge the concerns about the equilibration time necessary to reach a steady state in our landscape evolution model.

In our simulations, we observed that the modelled landscape achieved a dynamic equilibrium state approximately 2.4 Ma into the stage 1. This quicker equilibrium is attributed to the relatively low uplift rate (80 mm kyr$^{-1}$) and to the coarse grid spacing (100 m) used in our model. These conditions facilitated a faster approach towards equilibrium within the prescribed time frame of the stage 1.

Regarding the duration of the stages, extending the time frames for stages 1 and 2 may introduce complexities such as defining an appropriate duration that accurately reflects the geological settings of the UFZ. It raises questions about the representativeness of prolonged simulation durations for our specific study context.

To further substantiate our model settings and address your concerns, we are currently conducting sensitivity analyses concerning the K value. This analysis aims to elucidate the impact of varying K on the time scales required for reaching equilibrium. The results |

| | of these sensitivity tests will be included as supplementary data to provide comprehensive insights into the effects of these parameters on our simulation outcomes. We believe these efforts will enhance the robustness and relevance of our findings to the unique geological characteristics of the UFZ. |
|---|---|
| Lines 367-370: 1) This sort of key information: "This assumption is based on the overall tendency of high-east and low-west topography of the Korean Peninsula, supported by the long-term, regional westward tilting that was initiated during the Middle Miocene when the East Sea started to widen, and since which time the strongly asymmetric (high-east) Taebaek Mountain Range has been rapidly uplifted (Min et al., 2010; Kim et al., 2020)" should have come way earlier in the text. It would help frame the narrow problem this study is addressing and give the reader more context. 2) The Taebaek Mountain Range was not mentioned earlier in the text and was not identified in previous figures. As such, it is difficult to follow. | We acknowledge the need for a clearer presentation of the geological context in the "2. Study area' section. To address this, we will reorganize this section to provide a more comprehensive background on the regional geological development. Specifically, we will elaborate on significant geologic events such as the opening of East Sea, the formation of asymmetric Taebaek Mountain Range, and the most recent marine terrace formation along the west and east coasts of Korea Peninsula. We are sure this enhancement will offer a more robust framework for understanding the geological setting of our study area. |
| Lines 370-373: Again, this critical information should have come earlier, as it would greatly help framing the narrow question addressed by the paper: "In addition, the shore platform on the western coast of the peninsula (0 m a.s.l.; Choi et al., 2012a; Jeong et al., 2021) and marine terraces along the eastern coast (18–45 m a.s.l.; Choi et al., 2003a, b; Kim et al., 2007; Heo et al., 2014; Lee et | |

| | |
|---|---|
| al., 2015), formed at the same time (i.e., during MIS 5), indicate that this regional differential uplift has lasted until very recently." Please rephrase to clarify why this indicates that regional differential uplift lasted until recently. Also, define 'MIS 5'. | |
| Lines 373-375: Why have you not created a single model domain with a wider 'uplifted' region in the north and a narrower uplift region toward the south? Manipulating the uplift field could potentially achieve this. Having one single model domain, including both the northern and southern parts of the block, would make the paper easier to read and the results more clearly interpretable. | Thank you for your constructive comment. If you are suggesting that the width of model domain should remain constant while only the width of uplifted region changes, this approach might not accurately reflect the actual conditions of the UFZ's hanging wall block. The width disparity between northern and southern parts of hanging wall block of the UFZ is more than double, and similar variations are observed in the channel systems of the blocks. Maintaining a consistent model domain width could introduce unforeseen issues, such as response times. Therefore, we opted to model the study area in a way that closely mimics its complex real-world setting, despite the added complexity this approach may entail. |
| Lines 380-383: While I understand the setup used for the UFZ, it would be beneficial to perform a sensitivity analysis for the distance between the MDD and the UFZ (e.g., having the UFZ initially closer to the MDD). | We fully acknowledge the importance of conducting a sensitivity analysis on the distance between the MDD and the UFZ as it could influence the response time. However, we have decided not to conduct the sensitivity analysis because our primary objective of this study is to demonstrate the applicability of our hypothesis under the specific conditions of the UFZ. |
| Lines 382-385: As discussed above, sensitivity analysis is essential for these parameters (K, Kd, m, and n). | We have been conducting sensitivity analyses K, m, and n, and we will provide the results as supplementary data. We have decided to use $K_d$ value of 0.001 without sensitivity analysis, as this value falls within a reasonable range. |
| Lines 389-390: This key information needs to come earlier in the text (introduction/study area). Please elaborate more. Until this point, we had no information about the 'backbone' mountain range of the Korean Peninsula, for which long-term | We acknowledge the need for a clearer presentation of the geological context in the "2. Study area' section. To address this, we will reorganize this section to provide a more comprehensive background on the regional geological development. Specifically, we will elaborate on significant geologic events such as the opening of East Sea, the formation of asymmetric Taebaek Mountain Range, and the most recent marine terrace |

| | |
|---|---|
| exhumation rates are available. This is important. | formation along the west and east coasts of Korea Peninsula. We are sure this enhancement will offer a more robust framework for understanding the geological setting of our study area. |
| Lines 394-396: While the parametrisation for U in phase 1 does seem reasonable, a sensitivity analysis is necessary. | We acknowledge the importance of validating the uplift rate (U) value through sensitivity analysis. However, our current study prioritizes understanding the landscape evolution around the UFZ. Therefore, despite the benefits of such sensitivity analysis for U, we will proceed with the suggested U values in our model if the values fall within reasonable ranges. |
| Lines 398-416: 1) Rework the text as it is slightly confusing and difficult to follow. 2) The chosen rates for the perturbation phase seem somewhat too specific, and I got slightly confused with the 'ratio of west/east channel incision' part. I suggest performing a sensitivity analysis again as you need to test how other parametrisation affects simulations. 3) Have the 'terrace uplift rates' calculated in the Study Area section played any role in the parametrisation of the models? If not, why not? What was the role of these calculated terrace uplift rates in the manuscript? | Thanks for your suggestion. We are currently conducting a sensitivity test on certain parameters, as previously mentioned. The result of this analysis will be included in in the final manuscript. We will also revise the paragraph accordingly to ensure clarity and coherence.

 * [Lines 398–416] *During stage 2 (Quaternary reverse faulting), the average uplift rate is set to be the highest at the location of the fault and to diminish linearly with increasing distance from the fault. To determine the maximum vertical displacement per event, we assumed that a maximum earthquake magnitude of $M_W$ 7.0 once per 20 kyr (Slemmons and Depolo, 1986; Kyung, 2010), although different maximum magnitude estimates ($M_W$ 4.6–5.6) have been proposed for the UIsan Fault (Choi et al., 2014). According to the empirical equation of Moss and Ross (2011), a $M_W$ 7.0 earthquake would generate a maximum vertical displacement of approximately 2.36 m. Therefore, we hypothesised a scenario in which a $M_W$ 7.0 earthquake produces a maximum vertical displacement of 2.36 m every 20 kyr.* |
| Line 419: Suggestion: "initial topography (i.e., topography achieved after stage 1)" or something similar to improve clarity on those lines. | We will modify the sentence.

 * [Lines 418–420] *Comparisons between the resultant topographies from Case A1 to Case B1 and from Case A2 to Case B2 allow us to detect the influence of initial topography (i.e., topography achieved after stage 1) on the subsequent geomorphic response to the same pattern of tectonic movement (i.e., uplift by faulting during stage 2).* |
| Line 422: Substitute "geomorphic indices analysis" for "topographic analysis" and change "verify" to | We will modify the sentence.

 * [Lines 422–423] *In addition, our model results can be used to compare our results obtained* |

| "compare." | from topographic analysis, CADRs, and channel incision rates calculation from [10]Be measurement, as these were used as inputs for the simulation. |
| --- | --- |
| Lines 425-426: Rephrase these sentences. Suggestion: "… quantitatively compare the simulated topography generated in the four cases". Delete "to compare the modelled topographies with the observed topography in the study area" as you mention this on lines 422-423. | We will rephrase those sentences.

* [Lines 423–426] We analysed Gilbert metrics and the χ index for the modelled topographies using TopoToolbox (Schwanghart and Scherler, 2014) and DivideTools (Forte and Whipple, 2018) to quantitatively compare the topographies generated in the four cases. |

**4 Results**

| Comment | Reply |
| --- | --- |
| Line 430: The placement of figures makes reading the text more difficult. | We will insert the figures and captions in the main text, following the author's guide of this journal.

 Author's guide says: …Figures and tables as well as their captions must be inserted in the main text near the location of the first mention (not appended to the end of the manuscript) and the figure composition must embed any used fonts. …

We think the placement of Figure 5 does not harm a readability. So, we are going to keep the present status. |
| Lines 439-445: This paragraph needs significant reworking. Its current form offers little quantitative description of the patterns of normalised channel steepness or knickpoints. In fact, from lines 440-445, you provide more interpretation than results. I suggest you elaborate on a similar presentation of results as the one done for the CARD. For instance, you could write something like: "... We find that normalised channel steepness varies from X to X | We appreciate the constructive comment. We will add a general description of the $k_{sn}$ results and retain the original manuscript content because it provides a fundamental interpretation that should precede the '5 Discussion' section.

* [Lines 439–445] We find that $k_{sn}$ varies from 0 to 238 $m^{0.9}$, with a regional mean of 24 and a standard deviation of 16. Values lower than the regional mean $k_{sn}$ are observed in the lowland of incised valley on the western flank. Values higher than the regional mean $k_{sn}$ appear from the foothill of the mountain range. Analyses of $k_{sn}$ and knickpoints on the longitudinal and χ-transformed stream profiles show that the channels on both (western and eastern) sides are in a transient state (Fig. 5). (…) The remaining knickpoints can be interpreted as being caused by |

| | |
|---|---|
| m0.9, with a regional mean of X and a standard deviation of X. Low values (ksn < X) are observed in. High values (ksn > XX) are ..." | *tectonic events, and are in accordance with the findings of a previous study (Kim et al., 2016a), which suggested on the basis of a 1-D model that the observed major knickpoints in the study area cannot have been formed by sea level changes since the global Last Glacial Maximum.* |
| Lines 441-442: It is necessary to detail how this excluding of artefact knickpoints was conducted within the Methods section with enough detail to ensure reproducibility. In addition, please provide quantitative statistics for the presence of 'artefact' and lithological knickpoints. How many lithological knickpoints did you extract? What is their spatial distribution? At each lithological transition, do you observe knickpoint clustering? Does every river crossing such lithological transitions perpendicularly show knickpoints? Do they show similar magnitudes? | As we mentioned in that sentence, we excluded artefact knickpoints and those associated with lithological boundaries by manually examining satellite images and geological maps, and conducting field observations. This means that we manually confirmed whether automatically identified knickpoints coincided with the known artefacts and/or lithological boundaries, rather than relying on quantitative statistics from the toolkits for exclusion. In addition, we found clustering of knickpoints at channels that intersect the lithological boundaries. We have revised the sentences to clarify these methods and observations.
**\* [Lines 441–442]** *We manually excluded artefact knickpoints (e.g., known anthropogenic features such as dams and reservoirs) and lithological boundaries by examining satellite images and geological maps and field observations.* |
| Lines 443-445: The finding from previous literature that major knickpoints in the study area cannot be driven by eustatic changes should have been presented and explained earlier in the text. | We appreciate your suggestion. It would indeed have been beneficial to introduce this information earlier in the text, possibly in the sections of study area or method. However, the study area section primarily focused on describing regional tectonic history and local landforms, while the method section is concentrated on techniques used. So, we chose to include the mentioned finding in the context of discussing knickpoint extraction and differentiating tectonically induced knickpoints from those caused by artefacts and lithological boundaries. We could manually remove those two types of knickpoints, as we mentioned in the previous sentence. The sentence in question aimed to clarify why the major knickpoints in the study area are unlikely to have resulted from base-level changes. So, we believe this approach maintains the flow and relevance of the information within previous scholarships to our findings. |
| Lines 455-460: Please rework this paragraph. First, | Thank you for your suggestion. First, we will describe the patterns of each metric in |

| | |
|---|---|
| it is better to describe the observed pattern per topographic metric. For instance, when you mention that you have a high value, it is necessary to be explicit. How high? Moreover, I am confused about which ways Fig. 7 is different from Fig. 8. Having many variables that arguably serve the same function (i.e., Gilbert metrics to gauge cross-divide relief asymmetry) makes the visualisation and description of the results worse. | detail. Additionally, we will merge Figs. 7 and 8 into a single figure and omit the channel head elevation and gradient plots, as they are unnecessary.

 * [Lines 455–460] *We plotted our topographic analysis results (Fig. 6) to determine whether and if so, how the topographic metrics vary along and across the MDD (Fig. 7). The along-MDD variation in each topographic metric shows the relative highs and lows. The swath profile exhibits the highest peak around 42 km of the horizontal axis and relatively high peaks around 25, 51, and 60 km (Fig. 7a). The western $k_{sn}$ with a base-level elevation of 50 m shows the highest value 59–70 km of the horizontal axis. One with a base-level elevation of 200 m has the higher values around 43 and 59–70 km of the horizontal axis. The western relief with a base-level elevation of 50 m exhibits the higher values around 65 and 72–80 km, and one with a base-level elevation of 200 m has higher values around 43 and 63 km. Lastly, the western $\chi$ index with a base-level elevation of 50 m shows the highest peak around 41–50 km of the horizontal axis, and one with a base-level elevation of 200 m has relatively small variation along the MDD but shows the higher values 32–42, 53, and 60 km of horizontal axis.* |
| Line 478: 'Significant' here means statistically significant? | We will modify the sentence.

 * [Lines 478–479] *There are some statistically significant differences in topographic metrics between those for the western and eastern flanks along the MDD (Figs. 8a–8e).* |
| Lines 484-486: Higher by how much? | We will add the details.

 * [Lines 484–486] *Values of $k_{sn}$ for the eastern-flank channels are up to 200 % higher than those for the western-flank channels within the 0–60 km section, whereas those for the western-flank channels are up to 137 % higher than those for the eastern-flank channels within the 60–90 km section.* |
| Lines 509-510: Please elaborate more on the sentence: "This pattern contrasts with the main spatial trend of CADR but corresponds to the patterns shown by the other geomorphic indices (Figs. 7 and 8)." | We will add some details.

 * [Lines 509–510] *These higher CADRs than their adjacent ones contrast with the main spatial trend of CADR which decreases towards the both ends of the MDD. However, these higher CADRs corresponds to the pattern of topographic metrics, such as mean upstream relief and $k_{sn}$, which also increase.* |
| Lines 510-511: You state that the CADRs on the | Comparing each pair of basins, the difference in CADRs between western and eastern |

| | |
|---|---|
| western flank river basins are generally higher than those on the eastern flank. How much higher? | catchments varies, depending on the samples. So, it is difficult to specify an exact value. Instead, we have included the maximum value.

**\* [Lines 510–511]** *Second, CADRs on the western flank are up to ~100 mm kyr-1 higher than those on the eastern flank.* |
| Lines 512-516: These sentences need reworking. No previous investigation or results showed potential influences of lithology on CADRs. In addition, Fig. 1b does not show a clear potential explanation, as suggested here. As I mentioned before, I suggest adding some form of analysis of catchment lithology to the paper. | We appreciate your suggestion. We agree the current sentences may mislead readers. Therefore, we will remove them. |
| Line 546: Suggestion: Use 'model domain' instead of 'modelled areas'. | We will make a change throughout the manuscript.
**\* [Lines 545–546]** *The MDDs of the initial topographies in Cases A#, which were the models simulated using spatially uniform uplift rate during stage 1, occupy their positions in the centre of the modelled domains (Figs. 10a and A1a).*
**\* [Lines 342–343]** *(b) The four model cases (A1-A2, B1-B2) used to test different conditions of spatial uniformity/non-uniformity of uplift during stage 1 and the width of the modelled domain.*
**\* [Lines 361–362]** *With this model structure, we tested four cases differentiated by varying two parameters: (1) spatial uniformity of uplift rate in the first stage, and (2) the width of the modelled domain (Fig. 4b).*

We will also revise the other parts of this manuscript in the same way. |
| Lines 545-552: It is important to realise that the integral metric χ should be calculated differently if the background rock uplift is not spatially uniform in a formulation accounting for the spatial gradient in uplift. This is explained in detail in Willett et al. (2014). Additionally, there is a blog entry in the | Thanks for sharing useful information for calculation of χ index in nonuniform condition. We fully acknowledge the importance of calculating χ index in a different manner. We will recalculate the χ index following Eq. (5) of Willett et al. (2014) and update the result. |

| | |
|---|---|
| TopoToolbox blog that elaborates on that, introducing an algorithm tailored to perform such a calculation using TopoToolbox's dependencies:

(https://topotoolbox.wordpress.com/2020/11/13/use-of-chi-analysis-in-experimental-landscapes-dulab/) | |
| Lines 550-552: This sentence can use rephrasing to improve clarity and readability. | We divided this sentence into two sentences for clarity and readability.
**\* [Lines 550–552]** *The initial topographies of Cases A# and Cases B# exhibit differences in modelled positions of the MDDs and the patterns of χ indices. These differences are likely due to the variation in the spatial uniformity of uplift rate during stage 1 (uniform vs. non-uniform; Fig. 4b).* |
| Lines 562-565: I was slightly confused with the phrasing here. What do you mean? These two sentences can use some reworking. | We will remove these sentences to avoid any confusion. |
| Lines 585-586: Please elaborate more on this sentence: "higher sensitivity of MDD to fault slip in Case B2 may be attributable to its shorter channels compared with Case B1". Why? | We will revise the sentence for greater clarity.
**\* [Lines 585–586]** *The heightened sensitivity of the MDD to uplift in Case B2 can be attributed to its shorter channels compared to those in Case B1, allowing the signal of fault activity to propagate more quickly from downstream to upstream.* |
| colspan="2" **5 Discussion** | |
| **Comment** | **Reply** |
| Line 627: 'Areas with lower swath profile' is awkward. What does a lower swath profile mean? A swath profile shows mean, maximum, and minimum elevation values. How lower? In addition, it would be beneficial to be explicit about what a | We will revise the sentence to enhance clarity.
**\* [Lines 627–629]** *Areas along the MDD where the swath profile, $k_{sn}$, and relief values are relatively lower compared to other parts are interpreted as zones of lesser tectonic activity.*

Yes, we calculated bedrock uplift rate with strath terraces and CADRs from riverine sediments. However, the topographic metrics we analysed represent the quantitative |

| | |
|---|---|
| 'lower degree of tectonic intensity' means. To parametrise your model scenarios, you estimated some average long-term surface uplift rates at the fault values. How about being more precise in analysing the results and comparing them with those? | characteristics of **cumulative** topography integrating both initial topography and recent tectonic movements, including uplift due to faulting along UFZ). So, directly comparing the uplift 'rates' or denudation 'rates' with topographic metrics may not always yield precise. |
| Lines 654-656: Again, it is necessary to introduce these key results from previous studies early in the text. This feels important. They should not appear out of the blue in the discussions. | We will add this information in the '2 Study area' section and will reference it again in this part for comparison with our result.
 **\* [Lines 100–101]** *This definition includes some strands of exposed faults along the mountain front and several strands of buried faults near the centre of the incised valley (Fig. 1b). They also suggested the UFZ can be divided into northern and southern segments based on the differences in fault-hosting bedrocks and width of the deformation zone. The northern part of the UFZ consists of Late Cretaceous to Paleogene granitic rocks and has wide deformation zone, while the bedrock of its southern part is composed of Late Cretaceous sedimentary rocks and the deformation is focused along the narrow zone (Cheon et al., 2023).* |
| Line 656: Delete 'as follows'. | We will revise the sentence.
 **\* [Lines 656–657]** *We attribute this difference to the different segmentation criteria used and argue that our geomorphic-based fault segmentation has several advantages.* |
| Lines 656-664: These sentences are confusing and should be reworked. It would be more beneficial to discuss the reasoning behind Cheon et al.'s (2013) segmentation of the UFZ in relation to your findings. | We agree the current text reads confusing. We will revise the sentences to enhance clarity and coherence.
 *\* [Lines 670–673] A recent study (Cheon et al., 2023) also divided the incised valley containing the UFZ on the basis of: (1) differences in fault-hosting rocks, and (2) width of the deformation zone. These authors segmented the UFZ into only two parts, with the division occurring between what we identify as segments 3 and 4 in the current study.* |
| Lines 667-670: Please rephrase the sentences "The χ index represents the longer-term view for topography owing to its reliance on the integral method from the far downstream to the channel | Thanks for recommending nice phrasing. We will rephrase the sentences.
 **\* [Lines 667–670]** *The χ index is suitable for indicating potential future divide mobility, while cross-divide differences in mean upstream relief are better suited to evaluate short-term divide mobility (Forte and Whipple, 2018; Zhou et al., 2022).* |

| | |
|---|---|
| head (Forte and Whipple, 2018; Zhou et al., 2022). Other geomorphic indices, such as mean upstream gradient and relief, respond sensitively to". This can be better elaborated. For instance, the phrasing of 'longer-term view for topography' is poor. Perhaps explain that the χ method is well suited to assess the long-term stability of drainage divides, while cross-divide differences in steepness are better suited to evaluate short-term divide stability. | |
| Lines 670: I disagree for the reasons I explained when discussing the Methods section. I find the modelling exercise's parametrisation somewhat arbitrary, and other parameter values need to be tested. | We have started sensitivity analysis for stream-power parameters, such as K, m, and n. Except for those values, the remaining values, such as distance between the UFZ and the MDD and uplift rate, are indeed within a reasonable range. We will include the result of sensitivity analysis as supplementary data. |
| Lines 672-673: I feel that it is necessary to elaborate more extensively on uncertainties associated with the modelling exercise. | We also agree that it is important to be aware of the uncertainties associated with the modelling, as you mentioned. To address this, we will add a discussion of an additional uncertainty in our modelling.

 **\* [Lines 670–673]** *Although we employed realistic settings for all boundary conditions in the models based on a comprehensive understanding of the tectonic, geological, and geomorphic processes in the study area, it is acknowledged that there are likely to be discrepancies between the modelled and actual settings of variables (e.g., coefficient of erosion, uplift rate, and its spatial gradient)* *and epistemic uncertainties*. |
| Lines 673-675: I could not follow these sentences: "Comparing geomorphic indices that are sensitive to minor variations in boundary conditions could lead to a misinterpretation of the geomorphic evolution. For these reasons, we chose to focus on | We will remove these sentences to avoid any confusion. |

| | |
|---|---|
| a comparison of the pattern of χ indices." | |
| Lines 676-678: I could not follow the ideas expressed in these sentences. Why do the variations in morphology along the MDD make comparing the morphology on the easter-western flanks of the MDD difficult? | We will remove these sentences to avoid any confusion. |
| Line 678: Maybe I am missing it, but I do not recall the mean values per each topographic metric presented earlier in the text. Because of this, I find it hard to follow this statement. | This paragraph addresses the potential risk of comparing each topographic metric across the entire western and eastern flanks of the MDD. For clarity, we will rephrase this paragraph.
 **\* [Lines 676–682]** *Since the topography along the MDD varies significantly, each metric (e.g., mean upstream relief or χ index) will encompass a broad spectrum of values. Comparing these means and standard deviations from the western and eastern flanks across the entire MDD might mask any genuine differences between the flanks, leading to a 'Type II error (false negative: failing to detect a real difference)'. Therefore, we compared each topographic metric from the western- and eastern-flank segment by segment.* |
| Lines 676-682: This paragraph was challenging to follow (and I have not fully understood it). This needs rework. | |
| Lines 683-688: These sentences were complex to follow and felt repetitive (perhaps that should belong in the Results section). What does 'inconsistent' here mean? "In contrast with all other geomorphic indices, differences between the western-flank and eastern-lank χ index values are inconsistent." | The term "repetitive" is not applicable to these sentences as they discuss geological segmentations introduced in section '5.1 Segmentation of the UFZ'; therefore, this content cannot be presented earlier. In this context, a "consistent" pattern in χ index values would indicate that one side of a drainage exhibits lower χ index values than the opposite side, correlating with higher CADR and mean upstream relief. This pattern suggests greater stream power, promoting divide migration away from the drainage. However, to better convey the relationship between these geomorphic indices, we will replace "consistent" with "coupled" and clarify the sentence accordingly.
 **\* [Lines 683–685]** *For segment 2–5, all topographic metrics, except for χ index, generally show a coupled pattern (higher western-flank mean upstream relief and CADR), which indicate higher erosion rates on the western flank (Fig. 9b).* |
| Line 688: What do you mean by inconsistent | Here, 'inconsistent' refers to variations in the χ anomaly between western and eastern |

| | |
|---|---|
| pattern in χ? And why is it 'decoupled' (is this the best word here?) from catchment-averaged denudation rates? | flanks for each segment. For example, for segments 1 and 2, the χ indices are higher on the western flanks than on the eastern ones. In contrast, for segment 3, the χ indices on both flanks are statistically similar. For segments 4 and 5, the situation reverses, with the lower χ indices on the western flanks compared to the eastern ones. We have documented these variations as 'inconsistent patterns in χ'. We acknowledge our phrasing could be clearer. Therefore, we will revise the sentence to improve understanding.

**\* [Lines 688–689]** *This inconsistency of χ anomaly throughout the study area is related to its decoupling from CADR, channel incision rate, and mean upstream relief in segments 1 and 2.* |
| Line 690: I suggest using 'agrees' or 'consistent' instead of 'coupled' or 'decoupled' throughout the text. | As we demonstrated in our previous two responses, the terms 'consistency' and 'inconsistency' of χ anomaly differ from the 'coupled' and 'decoupled' χ indices. The former terms refer to whether 'the pattern of χ anomaly is (not) consistent across all segments.' The latter ones describe 'the implications χ index in relation to other metrics.' For example, when χ index, channel incision rate, CADR, and mean upstream relief are all higher in the western flank, but only the χ index indicates a lower erosion rate for the western flank, we refer to this situation as 'decoupled χ'. Therefore, we maintain to use 'coupled' and 'decoupled'. |
| Lines 683-694: This paragraph needs rework to improve clarity and readability. | We will rework this paragraph for clarity and readability.

**\* [Lines 683–694]** *For segment 2–5, all topographic metrics, except for χ index, generally show a coupled pattern (higher western-flank mean upstream relief and CADR), which indicate higher erosion rates on the western flank (Fig. 9b). In contrast with mean upstream relief and CADRs, the χ anomaly between western and eastern flanks for each segment is inconsistent. The western-flank χ indices in segments 1 and 2 are higher than those of the eastern flank ($p$-value < 0.05), the same as those of the eastern flank in segment 3 ($p$-value > 0.05), and lower than those of the eastern flank in segments 4 and 5 ($p$-value < 0.05). This inconsistency of χ anomaly throughout the study area is related to its decoupling from CADR, channel incision rate, and mean upstream relief in segments 1 and 2. The χ indices in segment 1 are decoupled from the higher CADR and incision rate on the western flank, and those in segment 2 are decoupled* |

| | |
|---|---|
| | *from not only CADR and incision rate but also the mean upstream relief. These decoupled χ indices in segments 1 and 2 (i.e., lower χ indices on the eastern flank of TMR) contradict what would be generally expected from the higher CADRs, channel incision rate, and mean upstream relief on the western flank compared with the eastern flank.* |
| Line 695: "To facilitate the investigation of the geomorphic evolution of the study area" reads awkwardly here. I suggest some rephrasing. | We will revise the sentence to enhance clarity.
 **\* [Lines 695–696]** *To clarify our landscape evolution modelling approach, we grouped the five proposed segments into two distinct sections corresponding to the northern and southern parts of the UFZ.* |
| Lines 695-701: I am confused. You classified the UFZ into two segments (north and south). Isn't this the precise classification you criticised in the paragraph starting at line 654? | Thank you for your constructive comment. The grouping of the five segments we proposed is just for simplification of landscape evolution modelling: a wider northern part and a narrower southern part. That is why we used the term 'part', not the 'segment' in this context. We defined the boundary between the northern and southern parts at the boundary between segment 2 and segment 3. However, the recent study (Cheon et al., 2023) proposed only two segments, with their boundary placed between our segment 3 and segment 4. We will revise the text to clarify this distinction.
 **\* [Lines 655–656]** *These authors divided the UFZ into only two segments, with the division occurring between what we identify as segments 3 and 4 in the current study.* |
| Lines 722-723: By simply multiplying the integral metric by K, one can estimate channel response timescales (e.g., Gallen, 2018). As such, you could bracket reasonable channel response timescales for the UFZ, effectively testing this hypothesis. | Thank you for your head-forward comment. We agree the calculation of channel response time scale should benefit the better understanding of USF evolution. However, the content is beyond the scope of the current study. So, we will leave the analysis for the next, follow-up study. |
| Lines 724-726: A complete sensitivity test for different parametrisations for rock uplift is necessary for supporting this statement. | **[Lines 724–726]** Topographic metrics might be expected to have responded less sensitively to uplift in segment 1 because of its lower tectonic activity than that of segment 2.

 This statement is saying the general premise that the geomorphic response would be more sensitive and bigger if the tectonic activity is higher. For example, fluvial erosion and hillslope diffusion processes would be faster in the tectonically more active |

| | regions. This general premise could be verified if we performed the sensitivity test for the rock uplift rates, but we considered that this is out of scope of this study. |
|---|---|
| Lines 729-734: Considering the very low K value used in the simulation, has the landscape achieved a steady state in this modelled scenario with only 3 Myr of model run? What was the criteria for defining steady state here? This feels awkward as my simulations take me much more time (hundreds of Myr) to achieve a steady state if I use K values similar to yours. It would have been nice to have snapshots of erosion rates presented to the reader after phase 2 of the simulation. | Thank you for your observations regarding the duration of the model phases and the associated K values used in our simulation. We acknowledge the concerns about the equilibration time necessary to reach a steady state in our landscape evolution model.

In our simulations, we observed that the modelled landscape achieved a dynamic equilibrium state (i.e. no more discernible change in elevation with time) approximately 2.4 Ma into the stage 1. This quicker equilibrium is attributed to the relatively low uplift rate (80 mm $kyr^{-1}$) and to the coarse grid spacing (100 m) used in our model. These conditions facilitated a faster approach towards equilibrium within the prescribed time frame of the stage 1.

Regarding the duration of the stages, extending the time frames for stages 1 and 2 may introduce complexities such as defining an appropriate duration that accurately reflects the geological settings of the UFZ. It raises questions about the representativeness of prolonged simulation durations for our specific study context.

To further substantiate our model settings and address your concerns, we are currently conducting sensitivity analyses concerning the K value. This analysis aims to elucidate the impact of varying K on the time scales required for reaching equilibrium. The results of these sensitivity tests will be included as supplementary data to provide comprehensive insights into the effects of these parameters on our simulation outcomes. We believe these efforts will enhance the robustness and relevance of our findings to the unique geological characteristics of the UFZ. |
| Lines 751-793: You are missing a big opportunity to use your paired catchment-averaged denudation | Thank you for your suggestion. However, we believe that including calculations of divide retreat rates and the corresponding discussion would extend beyond the scope |

| | |
|---|---|
| rates to estimate drainage divide retreat rates using an approach similar to Hu et al. (2021) and Stokes et al. (2023). | of the current study. We intend to maintain our focus on the applicability of geomorphic indices to delineate fault segments. |
| Lines 761-762: Use 'indicates eastward divide migration' instead of 'is related to the...' | We will revise the sentence.
**\* [Lines 761–762]** *The higher $k_{sn}$ on the western flank indicates eastward divide migration, whereas the higher χ index on the western flank indicates westward divide migration.* |
| Lines 779-781: This is difficult to follow; please elaborate more: "Therefore, we interpret that the streams flowing within the drainage in the vicinity of the MDD and the elevated ridge on the western flank of segment 1 are the results of antecedent streams." | We will revise the sentence.
**\* [Lines 779–781]** *Therefore, we interpret that the streams flowing within the internal sub-basin surrounded by the MDD and the elevated ridge on the western flank (Fig. 12c) are the antecedent streams, flowing east to west.* |

**6 Conclusion**

| Comment | Reply |
|---|---|
| Lines 794-826: The conclusion is way too long. Rework is necessary. | We will make the section 'Conclusion' brief. Especially, we will remove the fourth paragraph and detailed explanations in the second and third paragraphs (total of 363 words).

**\* [Lines 794–826]** *The Ulsan Fault Zone (UFZ) has been one of the most active fault zones on the Korean Peninsula since its reactivation ~ 5 Ma. Our study area, the eastern, mountainous, hanging wall block of the UFZ, has undergone regional uplift under an ENE–WSW-oriented neotectonic maximum horizontal stress after 5 Ma. This study aimed to evaluate the relative tectonic activity along the UFZ, characterise the past and present geomorphic processes operating along the UFZ, and infer landscape evolution patterns in response to tectonic perturbation involving reactivation of the UFZ.*
*We evaluated the relative tectonic activity along the fault zone using topographic metrics, and catchment-averaged denudation rates (CADRs) and bedrock incision rate derived using in situ* |

*cosmogenic [10]Be. We divided the UFZ into five geological segments based on the relative tectonic activity that we assessed. This study represents the first segmentation result based on the relative tectonic activity of the UFZ inferred from topographic metrics.*

*We also interpreted the tectono-geomorphic evolution of the study area by modelling landscape evolution and comparing the values and patterns of topographic metrics of the modelled topography with those observed in the study area. We interpret that the northern UFZ (segments 1 and 2) underwent regional asymmetric uplift (westward tilting) prior to Quaternary reverse faulting since ~ 2 Ma. The southern UFZ (segments 3–5) was negligibly affected by asymmetric uplift before Quaternary reverse faulting, as channel lengths (distance between the Ulsan Fault and the channel head) were sufficiently short to adjust quickly to the uplift. Our analysis and interpretation of the tectono-geomorphic evolution of the UFZ show that inherited topography can influence the subsequent geomorphic processes and topographic response to neotectonic reverse fault slip. The topographic metrics we utilized can therefore be regarded as characterising not only the present topography, but also as holding information resulting from the accumulation of a history of tectonic and erosion.*

*Our study clearly demonstrates that topographic metrics can be used to infer differential tectonic activity (i.e., variable fault slip and surface uplift) and that modelling can be used to infer possible influences of inherited topography in intraplate regions with extremely low strain rates and fault slip rates, and extremely high erosion rates.*

**Figure Comments**

| Comment | Reply |
|---|---|
| Fig. 1: Panel A should prioritise showing topography rather than satellite imagery. In Panel B, including sampling sites and sampled catchments is necessary. Additionally, it would be important to show the Taebaek Mountain Range somehow. | In panel (a), we used the satellite image as it clearly illustrates the incised valleys along the major fault zones. In addition, the topography of study area is depicted in Figure 2. In panel (b), we will include the upstream areas of the CADR sampling sites. We will add Taebaek Mountain Range in the inset of panel (a) as well. |

[Figure]

(a)

China

Taebaek Mountain Range

Yellow Sea

East Sea

Fig. 1a

$M_w$ 5.4
(15 Nov. 2017)

Pohang

Gyeongju

Fig. 1b
Study area

$M_w$ 5.5
(12 Sep. 2016)

$M_L$ 4.0
(30 Nov. 2023)

Jain Fault

Miryang Fault

Moryang Fault

Yangsan Fault

Dongrae Fault

Ilgwang Fault

Ulsan Fault

East Sea

Pusan

★ Epicenter

— Major fault zone

N

50 km

(b)

129° 20' E   129° 30' E

35° 50' N

35° 40' N

35° 30' N

N

10 km

[UFZ]

——— Exposed fault

------ Buried fault

——— Other faults

○ Active fault outcrop/trench site

——— Main drainage divide of hanging wall

▢ CADR sampled catchments

▢ Quaternary fill and unknown area

▢ Miocene sedimentary and extrusive rocks

▢ Paleogene intrusive rocks

▢ Paleogene rhyolitic to dacitic volcanic rocks

▢ Cretaceous intrusive rocks

▢ Cretaceous rhyolitic to dacitic volcanic rocks

▢ Cretaceous sedimentary and volcanoclastic rocks

| Fig. 2: The river network in panel A feels strange. Instead, I suggest extracting (and showing) all rivers starting at the baselevel elevation of 0 m. Including sampling sites and sampled catchments is necessary. Why are the marine terrace uplift rates exhibited here? I do not recall them being discussed further in the text. I would also add the swath profile centre lines here. | We displayed only major streams and rivers in panel (a), but we will now illustrate the complete river system starting at the base level elevation of 0 m. The sampling sites and catchments are marked in other figures (Figs. 1b, 3a, and 3b). Therefore, we do not think it is necessary to show them in Fig. 2a. We have already noted the uplift rates of marine terraces as they were introduced in previous studies in the '2 Study area' section (lines 135–138) and provided the rationale for designing the landscape evolution model in the '3 Methods' section (lines 370–373). Additionally, there is no need to add the center line of the swath profile here, as the MDD itself serves as the center line for extracting the swath profile. |

[Figure]

| | |
|---|---|
| Fig. 4: While the concept is promising, the figure's complexity makes it challenging to grasp. Simplifying the visualisation, possibly by presenting the setup in a plan view, would improve clarity and comprehension. | In panel (c), we aimed to demonstrate the uniformity and spatial gradient of uplift rates during stages 1 and 2 for all cases. To facilitate comparison between the uplift rates of each case, we opted to present them as a line plot across the E–W direction. This choice was made because a plan view would require additional color bars to effectively express the variations in uplift rates. We will consider which presentation method better conveys the setup. |
| Fig. 5: 1) It is challenging to visualise patterns in channel morphology in panel a due to the drainage networks' incompleteness. I suggest extracting rivers assuming baselevel = 0 m, ensuring river networks are complete, extending downstream until the ocean. Including sampling sites and sampled catchments is necessary.

2) The interpretation of knickpoints in river profiles appears flawed. First, you are likely considering concavities in the long-profile as knickpoints (i.e., points identified by a downstream along-profile decrease in ksn). 'Concave' knickpoints should not be identified here. Additionally, there are many instances in panel D of significant along-profile breaks in channel slope that were not identified as knickpoints (e.g., around 200 m of elevation at a distance slightly below ten and slightly above five), and they should be. I guess these results are caused by the parametrisation used to extract knickpoints from topographic data. Furthermore, the exclusion of knickpoint at artefacts and | We appreciate your comments.

1) We did not use a base-level elevation of 0 m to extract the drainage network. This approach was avoided because it results in drainage networks that are too extensive, which complicates the description of topographic metric variations along the UFZ. The sampling sites and catchments are already marked in other figures (Figs. 1b, 3a, and 3b).

2) When extracting knickpoints using the algorithms of Mudd et al. (2014) and Gailleton et al. (2019), we also identified 'concave' knickpoints. Despite numerous attempts and parameter adjustments, these concave knickpoints were not always consistently eliminated. We manually excluded knickpoints associated with artefacts and lithological boundaries without arbitrariness. Additionally, we understand that $k_{sn}$ represents the slope profile in χ–z space. Therefore, we calculated $k_{sn}$ for each river segment and represented it with a pink-coloured line. If this affects readability, we are open to removing it. |

lithological boundaries also appears arbitrary. In summary, rework is necessary to ensure that knickpoints are identified accurately and consistently along the profiles. Finally, change the caption for the 'X-ksn plot'. Ksn is the slope of the profile in elevation-chi space.

[Figure]

Fig. 6: This figure's current format is not helpful for

Thank you. We have decided not to conduct a sensitivity analysis on how changing the

| | |
|---|---|
| the manuscript. I suggest that the sensitivity tests on how changing base level elevation affects chi patterns are presented as supplemental material. If you want to show spatial patterns on other channel morphology metrics, I suggest you depict complete river networks. I strongly recommend quantifying relief for each pixel for your DEM rather than 'channel relief', given that ksn is already a robust measure of local channel slope normalised by upstream drainage area. | base-level elevation affects the $\chi$ index because it requires significantly more time than anticipated and is beyond the scope of the current study. Previous studies and the governing equations already demonstrate that the $\chi$ index is influenced by base-level elevation. Additionally, calculating the relief for each pixel of the DEM is straightforward, but this data is not utilized throughout this manuscript. While $k_{sn}$ provides information on local channel slope, it may differ from the mean upstream relief at the channel head. |
| Fig. 7 and 8: Combine these two figures into a single figure, starting with the swath profile in panel a, followed by variations in chi, ksn, and upslope hillslope relief. Consider omitting unnecessary metrics to streamline the presentation. | We will combine Figs. 7 and 8 into a single figure and omit the channel head elevation and gradient plots in the combined figure. |

[Figure]

| Fig. 9: In Panel A, prioritise topography over satellite imagery for better visualisations of landscape features. For clarity, Panel B should focus on chi and mean upstream relief only. | We will change the basemap to a DEM to visualise the topography in panel (a). In panel (b), we will omit the channel head elevation and gradient plots. |
|---|---|
| |  |
| Fig. 10-12: I assume these figures will undergo | We will omit the channel head elevation and gradient plots. In addition, we will update |

| significant changes after revision. | the χ index calculation result after re-calculating it following Willett et al. (2014). |
|---|---|

**References**

Cao, K., Mai, H., Chevalier, M. L., and Wang, G.: Tectonic and Climatic Control on Quaternary Exhumation in the Eastern Pamir Domes, Western China: Insights From Geomorphic Approaches, Front. Earth Sci., 10, 1–20, https://doi.org/10.3389/feart.2022.839203, 2022.

Clubb, F. J., Mudd, S. M., Milodowski, D. T., Hurst, M. D., and Slater, L. J.: Objective extraction of channel heads from high-resolution topographic data, Water Resour. Res., 50, 4283–4304, https://doi.org/10.1002/2013WR014979.Reply, 2014.

Desormeaux, C., Godard, V., Lague, D., Duclaux, G., Fleury, J., Benedetti, L., and Bellier, O.: Investigation of stochastic-threshold incision models across a climatic and morphological gradient, Earth Surf. Dyn., 10, 473–492, https://doi.org/10.5194/esurf-10-473-2022, 2022.

Forte, A. M. and Whipple, K. X.: Criteria and tools for determining drainage divide stability, Earth Planet. Sci. Lett., 493, 102–117, https://doi.org/10.1016/j.epsl.2018.04.026, 2018.

Harel, M. A., Mudd, S. M., and Attal, M.: Global analysis of the stream power law parameters based on worldwide 10Be denudation rates, Geomorphology, 268, 184–196, https://doi.org/10.1016/j.geomorph.2016.05.035, 2016.

Hergarten, S., Robl, J., and Stüwe, K.: Extracting topographic swath profiles across curved geomorphic features, Earth Surf. Dyn., 2, 97–104, https://doi.org/10.5194/esurf-2-97-2014, 2014.

Hobley, D. E. J., Adams, J. M., Siddhartha Nudurupati, S., Hutton, E. W. H., Gasparini, N. M., Istanbulluoglu, E., and Tucker, G. E.: Creative computing with Landlab: An open-source toolkit for building, coupling, and exploring two-dimensional numerical models of Earth-surface dynamics, Earth Surf. Dyn., 5, 21–46, https://doi.org/10.5194/esurf-5-21-2017, 2017.

Keller, E. A. and Pinter, N.: Active tectonics, Prentice Hall Upper Saddle River, NJ, 338 pp., 1996.

Kirby, E. and Whipple, K.: Quantifying differential rock-uplift rates via stream profile analysis, Geology, 29, 415–418, https://doi.org/10.1130/0091-7613(2001)029<0415:QDRURV>2.0.CO;2, 2001.

Luo, Q., Schoenbohm, L., Rimando, J., Li, Y., Li, C., and Xiong, J.: Morphometric analysis of the North Liuleng Shan Fault in the northern Shanxi Graben System, China: Insights into active deformation pattern and fault evolution, Geomorphology, 440, 108862, https://doi.org/10.1016/j.geomorph.2023.108862, 2023.

Muscheler, R., Beer, J., Kubik, P. W., and Synal, H. A.: Geomagnetic field intensity during the last 60,000 years based on 10Be and 36Cl from the Summit ice cores and 14C, Quat. Sci. Rev., 24, 1849–1860, https://doi.org/10.1016/j.quascirev.2005.01.012, 2005.

Schwanghart, W. and Scherler, D.: Short Communication: TopoToolbox 2 - MATLAB-based software for topographic analysis and modeling in Earth surface sciences, Earth Surf. Dyn., 2, 1–7, https://doi.org/10.5194/esurf-2-1-2014, 2014.

Snyder, N. P., Whipple, K. X., Tucker, G. E., and Merritts, D. J.: Importance of a stochastic distribution of floods and erosion thresholds in the bedrock river incision problem, J. Geophys. Res. Solid Earth, 108, https://doi.org/10.1029/2001jb001655, 2003.

Stock, J. D. and Montgomery, D. R.: Geologic constraints on bedrock river incision using the stream power law, J. Geophys. Res. Solid Earth, 104, 4983–4993, https://doi.org/10.1029/98jb02139, 1999.

Theodoratos, N. and Kirchner, J. W.: Dimensional analysis of a landscape evolution model with incision threshold, Earth Surf. Dyn., 8, 505–526, https://doi.org/10.5194/esurf-8-505-2020, 2020.

Willett, S. D., McCoy, S. W., Taylor Perron, J., Goren, L., and Chen, C. Y.: Dynamic reorganization of River Basins, Science (80-. )., 343, https://doi.org/10.1126/science.1248765, 2014.

Yildirim, C.: Relative tectonic activity assessment of the Tuz Gölü Fault Zone Central Anatolia, Turkey, Tectonophysics, 630, 183–192, https://doi.org/10.1016/j.tecto.2014.05.023, 2014.

Zebari, M., Grützner, C., Navabpour, P., and Ustaszewski, K.: Relative timing of uplift along the Zagros Mountain Front Flexure (Kurdistan Region of Iraq): Constrained by geomorphic indices and landscape evolution modeling, Solid Earth, 10, 663–682, https://doi.org/10.5194/se-10-663-2019, 2019.

Zondervan, J. R., Stokes, M., Boulton, S. J., Telfer, M. W., and Mather, A. E.: Rock strength and structural controls on fluvial erodibility: Implications for drainage divide mobility in a collisional mountain belt, Earth Planet. Sci. Lett., 538, 116221, https://doi.org/10.1016/j.epsl.2020.116221, 2020.

---

## Author Response (AR2)

Reply for the comment on egusphere-2024-198 (Associate editor)

**Title: Topographic metrics for unveiling fault segmentation and tectono-geomorphic evolution with insights into the impact of inherited topography, Ulsan Fault Zone, Korea**

| Comment | Reply |
|---|---|
| Lines 10–11: You don't need to change this if you don't want to, but I think it might sound better if you replace the two instances of "the present" with "today's" in this sentence. | Thanks, we changed it.

**[Lines 10–11]** Quantifying today's topography can provide insights into landscape evolution and its controls, since it represents a cumulative expression of past and present surface processes. |
| Lines 12–13: Is this not redundant given the previous sentence? Consider revising. | We removed the type of fault but contained its strike and dip.

**[Lines 12–13]** The UFZ strikes NNW–SSE and dips towards the east. |
| Lines 13–16: rates | We changed it.

**[Lines 13–16]** This study investigates the relative tectonic activity along the UFZ and the landscape evolution of the hanging wall side of the UFZ, focusing on neotectonic perturbations using $^{10}$Be-derived catchment-averaged denudation rates and bedrock incision rates topographic metrics, and a landscape evolution model. |
| Lines 16: Consider deleting this sentence. It is inferred from the previous sentence. | We deleted this sentence. |
| Line 17: their | We changed it.

**[Line 17]** Five geological segments were identified along the fault, based on their relative tectonic activity and fault geometry. |
| Lines 44–46: I suggest replacing this with "has been" | We modified the sentence.

**[Lines 44–46]** It has been applied to determine whether a landscape under specific conditions is in a steady state or transient state, and to assess long-term drainage mobility (Willett et al., 2014; Forte and Whipple, 2018; Kim et al., 2020; Hu et al., 2021; Lee et al., 2021). |

| | |
|---|---|
| Lines 47–50: I would put an "e.g.," here, as this is a partial list. | Thanks, we added it.

**[Lines 47–50]** We can test the site-specific parameters constrained by empirical data (e.g., coefficient of diffusivity, coefficient of fluvial erosion efficiency, and local uplift rate) and determine a range of reasonable values through modelling (e.g., Tucker et al., 2001; Braun and Willett, 2013; Goren et al., 2014; Campforts et al., 2017; Hobley et al., 2017; Barnhart et al., 2020; Hutton et al., 2020). |
| Lines 58–59: "is" implied this is true of all landscapes. In some situations the traces of initial topography can be erased. I suggest replacing "is" with "can be". Alternatively you can specify that this hypothesis refers specifically to your study area. | We replaced "is" with "can be" and also constrained it to our study area.

**[Lines 58–59]** We hypothesize that the influence of inherited topography can be non-negligible in our study area where the slip rate is low, and the erosion rate is high, and topographic metrics would indicate it. |
| Lines 107–108: Can you add a few words about how these are calculated? That is, what measurements are the rates based on? C14 on the terraces? Cosmogenics? A few words here will suffice. | We already documented the details in Table 1 but did not make it clear here. We added the information on the dated material and dating method in this sentence.

**[Lines 107–108]** (a) Previously determined uplift rates (in mm kyr$^{-1}$) of marine terraces near the UFZ, based on the OSL ages of raised beach sediments (details about these rates are in Table 1; Choi et al., 2003a, b; Kim et al., 2007; Heo et al., 2014). |
| Lines 122–123: I would say "categorized based on their draining into the catchments either north or south of the valley floor divide". | We modified the sentence.

**[Lines 122–123]** The western-flank channels are categorized based on their draining into the catchments either north or south of the valley floor divide. |
| Lines 127–130: redundant, don't need it. | We will delete those words. |
| Lines 139–141: based on what? Evidently OSL. Say that here. | We added dating method information.

**[Lines 139–141]** Further, studies of marine terraces have proposed paleo-shoreline elevations and the OSL ages of beach-sediment layers for each terrace sequence (Choi et al., 2003a, b; Kim et al., 2007; Heo et al., 2014). |
| Lines 166–177: The normalised channel | We moved the Eq. (1) to the '3.3 Landscape evolution modelling' section and deleted the Eq. (2). Then, we |

| | |
|---|---|
| steepness is purely geometric. It is entirely defined by equation 3. It can be linked to equations 1 and 2, but equation 1 assumes some form of the erosion rule, which we know is, at best, an approximation, whereas equation 3 doesn't really include any assumptions: it is simply an empirical statement derived from topographic data.

You will use equations 1 and 2 later in the paper, but I think you should introduce those equations when you begin to talk about modelling, and for the section on $k_{sn}$ just start with equation 3 (since equations 1 and 2 are not used to measure $k_{sn}$). | defined the (normalised) channel steepness index with Eqs. (3a) and (3b) only related with the geometry.

**[Lines 166–177]** The channel under the steady-state condition in which the uplift, climate, and rock resistance are spatially uniform, maintains a graded profile, following a power-law equation (Hack, 1973; Flint, 1974):

[Lines 316–317] The bedrock channel incision rate, $E$, can be expressed by Eq. (3), which describes its relationship with channel bed shear stress (Howard and Kerby, 1983; Seidl and Dietrich, 1992; Sklar et al., 1998).
$E = KA^m S^n$  (3)
where $K$ is a dimensional coefficient of fluvial erosion efficiency with a unit of $[L^{1-2m}T^{-1}]$ encapsulating different controls on erosion, such as rock resistance, climate, bedload sediment grain size, and channel width length relationship (Stock and Mongomery, 1999; Whipple and Tucker, 1999; Snyder et al., 2000; Whipple and Tucker, 2002); $A$ $[L^2]$ is drainage area; $S$ $[L\ L^{-1}]$ is the slope; and $m$ and $n$ are exponents of drainage area and slope, respectively.

We also changed the equation numbers according to these modifications. |
| Lines 309–310: When you use version 3 to get dates, it outputs a version number. I did that on test data today (17 July) and the version number is 3.0.2. I recommend reporting this version number. | We added the version information.

**[Lines 309–310]** We calculated exposure ages using the CRONUS-Earth online calculator (Balco et al., 2008; version 3.0.2), applying the LSDn scaling scheme (Lifton et al., 2014). |
| Lines 319–321: These numbers vary a lot between sites. I'm curious why you didn't use the basin averaged erosion rates alongside the chi profiles to back-calculate K?
I agree with the reviewer that a sensitivity analysis would be welcome here (it doesn't need to be extensive, just some idea of how much the answers change if you pick 2 or 3 different values for these parameters, sensibly selected, and see | We carried out the simple sensitivity analysis for the parameters (e.g., channel slope exponent, channel concavity index, and erodibility coefficient) with several selected values. We attached all modelled results as an excel file (Supplementary 2) and their interpretation on how each model input parameter affects the modelling results as a text (Supplementary 1). Please refer to those supplementary files. |

| | |
|---|---|
| how they affect the result. The numbers you use are not arbitrary, but they are highly uncertain. | |
| Line 330: If you are using this equation in the model there is no need to report equation 2. | We deleted the equation 2. |
| Lines 433–434: Did you plot knickpoints in chi space? Or as a function of elevation (i.e., do a probability distribution of knickpoint elevation on the east and west flanks)? I ask because this could tell you how well knickpoints line up on one side of the MDD or across the MDD. | We plotted the knickpoints on the longitudinal profiles of the channels on the western and eastern flanks (Fig. 5d). However, we could not find any patterns in elevation of knickpoint's distribution on the either side of the MDD. |
| Line 469: In a basin, $k_{sn}$ can vary a lot. Do you have the variability of this metric? It would be useful to have some kind of uncertainty plotted here. | We changed the Figure 7b, adding the 1σ uncertainty of the normalised steepness index values on the western and eastern sides (base-level elevation of 50 m). We also modified the caption, according to the changes in the figure.

**[Line 469]** (b) Catchment-averaged $k_{sn}$. 1σ uncertainties of the $k_{sn}$ values extracted with the base-level elevation of 50 m on the western and eastern flanks are marked with the red- and blue-shaded areas, respectively. |

Seg. 1     Seg. 2     Seg. 3     Seg. 4     Seg. 5

(a) Elevation (m)

(b) $k_{sn}$ (m$^{0.9}$)

(c) Mean upstream relief (m)

(d) $\chi$ index (m)

(e) CADR (mm kyr$^{-1}$)

W1 / E1
W2 / E2
W3 / E3
W4 / E4
W5 / E5
W6 / E6
W7 / E7
W8 / E8

North   0   10   20   30   40   50   60   70   80   90   South

Distance along the main drainage divide from the north (km)

**Topographic metrics**
— Western side (base-level elevation = 50 m)
— Western side (base-level elevation = 200 m)
— Eastern side (base-level elevation = 50 m)

**Swath profile**
Mean elevation   Elevation range
Inferred segment boundary

**CADR (with 1σ uncertainty)**
Western side
Eastern side

| Figure 10: Awkward phrasing. Perhaps better to say "Channels present in stages 1 and 2"? | We changed it. |
|---|---|

We changed it.

**[Figure 10b]** Channels present in stages 1 and 2

| Line 651: Somewhere in this discussion there needs to be some explanation of why the strath-based incision rates are so different from the CADR rates. | We acknowledge the need of explanation on the difference between the (bedrock) channel incision rate and CADR. However, the section '5.2 Geomorphic evolution of the eastern block of the UFZ in response to tectonic movement' is not about the difference between them. So, we added several sentences in the caption of Figure 3, which shows the results of CADR and the exposure age of the strath surface.

**[Line 262]** The discrepancy between CADR and bedrock incision rate is tentatively caused by (1) the difference |

|  | between the integration time of CADR and the exposure age of strath surface and (2) the difference of spatial scales which is represented by those two methods. |
|--|-----------------------------------------------------------------------------------------------------------------------------------------------------------------|